# DNMT1-mediated regulation of somatostatin-positive interneuron migration impacts cortical architecture and function

Julia Reichard [1,2,14], Philip Wolff [1,2,14], Song Xie[3,4], Ke Zuo[3,5,6], Camila L. Fullio [7,8], Jian Du[1], Severin Graff[2,9], Jenice Linde[1,2], Can Bora Yildiz [1,2], Georg Pitschelatow[1,2], Gerion Nabbefeld [2,10], Lilli Dorp [1], Johanna Vollmer [1], Linda Biemans[1], Shirley Kempf[1], Minali Singh [11], K. Naga Mohan[11], Chao-Chung Kuo[12], Tanja Vogel [7], Paolo Carloni[3,4], Simon Musall[2,9,13] & Geraldine Zimmer-Bensch [1,2] ✉

The coordinated development of cortical circuits composed of excitatory and inhibitory neurons is critical for proper brain function, and disruptions are linked to a spectrum of neuropsychiatric disorders. While excitatory neurons are generated locally in the cortical proliferative zones, inhibitory cortical interneurons (cINs) originate in the basal telencephalon and migrate tangentially into the cortex. Here, we show that DNA methyltransferase 1 (DNMT1) is essential for the migration and integration of somatostatin (SST)-expressing interneurons in mice. *Dnmt1* deletion causes premature exit of SST⁺ cINs from the superficial migratory stream and alters the expression of key developmental genes. Unexpectedly, *Dnmt1*-deficient SST⁺ interneurons also exert non-cell-autonomous effects on cortical progenitor cells, resulting in subtle yet lasting alterations in cortical layering. These findings propose a role for DNMT1 in governing the migration of SST⁺ interneurons and mediating their instructive signaling to cortical progenitor cells, thereby shaping cortical architecture and influencing long-term network function.

The intricate neuronal circuitry of the mammalian cerebral cortex, composed of excitatory and inhibitory neurons, empowers its enormous cognitive capabilities. The circuit formation involves coordinated developmental steps: proliferation, differentiation, migration, morphological maturation, and synapse formation[1].

Excitatory cortical projection neurons originate from radial glial cells (RGCs) and intermediate progenitors (IPCs) located in the cortical

[1]RWTH Aachen University, Division of Neuroepigenetics, Institute of Zoology (Biology 2), Worringerweg 3, Aachen, Germany. [2]Research Training Group 2416 MultiSenses – MultiScales, RWTH Aachen University, Aachen, Germany. [3]Institute of Neuroscience and Medicine (INM-9) Computational Biomedicine, Forschungszentrum Jülich GmbH, Jülich, Germany. [4]RWTH Aachen University, Department of Physics, Aachen, Germany. [5]College of Pharmacy (International Academy of Targeted Therapeutics and Innovation), Chongqing University of Arts and Sciences, Chongqing, PR China. [6]Department of Physics, University of Cagliari, Cagliari, Italy. [7]Institute for Anatomy and Cell Biology, Department of Molecular Embryology, Faculty of Medicine, Albert-Ludwigs-University Freiburg, Freiburg, Germany. [8]Faculty of Biology, Albert-Ludwigs-University Freiburg, Freiburg, Germany. [9]Institute of Biological Information Processing (IBI-3) Bioelectronics, Forschungszentrum, Jülich, Germany. [10]RWTH Aachen University, Division of Neurophysiology, Institute of Zoology (Biology 2), Worringerweg 3, Aachen, Germany. [11]Molecular Biology and Genetics Laboratory, Department of Biological Sciences, BITS Pilani, Hyderabad Campus, Hyderabad, India. [12]Genomics Facility, Interdisciplinary Center for Clinical Research (IZKF), RWTH Aachen University, Aachen, Germany. [13]Institute of Experimental Epileptology and Cognition Research, University of Bonn, Bonn, Germany. [14]These authors contributed equally: Julia Reichard, Philip Wolff. ✉e-mail: zimmer@bio2.rwth-aachen.de

proliferative zones, through direct and indirect neurogenesis. Symmetric divisions of RGCs initially expand the progenitor pool, later shifting to asymmetric divisions to produce neurons and RGCs, or IPCs[1]. EOMES-expressing IPCs localize to the subventricular zone (SVZ) and generate neurons for all cortical layers after a few self-renewal divisions[2]. Cortical excitatory neurons are produced in a time-dependent manner, with deep-layer neurons born first and later-born neurons forming superficial layers, coming with unique functional features and connectivity[3]. Cortical neuron generation is modulated by both intrinsic and extrinsic factors[1,2]. Intrinsic mechanisms include (epi)genetic factors that dictate the developmental potential of progenitors. Extrinsic factors encompass signaling molecules from neighboring cells, postmitotic neurons in the cortical plate (CP), embryonic cerebrospinal fluid secreted by the choroid plexus, and incoming thalamic afferents[4–7]. These signals influence the timed self-renewal and neurogenic potential of cortical progenitors. Additionally, invading cortical inhibitory interneurons (cINs) have been proposed to influence EOMES[+] IPCs, thereby impacting the proper formation of cortical layers[8].

Despite composing only 10–20% of the neuronal population, cINs, including parvalbumin (PV) and somatostatin (SST) interneuron subtypes, are pivotal for precise information processing by inhibiting both excitatory and other inhibitory neurons[9]. Disturbing their development impairs the balance between excitation and inhibition, leading to abnormal cortical activity and cognitive impairments associated with various neurological and neuropsychiatric disorders, including schizophrenia and epilepsy[10,11]. The different types of cINs originate in three subpallial structures: the medial ganglionic eminence (MGE), the caudal ganglionic eminence (CGE), and the pre-optic area (POA)[9]. The MGE gives rise to PV[+] and SST[+] interneurons, whereas the CGE generates serotonin 3-A receptor (5HTR3a) subtypes[11]. The POA produces mostly neurogliaform cells and neuropeptide Y (NPY) expressing multipolar interneurons[11]. Postmitotic cINs migrate tangentially through the basal telencephalon into the cortex[11]. They invade the developing cortex along the marginal zone (MZ) and SVZ/intermediate zone (IZ), where they spread tangentially before switching to radial migration to enter the CP. This migration is guided by a diverse array of intrinsic and extrinsic cues[11–15]. For example, the transcription factors ARX and LHX6, expressed in MGE-derived cINs, directly and indirectly regulate the expression of cytokine receptors that mediate the attraction to CXCL12, which is expressed in the MZ migratory route of the developing cortex[16,17]. After tangentially spreading over the cortical areas, cINs invade the CP. The CP expresses NRG2, which exerts attractive effects on ERBB4-expressing interneurons[18].

Epigenetic mechanisms, including histone modifications and DNA methylation, modulate intrinsic transcriptional programs[10,19]. It was previously reported that the DNA methyltransferase 1 (DNMT1) regulates gene expression in postmitotic POA-derived cINs, promoting their proper migration and survival through non-canonical interactions with histone modifications[20]. Postmitotic MGE-derived cINs, which give rise to SST[+] and PV[+] interneuron subtypes, also express DNMT1[20]. However, DNMT1 function in these subsets of immature, migrating cINs is still unknown.

SST-expressing cells constitute about 30% of all cINs and play vital roles in inhibitory control, network synchronization, circuit plasticity, and cognitive functions[21]. SST[+] cINs encompass two main subtypes: Martinotti cells, which migrate along the MZ during development and form axon collaterals in layer I, and non-Martinotti cells, which disperse via the SVZ/IZ and have different layer distributions as well as targeting properties.

The distinct migratory paths of SST[+] interneurons are regulated by ARX and LHX6, as well as MAFB[16,22,23]. Specifically, MAFB marks Martinotti cells and, along with ARX and LHX6, directs their migration along the MZ[16,22,23]. This migration is closely associated with the subsequent formation of axonal arbors in layer I[23].

Recognizing the functional relevance of SST[+] cINs in cortical circuitry and the necessity of their precise development for cortical function, we investigated the role of DNMT1 in this process. Addressing this gap in knowledge advances the understanding of cell-type-specific epigenetic regulation in cortical network formation.

Our findings show that reduced *Dnmt1* expression in SST[+] cINs disrupts the transcriptional and DNA methylation profiles of key genes for cIN development, such as *Arx*, altering the migration of these cells along the MZ. This misregulation impacts excitatory cortical progenitor dynamics, affecting neurogenesis and cortical layering, ultimately leading to functional and behavioral abnormalities in adulthood.

## Results

### Molecular dynamics simulations propose DNMT1´s catalytic domain binding to unmethylated DNA

DNMT1 is described as a maintenance methyltransferase, preserving DNA methylation during replication. This view is largely based on a cryo-EM structure, showing a binding of DNMT1's catalytic domain to the hemimethylated oligo 5′-ApCpTpTpApCpGpGpApApGpGp-3′ (HDNA)[24], and the X-ray structure of the unmethylated oligo 5′- Tp CpCpCpGpTpGpApGpCpCpTpCpCpGpGpCpApGpGp-3′ (UMDNA), which reveals weaker binding[25]. The latter observation is in apparent contrast to the reported dissociation constant ($K_d$) of the DNMT1/ UMDNA complex in aqueous solution in the nM range[26], comparable to that of DNMT1/HDNA and other high-affinity DNA/protein complexes (Supplementary Table S1). Together with recent findings reporting DNMT1 expression and functions in postmitotic and even adult neurons, such as POA and MGE-derived cortical GABAergic interneurons[20,27–29], this proposes DNMT1 to have enzymatic functions in neurons other than the methylation of HDNA.

Visual inspection of the X-ray structure[25] suggests that the weak interactions between the UMDNA and the catalytic domain of DNMT1 are caused by additional intermolecular contacts of UMDNA with two surrounding DNMT1 molecules in the crystal lattice (Fig. 1a, Supplementary Fig. S1a–f). Consistently, in three independent 500-ns-long molecular dynamics (MD) simulations in solution, in conditions similar to those used for the $K_d$ measurements[26] (details are provided in Supplementary Tables S1–5 and the Supplementary methods), the two moieties undergo a significant rearrangement at the end of the dynamics. Almost all of the interactions of the UMDNA's 5′ region are formed with the catalytic domain, which performs the enzymatic reaction (Fig. 1b–g, Supplementary Fig. S1g, h). A CpG moiety interacts with Arg1234 and Asn1236 (Fig. 1c), as seen also in the cryo-EM structure of the HDNA/DNMT1 complex[24] (PDB ID: 7XI9, Supplementary Table S2). The number of contacts between the protein and UMDNA in solution, which lacks the additional DNMT1 molecules that are present in the crystal (Fig. 1a, b), increased by about 60% compared to the crystal structure (Fig. 1d–g, Supplementary Fig. S1g, h). Bioinformatic analysis showed that the solution MD structure aligns with other DNA/protein complexes with affinities in the nM range: the number of contacts for $A^2$ of contact surfaces is 0.17, which is comparable to other complexes ranging from 0.17 to 0.21 (Supplementary Table S1). Thus, DNMT1 binds more strongly to UMDNA in solution than in the crystal phase, likely due to the absence of crystal packing forces, consistent with $K_d$ measurements in solution[26].

Our analysis supports DNMT1's broader functional role beyond maintenance methylation, including potential DNA methylation-dependent regulation in postmitotic, non-replicating neurons, provided that the MD simulations reproduce the in vivo conditions.

### DNMT1 regulates key genes governing cIN development in embryonic SST[+] cells

To examine the role of DNMT1 in SST[+] cIN development, we generated a conditional knockout (KO) mouse model, in which *Dnmt1* deletion is induced by *Cre* expression in postmitotic SST[+] cells (*Sst-Cre/tdTomato/*

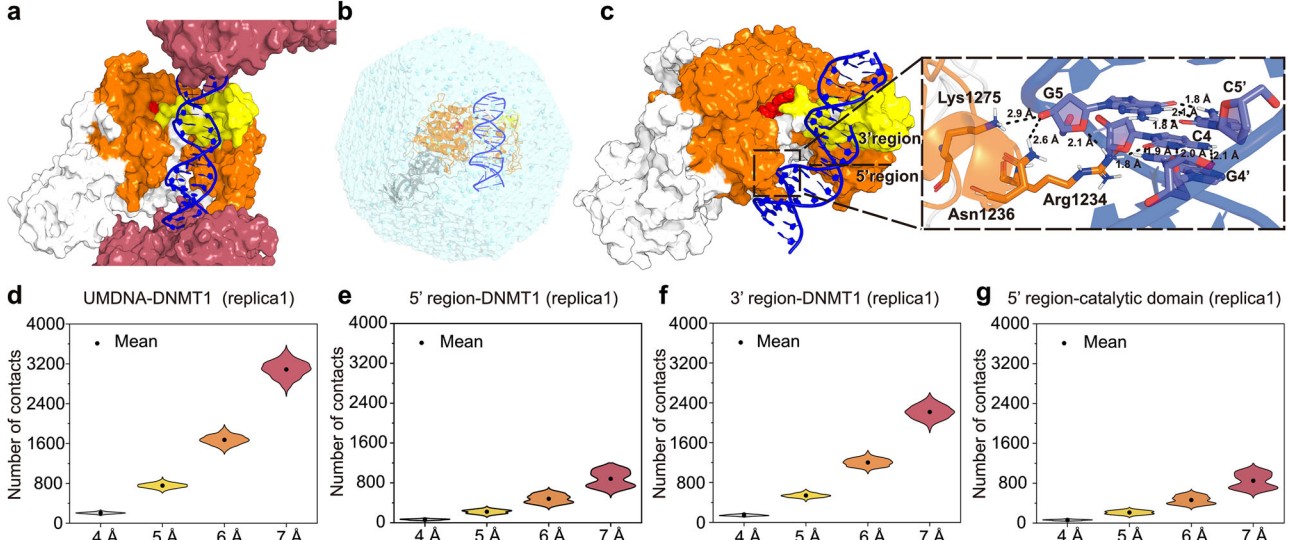

**Fig. 1 | Molecular dynamic simulations of DNMT1 binding to unmethylated DNA (UMDNA). a** DNMT1/UMDNA/SAH complex unit in the X-ray structure (PDB ID: 3PTA). Inter-unit DNMT1 protein regions (rose-pink) are mostly positively charged and interact with the UMDNA (blue). Intra-unit DNMT1 regions include the auto-inhibitory linker BAH1/BAH2 (white surface), catalytic domain (orange), SAH (red spheres), and CXXC domain (yellow). More information is presented in Supplementary Fig. S1b. **b–g** Simulations of DNMT1/UMDNA interactions in aqueous solution. **b** Illustration of the DNMT1/UMDNA complex in solution. **c** Structure of

the complex at the endpoint of one of three MD simulations (same coloring scheme as in (**a**)), with residues at the CpG site interacting with the catalytic domain highlighted by sticks. Only hydrogen atoms bound to polar groups are shown (as black dashed lines). **d–g** Number of contacts between DNMT1 and UMDNA (**d**), the 5′ (**e**), and 3′ regions (**f**), as well as between the catalytic domain of DNMT1 and the 5′ region (**g**) during the last-100-ns simulation (being consistent across replicas (Supplementary Fig. S1g, h). Raw data are available via the hyperlinks listed in the Data Availability Statement.

*Dnmt1 loxP²*. *Sst-Cre/tdTomato* mice served as controls (Supplementary Fig. S2a). The *Sst-Cre* deleter strain is a well-established model, with *Cre*-dependent *tdTomato* expression being evident as cells leave the MGE[30,31]. Validation of the mouse models is shown in Supplementary Fig. S2b–n.

We focused on embryonic day (E) 14.5, a critical stage characterized by extensive SST⁺ cIN migration through the basal telencephalon and the onset of tangential migration within the cerebral cortex[11]. To identify putative DNMT1 target genes, we conducted RNA sequencing and methyl-sequencing of FACS-enriched E14.5 *Sst-Cre/tdTomato/ Dnmt1 loxP²* (*Dnmt1* KO) and *Sst-Cre/tdTomato* (control) cells from the basal telencephalon. We revealed 1,160 differentially expressed genes (DEGs) and 5271 differentially methylated regions (DMRs) overlapping annotated genomic regions (Fig. 2a–c, Supplementary Data 1, 2). Correlating DMRs with the DEGs, we identified 488 genes exhibiting both transcriptional alterations and DNA methylation changes in *Dnmt1* KO cells (Fig. 2c, Supplementary Data 1–3). Consistent with previous findings[20,28], we observed up- or downregulated genes, as well as DMRs showing gain and loss in DNA methylation upon *Dnmt1* deletion. This aligns with findings from others, demonstrating that reduced *Dnmt1* and *Dnmt3a* expression leads to both gene activation and repression[32,33]. Downregulated genes are often interpreted as indirect effects, for instance, resulting from *Dnmt1* deletion-induced upregulation of a transcriptional repressor or adaptive transcriptional responses to physiological changes[28,33,34]. Given the partially redundant functions of DNMT1 and DNMT3A[35], compensatory DNMT3A activity may contribute to the increased DNA methylation levels observed after *Dnmt1* deletion (Supplementary Data 2).

To identify potential targets of DNMT1-dependent repressive DNA methylation, we focused on the 279 of the 488 genes, which were upregulated in *Dnmt1* KO samples and displayed DMRs (Fig. 2c). Notably, 101 of these 279 genes contained DNMT1 binding motifs that we identified through additional ChIP-sequencing experiments (Fig. 2c, d, Supplementary Data 4).

Gene ontology (GO) enrichment analysis of upregulated genes with altered DNA methylation in *Dnmt1* KO cells revealed an

overrepresentation of genes associated with nervous system development, neuron differentiation, cell-cell signaling, locomotion, neurogenesis, and cell differentiation (Fig. 2f, Supplementary Fig. S3a; Supplementary Data 5). A similar enrichment was observed for the entire set of significantly upregulated genes in *Dnmt1* KO cells, regardless of concurrent DNA methylation changes (Supplementary Fig. S3d, Supplementary Data 6). Notably, many genes upregulated in E14.5 *Dnmt1* KO samples were downregulated in embryonic stem cell (ESC)-derived *Dnmt1^tet/tet* neurons overexpressing DNMT1[36], highlighting DNMT1's repressive role in neurodevelopmental gene regulation (Fig. 2e, Supplementary Data 7, 8). Moreover, DNMT1-regulated genes in SST⁺ cINs exhibited significantly longer coding sequences, transcript lengths, genome spans, and extended 3′ and 5′ UTRs compared to the entire set of genes detected in SST⁺ cINs (Chi-squared test; Supplementary Fig. S3b, c, e, f, and S4b, c). This aligns with the observation that *Mecp2* deletion, a reader of DNA methylation[37], similarly leads to the upregulation of long genes expressed during neurodevelopment[38].

Among DEGs with increased expression and DMRs in *Dnmt1* KO cINs, we identified genes coding for key cortical interneuron differentiation and maturation regulators, including *Arx, Zeb2, Dlx2, Dlx5, Sox6, Maf, Tcf4, Satb1*, and *Lhx6*[11,39–41] (Fig. 2a, b). In addition to these well-characterized transcription factors governing cIN fate and migration, we identified genes impacting locomotion and cell-cell signaling (Fig. 2a, b, f). These included *Erbb4, Reln, Robo2*, and members of the Eph/ephrin family, such as *Efnb2*, all of which encode proteins critical for directional guidance and neuronal positioning[18,42,43]. Moreover, brain development and disease-relevant epigenetic regulators and chromatin modifiers like *Atrx, Setd2, Kmt2a, Ash1l*[44,45], as well as the lncRNAs *Dlx6os1*[46] and *Kcnq1ot1*[47] were increased in expression and changed in DNA methylation in *Dnmt1* KO cINs (Fig. 2a, b).

Among the upregulated genes, *Arx, Zeb2, Dlx2, Dlx5, Ash1l*, and *Kmt2a* exhibited DMRs characterized by reduced DNA methylation levels (Fig. 2b). In contrast, *Erbb4, Tcf4, Dlx6os1, Kcnq1ot1, Reln, Efnb2*, and *Atrx* display sites with both a loss and gain in DNA methylation (Fig. 2b, Supplementary Data 3). *Dnmt1* deletion in E14.5 SST⁺ cINs

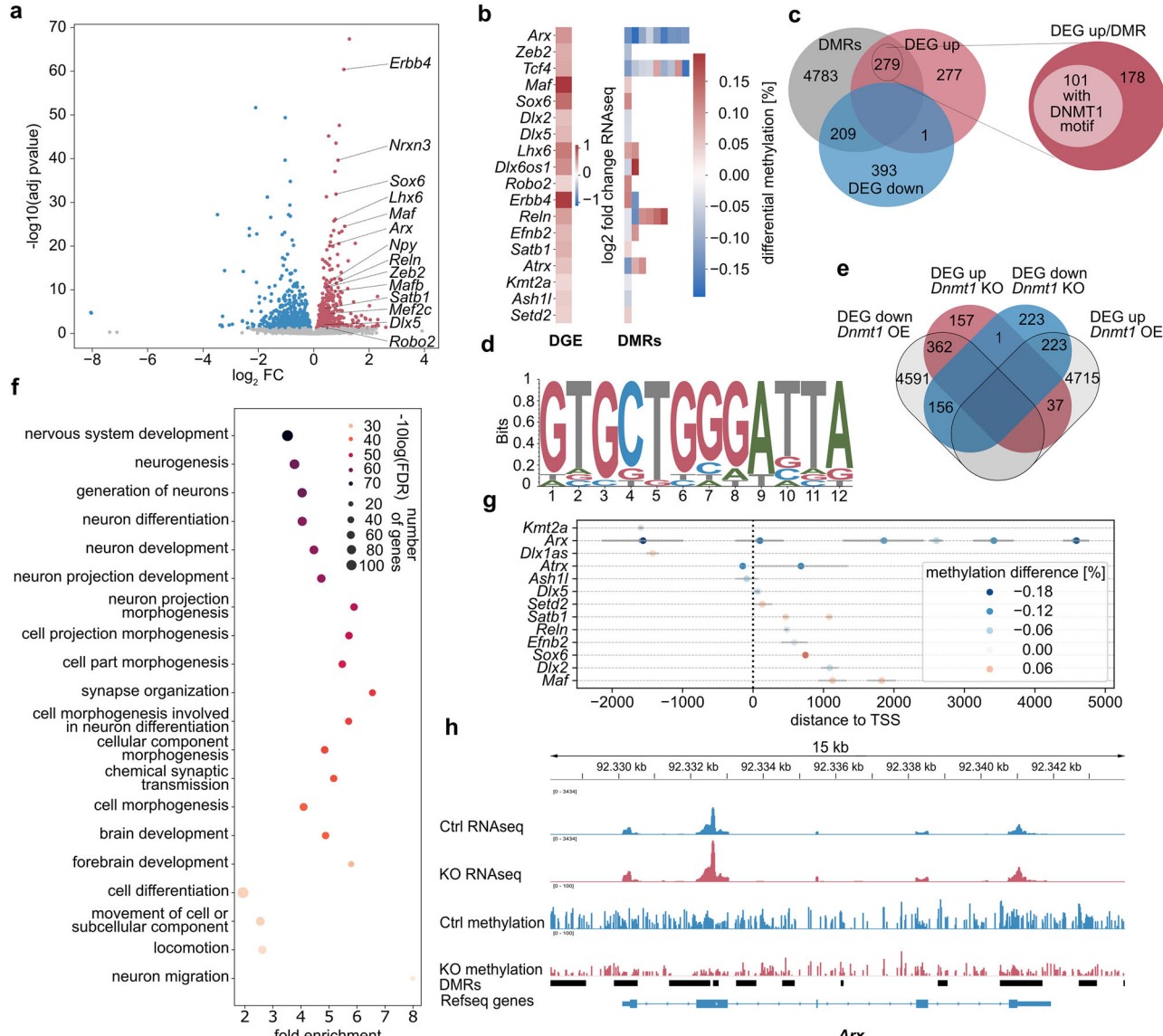

**Fig. 2 | DNMT1 regulates the expression of hub genes of cIN development in migrating E14.5 SST+ interneurons.** Differential gene expression and methylation analysis of FAC-sorted *Sst-Cre/tdTomato* control (Ctrl) and *Sst-Cre/tdTomato/Dnmt1 loxP²* (knockout; KO) cells isolated from the E14.5 basal telencephalon based on total RNA and enzymatic methyl-sequencing. For RNA- and methyl-sequencing, we separately processed two samples per genotype, each consisting of cells pooled from multiple embryos (RNA sequencing: Ctrl *N* = 23 embryos; KO *N* = 14 embryos; methyl-sequencing: Ctrl *N* = 15 embryos; KO *N* = 11 embryos). **a** Volcano plot depicting the differentially expressed genes (DEGs) between control and KO samples. Genes annotated as significantly changed (adjusted *p* value < 0.05) are colored blue and red, with red depicting increased expression in KO. Two-sided Wald test with Benjamini–Hochberg adjustment (DESeq2). **b** Heatmap of selected genes (coding for transcription factors and guidance cues governing cIN development, as well as epigenetic modifiers), color-coding differential gene expression (DGE) and differential methylated regions (DMRs). **c** Venn diagram illustrating the overlap between up- or downregulated genes and genes associated with or containing a differentially methylated region (DMR). The intersection of upregulated and differentially methylated genes after knockout was tested for enrichment of the DNMT1 binding motif identified by DNMT1-ChIP-sequencing, shown in **d**. **d** DNA motif enriched in DNMT1-interacting chromatin, detected by ChIP-sequencing (*N* = 2 biological replicates). **e** DEGs identified in *Dnmt1* KO samples were compared to DEGs determined in embryonic stem cell-derived murine neurons overexpressing DNMT1. **f** Selection of brain development-related gene ontology (GO) terms enriched in genes that were both upregulated and differentially methylated in E14.5 *Sst-Cre/tdTomato/Dnmt1* KO cells (*ShinyGO* 0.82). **g** Position of differentially methylated regions relative to the transcription start sites of genes related to cIN development. Horizontal bars indicate the region's size, and the color code represents the mean methylation change for the respective region. **h** RNA sequencing tracks combined with the methylation profile of the *Arx* gene locus obtained from Ctrl and KO samples. Further information: Supplementary Tables 1–8. Raw data are available via the hyperlinks listed in the Data Availability Statement.

resulted in DMRs near transcriptional start sites (TSS) and within intragenic regions, in addition to intergenic loci (Supplementary Data 2). Intragenic DNA methylation has diverse regulatory functions and is particularly implicated in neuronal differentiation and migration[48,49]. Consistently, many dysregulated genes in E14.5 *Dnmt1* KO samples with changes in DMRs across the gene body encode proteins essential for cIN development and migration, such as ARX[16],

ZEB2[50,51], TCF4[40] and ERBB4[18]. Given that promoter DNA methylation typically correlates with transcriptional repression[52], we further examined methylation changes at the TSS of the aforementioned genes dysregulated in *Dnmt1* KO SST+ cINs (Fig. 2g; Supplementary Data 7). Among the genes known to govern cIN development, *Arx*, *Dlx2*, *Dlx5*, and *Reln* displayed reduced DNA methylation at or near TSSs (Fig. 2g, h), aligning with their elevated expression in E14.5 *Dnmt1*

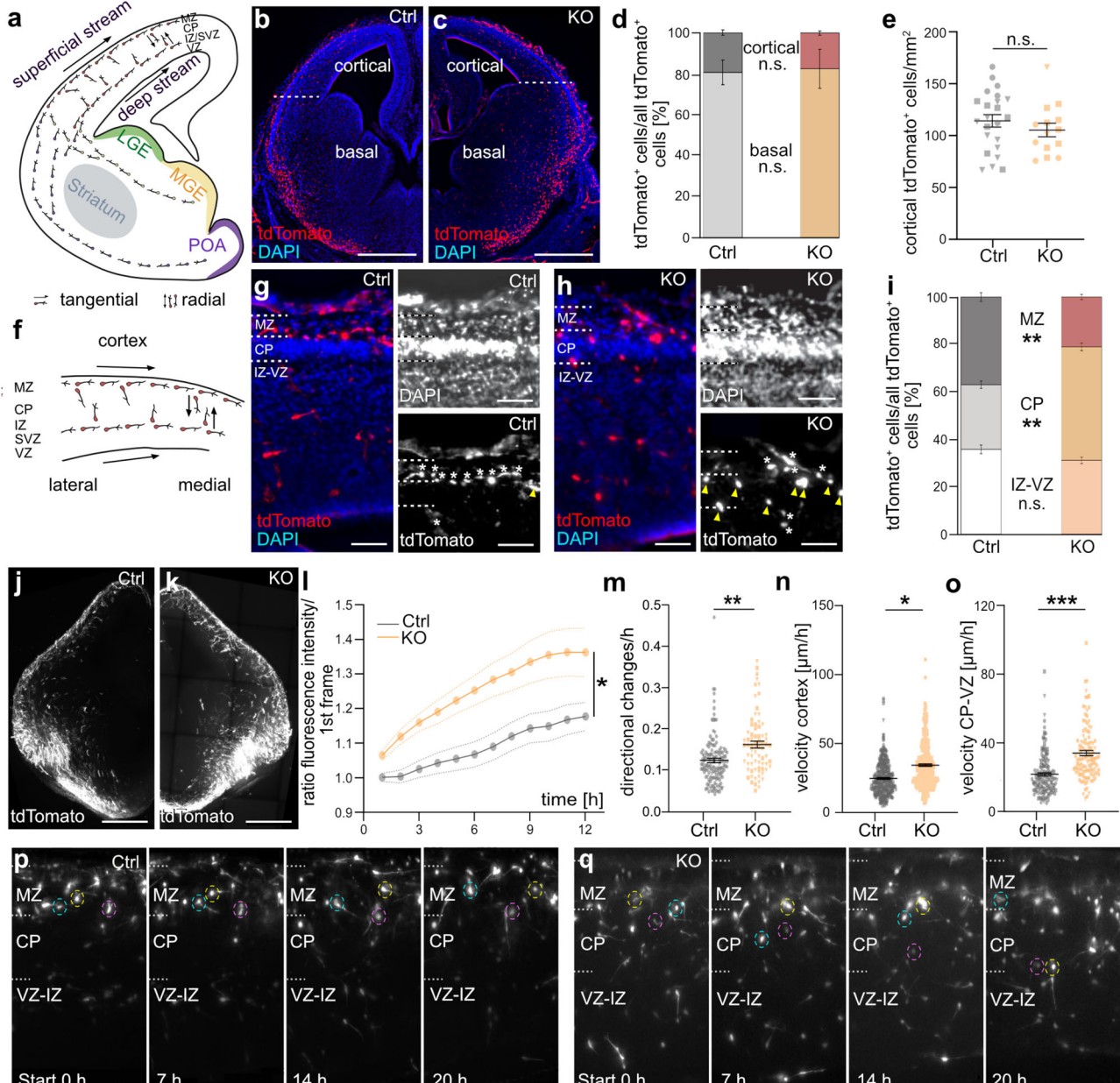

**Fig. 3 | DNMT1 regulates the migration of SST⁺ interneurons in the developing cortex. a** Scheme of cIN migration routes (E14.5, coronal section, hemisphere). **b**, **c** *Sst-Cre/tdTomato* (Ctrl; **b**) and *Sst-Cre/tdTomato/Dnmt1 loxP²* (KO; **c**) brain sections (50 μm). Scale bars: 500 μm. **d** TdTomato⁺ cell distribution in basal telencephalon vs. cortex (E14.5; Ctrl: *n* = 12 sections, KO: *n* = 8 sections, *N* = 3 embryos/genotype). **e** TdTomato⁺ cell count within the cortex normalized to the area (50 μm, Ctrl: *n* = 14 sections, KO: *n* = 8 sections, *N* = 3 embryos per genotype). **f** Migration trajectories within the E14.5 cortex. **g**, **h** Ctrl and KO cortices (E14.5); white asterisks: cells in MZ or IZ/SVZ; yellow arrows: cells in CP; scale bars: 100 μm. **i** Proportional distribution of tdTomato⁺ cells within the cortical zones normalized to the overall tdTomato⁺ cell count (E14.5, Ctrl: *n* = 14 sections, KO: *n* = 8 sections, *N* = 3 embryos per genotype). **j**, **k** Temporal z-projections of tdTomato⁺ cell migration over 20 h

(350 μm, Ctrl vs. KO). Scale bars: 500 μm. **l–o** Quantifications: **l** tdTomato⁺ fluorescence in CP (first 12 h; Ctrl: *n* = 4, KO: *n* = 6 slices, *N* = 3 embryos/genotype); **m** average directional changes (Ctrl: 134 cells/5 slices, *N* = 4 embryos; KO: 74 cells/3 slices, *N* = 3 embryos); **n** velocity of all migrating cells (Ctrl: 377 cells/6 slices, *N* = 5 embryos; KO: 323 cells/5 slices, *N* = 3 embryos); **o** velocity in CP–VZ (Ctrl: 193 cells/6 slices, *N* = 5 embryos; KO: 114 cells/3 slices, *N* = 3 embryos). **p**, **q** Migration of tdTomato⁺ cells (circled, 350 μm slices, Ctx; scale: 50 μm). *p < 0.05, **p < 0.01, ***p < 0.001; n.s.: not significant (further information: Supplementary Data 10). Ctrl control, CP cortical plate, VZ ventricular zone, SVZ subventricular zone, MZ marginal zone, IZ intermediate zone, Ctx cortex. Data points sharing the same symbol in (**e**) and (**m**–**o**) represent one embryo. Nested two-way ANOVA in (**d**), (**e**), (**i**), (**l**–**o**). Error bars: ±SEM.

KO cells (Fig. 2a, b). Indeed, *Arx* exhibited the most pronounced reduction in DNA methylation, with a high number of hypomethylated regions detected close to the TSS and across the gene body (Fig. 2b, h; Supplementary Data 2). Moreover, *Kmt2a, Atrx*, and *Ash1l* showed reduced methylation levels close to the TSSs (Fig. 2b, g; Supplementary Data 2), suggesting that DNMT1 may normally act to repress these epigenetic regulators. In support of this, *Atrx* and *Ash1l*, but also *Setd2*

and *Kcnq1ot1*, as well as the transcription factors *Sox6* and *Dlx2*, exhibit the DNMT1 binding motif (Fig. 2d; Supplementary Data 4). Their derepression in the *Dnmt1* KO cells could contribute to broader chromatin remodeling and transcriptional dysregulation.

In sum, our findings suggest that DNMT1 establishes proper gene regulatory networks essential for the correct migration and differentiation of cortical SST⁺ interneurons.

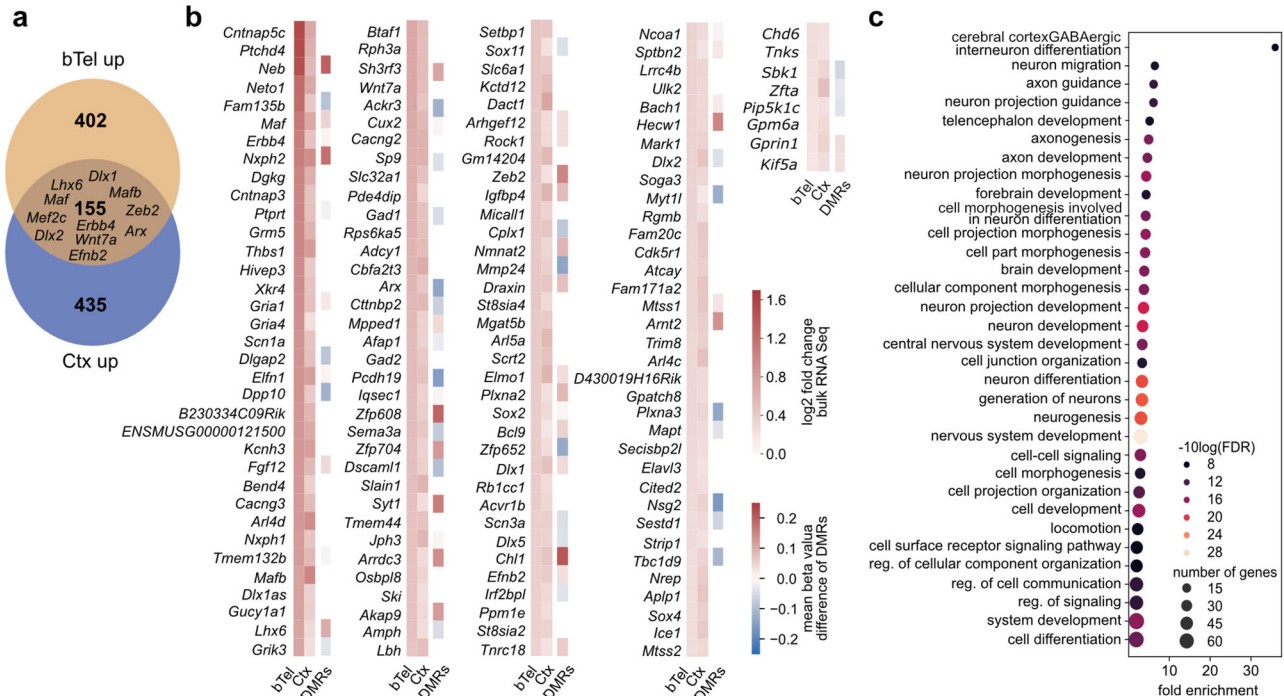

**Fig. 4 | Differential gene expression (DGE) analysis in SST⁺ cINs from the E14.5 cortex.** Differential gene expression (DGE) analysis of FAC-sorted *Sst-Cre/tdTomato* (Ctrl) and *Sst-Cre/tdTomato/Dnmt1 loxP²* (KO) cells isolated from the E14.5 cortex, compared to the DGE between Ctrl and KO samples from the basal telencephalon. For cortical analyses, three samples per genotype were separately processed, consisting of cells pooled from multiple embryos (Ctrl N = 7 embryos; KO N = 8 embryos). **a** Venn diagram illustrating the overlap between upregulated genes in KO cells from the basal telencephalon and the cortex (E14.5). **b** Heatmap illustrating the log2-fold change for the genes upregulated in KO cells from the basal telencephalon and the cortex. Moreover, changes in DNA methylation (from the basal telencephalon dataset) are depicted. **c** Gene ontology (GO) terms enriched in genes that were upregulated in E14.5 KO cells from both the basal telencephalon and the cortex. Further information: Supplementary Data 9. Raw data are available via the hyperlinks listed in the Data Availability Statement.

## *Dnmt1* deletion in postmitotic SST⁺ interneurons causes a premature exit from the superficial migratory stream in the embryonic cortex

To investigate the potential impact of DNMT1 on SST⁺ cIN migration, we next analyzed brain sections of *Sst-Cre/tdTomato/Dnmt1 loxP²* (*Dnmt1* KO) and *Sst-Cre/tdTomato* (control) embryos. We initially focused on E14.5, when the first SST⁺ interneurons have reached the cerebral cortex (Fig. 3a–c). Cortical dimensions were analyzed to confirm that both genotypes were at the same developmental stage (Supplementary Fig. S5a, b). In situ examination of *tdTomato*-expressing cells in E14.5 coronal brain sections revealed overall comparable numbers of SST⁺ cINs in the cortex and the basal telencephalon in both genotypes (Fig. 3d, e), indicating that the migration to the cortex was not impaired. However, within the cortex, we observed a reduced proportion of tdTomato cells migrating along the MZ and a significant increase of cells within the CP in *Dnmt1* KO brain sections (Fig. 3f–i). Live-cell imaging of organotypic brain slices at E14.5 for 20 h confirmed an elevated fraction of *Dnmt1* KO cells that deviated from the superficial migratory stream and entered the CP (Fig. 3j–q, Supplementary Movies 1 and 2). Notably, *Dnmt1* KO cells displayed a significantly increased frequency of directional changes compared to control *Sst-Cre/tdTomato* cells (Fig. 3m), suggesting a disruption in their ability to remain confined along the superficial migratory stream within the MZ. Moreover, their migratory pace was enhanced, which was particularly evident for radially migrating cells and cells migrating through the CP (Fig. 3n, o; Supplementary Fig. S6a). This aligns with an increased path length seen for the migrating *Dnmt1* KO cells in living brain slices (Supplementary Fig. S6b). Of note, in contrast to *Dnmt1*-deficient POA-derived *Hmx3*-expressing interneurons[20], we did not observe morphological abnormalities for migrating *Sst-Cre/tdTomato/Dnmt1 loxP²* cells (Supplementary Fig. S6c). Consistently, in vitro experiments

aiming to assess the morphology of *Dnmt1* siRNA-treated MGE cells (E14.5 + 1DIV) did not reveal detectable morphological differences compared to control conditions (Supplementary Fig. S6d–g).

Given the impaired migration within the cerebral cortex, we profiled the transcriptional changes underlying the premature exit of SST⁺ interneurons from the marginal zone and their increased invasion of the CP in E14.5 *Dnmt1* KO embryos. To this end, we conducted RNA sequencing on FACS-enriched *Sst-Cre/tdTomato* cells isolated from the E14.5 cortex of both genotypes (Fig. 4a–c, Supplementary Data 9). Notably, we obtained a prominent overlap with the upregulated genes identified in *Dnmt1* KO cells from the basal telencephalon, with master regulators of GABAergic interneuron differentiation and migration such as *Dlx1*, *Dlx2*, *Dlx5*, *Arx*, *Lhx6*, *Maf*, *Mafb*, *Mef2c*, *Satb1*, and *Zeb2* being altered in expression in both datasets (Fig. 4a–c). Of note, *Mef2c*, known to be regulated by *Maf* and *Mafb*, is a key factor in specifying PV⁺ cINs[53]. Similarly, *Dlx5* is implicated in PV⁺ cIN development[54].

Also, genes related to cell-cell signaling appeared significantly overrepresented among the commonly upregulated genes in *Dnmt1* KO cells from both compartments (Fig. 4c, Supplementary Data 9). This included *Erbb4* (Fig. 4a, b, Supplementary Fig. S6h, i), known to facilitate cIN invasion into the CP upon activation by its ligand Neuregulin 3 (NRG3)[18]. Its increased expression could thus mediate the premature exit of *Dnmt1* KO cells from the superficial migratory stream.

Additionally, *Efnb2* expression was persistently elevated in *Dnmt1* KO cells upon entering the cortex. Since EPHB1 is strongly expressed in the cortical MZ[12] and acts as a repulsive cue for migrating neurons from the basal telencephalon via reverse signaling[55], the increased *Efnb2* levels could further promote the premature exit from the MZ.

Together, these findings emphasize that DNMT1 is critical for regulating the proper development and migration of SST⁺ cINs within

the cortex at E14.5, acting through control of key genes involved in cIN migration and differentiation.

Although *Dnmt1*-deficient SST[+] interneurons prematurely invaded the CP by E14.5 within 20 h (Fig. 3j–l), we did not observe significant differences at later developmental stages (E16.5 and E18.5). This was neither the case for the proportion of cells that reached the cortex nor for the absolute number, density, or distribution of *Dnmt1* KO cells within the cortical regions quantified (Supplementary Fig. S7). The cortical dimensions were likewise similar between knockout and control embryos at E16.5 and E18.5 (Supplementary Fig. S5c–f), confirming that the embryos analyzed were of comparable developmental stages. This suggests that the impact of *Dnmt1* deletion on the directional migration of SST[+] cINs within the MZ may be transient.

## DNMT1-dependent regulation of SST[+] interneuron migration affects cortical progenitors non-cell-autonomously

The development of cINs and cortical excitatory neurons is intricately linked[8,56], and supernumerary cINs in the IZ elicit altered IPC numbers impacting the generation of upper-layer excitatory neurons[47]. Observing an impaired migration pattern of SST[+] cINs prematurely invading the CP and proliferative zones prompted us to investigate its influence on excitatory cortical progenitors and neurogenesis. Moreover, we found changed expression of genes involved in cell-cell signaling, known to modulate cortical neurogenesis, such as *Efnb2*[57], in *Dnmt1* KO cells. EFNB2 binds to EPHA4[58], which is expressed in RGCs[4]. Activation of EPHA4 receptors by EFNA5, imported by invading thalamic afferents, has been shown to regulate RGC division, impacting IPC generation and cortical layer formation[4].

To investigate whether the altered migration pattern and/or expression of signaling molecules in *Dnmt1* KO embryos affects the generation of cortical excitatory neurons, we performed immunostaining for TBR1, labeling early-born neurons of the preplate and layer VI[59,60]. We found a significant increase in TBR1[+] cell numbers within the CP in E14.5 *Dnmt1* KO brains (Fig. 5a–c), which became even more evident at E16.5 (Fig. 5f–h). At E18.5, the radial extension of both layer VI (TBR1[+] cells) and the remaining CP was significantly expanded in the conditional *Dnmt1* KO embryos compared to age-matched controls (Fig. 5k–m, p, q). We next investigated potential changes in EOMES[+] IPCs in *Dnmt1* KO embryos, which represent an intermediate progenitor stage from RGCs to neurons, proposed to give rise to most cortical projection neurons of all cortical layers[61,62], and had already been reported to be impacted by cINs[8]. While no gross changes in the SOX2[+] RGC pool were apparent (Supplementary Fig. S5g–i), we observed an increase in EOMES[+] IPCs at E14.5 (Fig. 5d, e) and E16.5 (Fig. 5i, j) in *Dnmt1*-deficient embryos, suggesting an enhanced IPC generation from RGCs. This coincides with the timing of the premature exit of *Dnmt1*-deficient SST[+] interneurons from the MZ at E14.5 and the elevated numbers of TBR1[+] neurons. As no differences in the pool of IPCs were detected at E18.5 (Fig. 5n, o), the deletion of *Dnmt1* therefore seems to elicit a temporally restricted non-cell autonomous effect on cortical IPCs and the production of deep-layer neurons.

## DNMT1 function in SST[+] interneurons modulates cortical progenitor and neurogenesis

The non-cell-autonomous effects of *Dnmt1* deletion in SST[+] cINs on cortical progenitors and the increased generation of deep-layer neurons suggest that SST[+] interneurons provide signalling cues influencing progenitor dynamics, similar to thalamic afferents importing EFNA5[4]. Notably, altered expression of signaling molecules known to modulate cortical neurogenesis and progenitor division, including *Wnt7a* and *Efnb2* (Fig. 4a–c, Supplementary Data 1 and 9) was detected in *Dnmt1* KO cells.

To further investigate potential cIN-progenitor interactions, we employed *CellChat* analysis of scRNA-sequencing data from E14.5 dorsal telencephalons, predicting ligand–receptor interactions between SST[+] cINs and cortical progenitor types: RGCs and IPCs (Fig. 6a, b, Supplementary Fig. S8a–c).

NRXN3-NLGN1 signaling had the highest probability for interactions between SST[+] cINs with both RGCs and IPCs. However, *Nrxn3* was not detected among the upregulated genes in *Sst-Cre/tdTomato/Dnmt1 loxP²* cINs at E14.5 (Supplementary Data 9). Moreover, EFNB2-EPHA4 signaling was revealed as a potential mediator of these interactions, with *Efnb2* expressing SST[+] MGE-interneurons interacting with RGCs as well as IPCs at E14.5 (Fig. 6a, b). ScRNA-sequencing data from the E16.5 dorsal telencephalon retrieved similar results (Supplementary Fig. S8d, e). Supporting this, *Efnb2* was upregulated in *Dnmt1*-deficient SST[+] cINs and altered in *Dnmt1*-overexpressing neurons (Fig. 2a, Fig. 4a–c; Supplementary Data 1, 7, 9). Given its known role in promoting neurogenesis at mid-corticogenesis[57] and its reported binding to EPHA4[63], which is expressed in RGCs[4], we hypothesized that increased *Efnb2* expression in *Dnmt1*-deficient SST[+] cINs contributes to enhanced RGC differentiation into IPCs and deep-layer neurons. Pair-cell assays confirmed that ephrinB2-Fc stimulation promotes neurogenic RGC divisions and IPC generation (Fig. 6c–g).

To examine altered cIN-progenitor interactions in *Dnmt1* KO embryos, we performed single-nucleus (sn) RNA-seq on E16.5 cortical cells from both genotypes (Fig. 6h–m). This approach allowed sample freezing for simultaneous processing and captured potential differences in cIN-progenitor interactions but also non-cell-autonomous effects on postmitotic neuron generation, which might become more pronounced at this developmental stage.

Aligning with the histological data, *Dnmt1* KO samples at E16.5 showed an increased proportion of deep-layer cortical neurons and a reduced fraction of upper-layer neurons (Fig. 6h–m; Supplementary Fig. S8f, h, i). Additionally, *Erbb4*, *Arx*, and *Maf*—key regulators of cIN development—along with numerous other genes previously identified as upregulated through bulk RNA sequencing of E14.5 cortical *Dnmt1* KO cells, displayed a significantly elevated expression in *Sst*[+]/*Gad2*[+] cells from the E16.5 *Dnmt1* KO samples (Supplementary Fig. S8g). Applying *CellChat* analysis, genotype-related differences in interactions between MGE interneurons (senders) and RGCs (receivers) were revealed, while this was not evident for the interaction between MGE interneurons and IPCs (numbers and interaction strength; Fig. 6l, m). This supports the instructive role of cINs on RGCs, and that this influence was increased in *Dnmt1* KO embryos. We further conducted MERFISH analysis in E16.5 brain sections of *Sst-Cre/tdTomato/Dnmt1 loxP²* and *Sst-Cre/tdTomato* embryos. Here, we also confirmed the elevated proportion of deep-layer neurons in *Dnmt1* KO brains and an increase in IPCs (Fig. 6n–q; Supplementary Fig. S9).

In summary, DNMT1 is essential for the proper migration of SST[+] cINs; its loss disrupts migratory trajectories, alters signaling dynamics, and promotes RGC differentiation into IPCs and deep-layer neurons.

## *Dnmt1* deletion-induced changes at embryonic stages manifest in an altered cortical architecture in adult mice

Finally, we investigated whether the *Dnmt1* deficiency-related embryonic defects result in long-lasting structural changes in the adult cortex. We conducted a comparative analysis of the distribution, density, and fate of *Sst-Cre/tdTomato* control and *Dnmt1* KO interneurons in cortices of adult mice. Immunohistochemistry confirmed SST expression in *Sst-Cre/tdTomato* cells, as well as the reported co-expression of NPY and CALB2 (calretinin)[64] in both genotypes (Supplementary Fig. S10a–d). However, the proportion of SST[+] cINs co-expressing PV[65] was significantly increased in *Dnmt1* KO mice compared to controls (Supplementary Fig. S10e–i). This is in line with the elevated transcription of *Dlx5* and *Mef2c* found in E14.5 *Dnmt1* KO cells (Figs. 2a, b and 4a, b), both coding for drivers of PV[+] cINs development[54,66,67].

*Sst-Cre/tdTomato* cIN numbers were comparable across all layers between genotypes (Fig. 7a–c), but cell density in the superficial layers was significantly increased in *Dnmt1* KO mice (Fig. 7c). This correlated

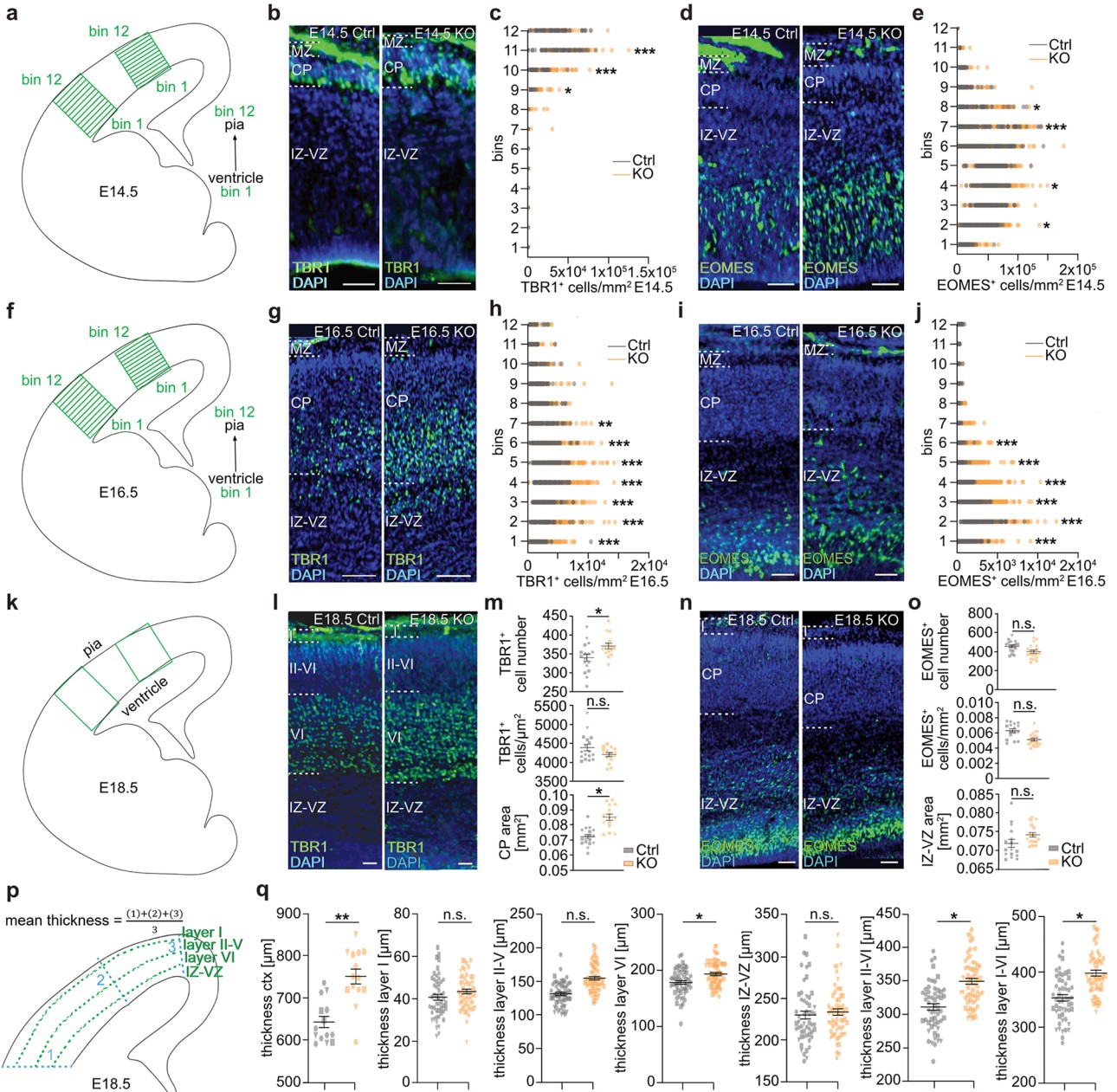

**Fig. 5 | Conditional deletion of *Dnmt1* in SST+ interneurons affects IPCs and the generation of TBR1+ cells. a, f** Schemes of lateral and dorsal bin positions in the cortex of coronal brain sections (E14.5 and E16.5), used for quantifications. **b–e** Immunostaining and quantification of TBR1+ neurons (**b, c**) and EOMES+ IPCs (**d, e**) in E14.5 50 μm coronal sections of *Sst-Cre/tdTomato* control (Ctrl) and *Sst-Cre/tdTomato/Dnmt1 loxP2* (KO) embryos (DAPI: blue). Two-way ANOVA with Bonferroni correction (*n* = 13 control, *n* = 9 KO sections; *N* = 3 embryos/genotype). **g–j** Immunostaining and quantification of TBR1+ neurons (**g, h**) and EOMES+ IPCs (**i, j**) (E16.5; 50 μm coronal sections). Two-way ANOVA with subsequent Bonferroni correction, *n* = 8 sections for control and KO from *N* = 3 embryos for both genotypes. **k** Schematic illustration depicting the lateral and dorsal cortical localizations in coronal sections (E18.5) for density and distribution quantification of EOMES+ and TBR1+ cells in an embryonic brain hemisphere. **l** TBR1 and **n** EOMES immunostaining; (DAPI: blue). **m, o** Quantitative analyses of the total numbers and densities of TBR1+ postmitotic neurons (**m**) and EOMES+ IPCs (**o**). The area of the CP and of the IZ-VZ region was analyzed in (**m**) and (**o**), respectively (nested two-way ANOVA with *n* = 11 sections for both genotypes from *N* = 4 embryos). **p** Scheme of the cortex (coronal view, E18.5), and parameters quantified in (**q**). **q** Quantification of the mean thickness of the E18.5 cortex, its layers, and transient zones (VZ-IZ). Three regions (lateral (1), dorsolateral (2), and dorsal (3)) were averaged to obtain the mean thickness, respectively, for each layer (nested two-way ANOVA with *n* = 12 control and *n* = 9 KO sections, *N* = 4 brains for both genotypes). Error bars: SEM *p < 0.05, **p < 0.01, ***p < 0.001 (detailed information: Supplementary Data 10). MZ marginal zone, CP cortical plate, IZ–VZ intermediate zone-ventricular zone, Ctrl control, n.s. not significant. Data points in (**m, o, q**) sharing same symbol represent one embryo. Scale bars: 50 μm in (**b, d, g, i, l, n**).

with a reduced radial extension of CUX1-positive layers II–IV and an expansion of deep layers V–VI, while overall cortical thickness remained unchanged (Fig. 7a, b, d).

The shift in layer proportions aligns with the increased number of TBR1+ neurons and IPCs at E14.5 and E16.5, as well as snRNA-seq and MERFISH findings (Figs. 5 and 6). These results indicate that *Dnmt1* deletion in SST+ interneurons non-cell autonomously alters cortical progenitor dynamics, impacting adult cortical architecture and SST+ cIN densities in superficial layers.

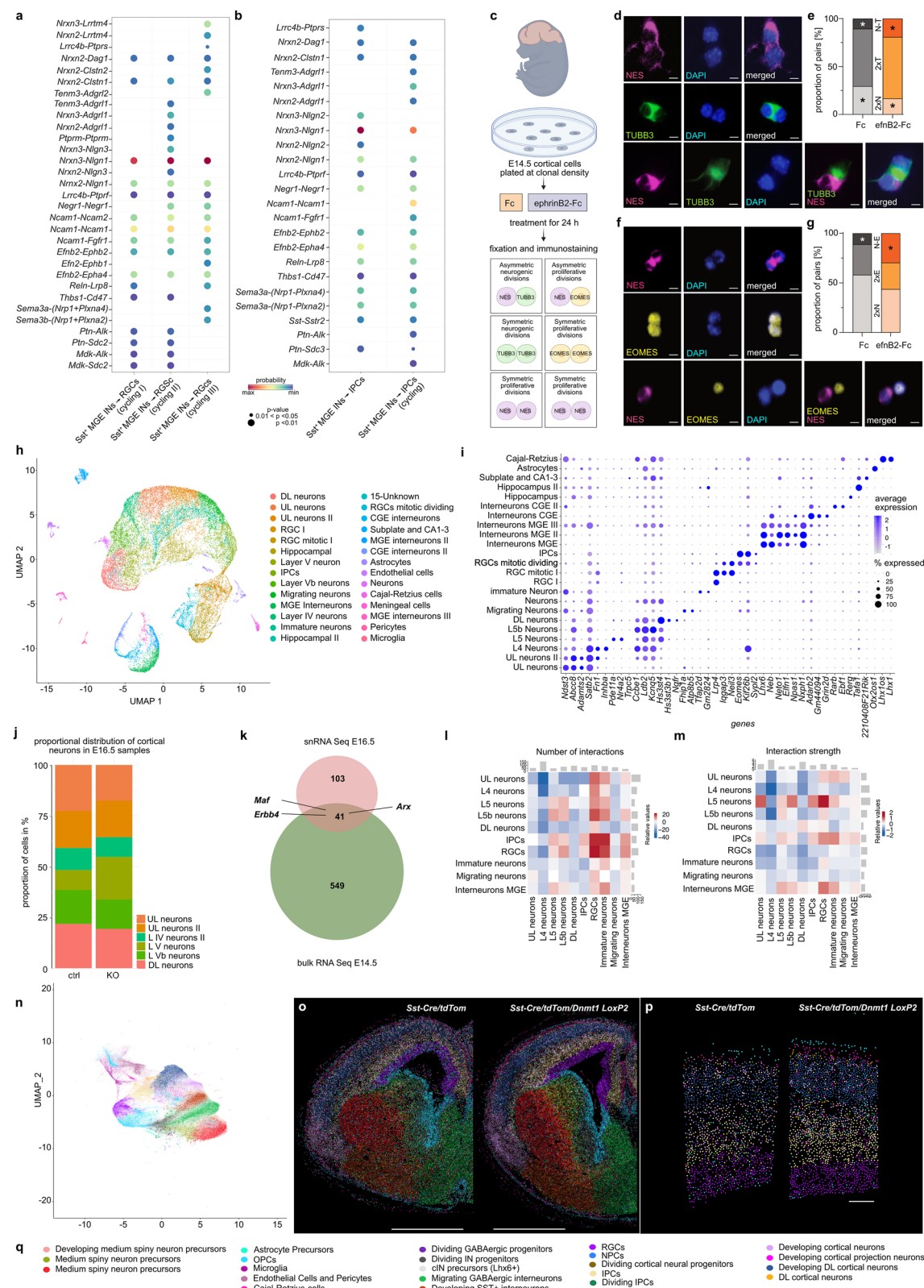

### Sst-Cre/tdTomato/Dnmt1 loxP² mice display functional abnormalities

To determine whether the observed structural alterations in the adult cortex also translate into functional changes, we conducted electrophysiological recordings in the primary somatosensory barrel cortex (S1) of awake head-fixed *Dnmt1* KO and control mice on a running wheel (Fig. 7e left, Supplementary Fig. S11). To simultaneously capture population and single-cell neural activity across the entire cortical depth, we recorded with high-density Neuropixels probes (Fig. 7e right; Supplementary Fig. S11a, b) while mice were passively stimulated with short air puffs to their whisker pad[68]. Tactile stimulation induced a clear transient response in average local field potentials (LFPs) in S1 of control mice, whereas LFP responses in conditional *Dnmt1* KO mice were much weaker and temporally delayed (Supplementary Fig. S11c).

**Fig. 6 | Non-cell-autonomous effects of SST⁺ interneurons on cortical progenitors. a, b** Ligand-receptor interactions between SST⁺ cINs to (**a**) RGC and (**b**) IPCs, extracted from differentially expressed genes (DEG; scRNA sequencing of E14.5 dorsal telencephalons ($N = 5$)). Dot color: communication probability; dot size: one-sided permutation test p-values. **c–g** Pair cell assay using E14.5 cortical neurons treated for 24 h with control-Fc or ephrinB2-Fc, followed by immunostaining for nestin (NES, magenta), β-III-tubulin (TUBB3, green), EOMES (yellow), and DAPI (blue). **c** Experimental design. Representative images are shown in (**d, f**). Scale bars: 5 μm. **e, g** Quantification of NES/NES (2xN), TUBB3/TUBB3 (2xT), NES/TUBB3 (N-T), EOMES-EOMES (2xE), and NES-EOMES (N-E) cell pairs, normalized to the total pair count. Unpaired two-tailed Welch's t-test with control-Fc: $n = 90$ cell pairs, efnB2-Fc: $n = 91$ cell pairs, from $N = 4$ experiments in (**e**); and control-Fc: $n = 90$ cell pairs, efnB2-Fc: $n = 94$ cell pairs, from $N = 4$ experiments in (**g**). **h–m** snRNA sequencing of cortical cells from E16.5 *Sst-Cre/tdTomato* (Ctrl) and *Sst-Cre/*

*tdTomato/Dnmt1 loxP²* (KO) embryos ($n = 2$ brains from $N = 2$ mice per genotype). **h** UMAP of identified cell clusters. **i** Dot plot of marker gene expression across clusters. **j** Bar plot: proportions of postmitotic excitatory neurons. **k** Venn diagram: overlap of upregulated genes in E14.5 FAC-sorted cortical KO cells (bulk RNA sequencing) and E16.5 cortical *Sst⁺/Gad⁺* KO cells (snRNA-seq). **l, m** *CellChat*-derived interaction maps (E16.5 cortical snRNA-seq data): number (**l**) and strength (**m**) of inferred cell–cell interactions in KO vs. Ctrl. **n–q** MERFISH-based spatial transcriptomics (E16.5; $n = 1$ section for KO and Ctrl, respectively). **n** UMAP of identified cell clusters by gene expression. **o** Spatial mapping of clusters onto anatomical regions in E16.5 sections, scale bar: 1000 μm. **p** Magnified sections from the cortex depicted in (**o**). Scale bar: 100 μm. **q** Cluster annotation for (**n–p**). *$p < 0.05$, **$p < 0.01$, ***$p < 0.001$, error bars: ±SEM. n.s.: not significant. Raw data are available via the hyperlinks listed in the data availability statement, Supplementary Data 10.

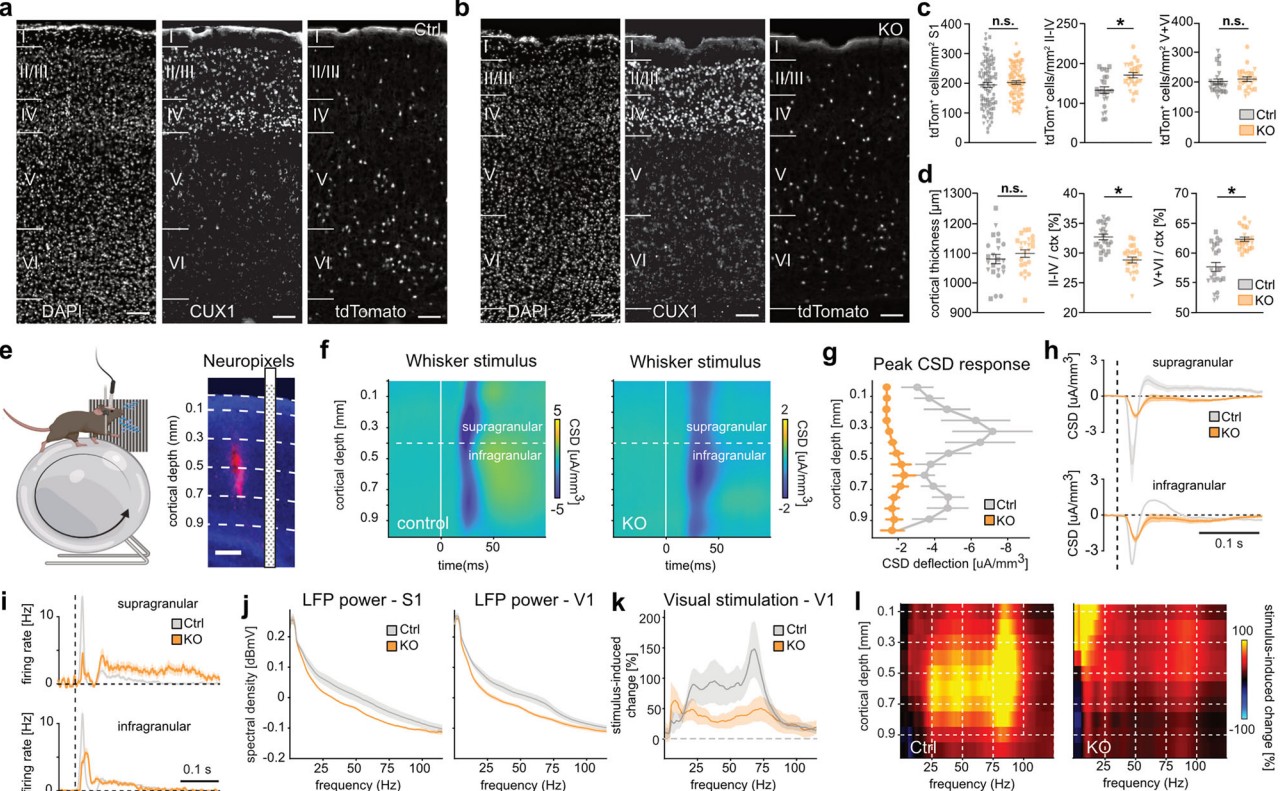

**Fig. 7 | Aberrant cortical architecture and functionality in *Dnmt1*-deficient mice. a, b** CUX1 immunostainings with DAPI (six-month-old male *Sst-Cre/tdTomato/Dnmt1 loxP²* (KO) and *Sst-Cre/tdTomato* mice (Ctrl), respectively). S1 sagittal slices from Bregma 1.32 and 1.44. Scale bars: 100 μm. **c, d** tdTomato⁺ cell density normalized to cortical area (**c**), cortical thickness, and proportional thickness of deep and upper layers (**d**; nested two-way ANOVA; Ctrl: $n = 24$ slices, KO: $n = 23$ for, $N = 3$ brains per genotype). *$p < 0.05$ (detailed information: Supplementary Table 10). Error bars: ±SEM from the mean. Data points with same symbols depict the same mouse. **e** S1 Neuropixels recordings in head-fixed mice on a wheel. Example brain slice: probe fluorescence and schematic position. Scale bar: 200 μm. **f** Average CSD across depth, following a brief tactile stimulus (white line). Blue and yellow colors indicate current sinks and sources. **g** Significant lower and delayed (above 400 μm) CSD response peaks across sessions in *Dnmt1* KO mice (peak amplitude$_{control}$ = −4.74 ± 1.22 μA/mm³,

peak amplitude$_{KO}$ = −1.41 ± 0.18 μA/mm³, $p = 0.0037$; peak time$_{control}$ = 27.20 ± 0.84 ms, peak time$_{KO}$ = 31.80 ± 1.03 ms, $p = 0.0041$; $n = 8$ sessions). **h** Average CSD traces across tactile responses in supragranular (top) and infragranular layers (bottom). **i** Peristimulus time histogram for the spiking response to tactile stimulation in supra- or infragranular S1 layers. **j** LFP spectral power density (PSD) across depth in S1 (left), V1 (right; S1: gamma power$_{control}$ = −23.69 ± 21.32 dBmV, gamma power$_{KO}$ = −62.22 ± 3.16 dBmV, $p = 0.0201$; V1: gamma power$_{control}$ = −25.65 ± 8.65 dBmV, gamma power$_{KO}$ = −66.70 ± 4.72 dBmV, $p = 0.0069$; $n = 8$ sessions). **k** Stimulus-induced PSD change in V1, relative to baseline (gamma$_{control}$ = 104.25 ± 24.42%, gamma$_{KO}$ = 27.75 ± 7.20%, $p = 0.0069$; $n = 8$ sessions). **l** Stimulus-induced PSD change across depth. **g, j, k:** Wilcoxon rank-sum test, mean ± SEM. Shading in (**h–k**) shows the ±SEM. 2 mice per genotype in (**g–k**). Raw data available via hyperlinks in the Data Availability Statement.

Both negative and positive response components were weaker in KO mice, suggesting a disruption in excitatory and inhibitory circuit functions. To dissect the spatiotemporal structure of neural responses across all cortical layers, we employed a current source density (CSD) analysis based on our LFP recordings[69]. Consistent with the structural alterations in *Dnmt1* KO cortices, we found clear differences in responses across layers between the genotypes. As expected from the

literature[70], the earliest responses for control mice (Fig. 7f left) occurred in the granular layer (~400 μm cortical depth), then spreading to layer II/III (100–300 μm) and layer V (500–900 μm), which indicates intact spatiotemporal processing of sensory signals. In contrast, the CSD profile of *Dnmt1* KO mice was severely disrupted, with tactile responses being largely confined to the deeper infragranular layers and only weak responses in the superficial layers (Fig. 7f

right). Quantification of CSD peak responses across the entire depth confirmed that the most pronounced differences between *Dnmt1* KO and control mice occurred in the upper cortical layers (Fig. 7g, h). Notably, while control mice exhibited a positive rebound following the initial negative CSD deflection, the responses in *Dnmt1* KO mice lacked this feature, possibly due to disrupted feed-forward inhibition (Fig. 7h).

To quantify the tactile responses of cortical S1 neurons, we also used spike-sorting to isolate the spiking activity of sorted clusters. Consistent with our earlier results, the average spiking activity of S1 neurons showed that tactile responses were much weaker in *Dnmt1* KO mice, especially in the superficial layers (Fig. 7i, Supplementary Fig. S11d). Moreover, spiking responses of upper-layer neurons were much longer-lasting compared to controls, with neural activity extending up to 400 ms after stimulation. This suggests reduced temporal precision in sensory processing in *Dnmt1* KO mice, potentially due to impaired feedback inhibition.

We also investigated whether functional alterations manifest at the level of cortical network oscillations. Gamma (γ) oscillations from 30–120 Hz, which strongly rely on the accurate function of SST[+] cINs[71,72], are essential for integrating neural networks within and across brain structures during cognitive processes, and abnormalities are a feature of cognitive diseases such as schizophrenia, Alzheimer's disease, and Fragile X syndrome[73]. Thus, changes in γ-oscillations represent a useful marker of function and dysfunction in cortical circuit operations[74]. Indeed, the high-frequency LFP power in S1 was significantly reduced in *Dnmt1* KO versus control animals (Fig. 7j left).

Similar disruptions in oscillatory network dynamics were observed in the primary visual cortex (V1; Fig. 7j right). Visual stimulation (5-s-long visual grating stimuli) reliably induced γ-oscillations in V1, with control mice showing a clear 70 Hz peak, consistent with stimulus-induced γ-activity (Fig. 7k, l). This effect was markedly reduced in *Dnmt1* KO mice, which instead exhibited increased low-frequency activity in superficial cortical layers (Fig. 7k, l). These oscillatory changes suggest that *Dnmt1* deletion disrupts neural networks, likely due to structural alterations in cortical architecture, particularly in superficial layers.

To distinguish whether these functional deficits stem from altered cortical structure or impaired SST[+] cIN function, we optogenetically activated SST[+] cINs and assessed their spontaneous firing, action potential waveform, and inhibitory capacity (Supplementary Fig. S11e–h). No significant differences emerged between control and *Dnmt1* KO neurons, indicating that SST[+] cIN functionality remained intact. Thus, the observed cortical network disruptions likely result from structural changes, such as superficial layer thinning and increased SST[+] cIN density, rather than a generalized impairment of SST[+] cIN function.

### *Sst-Cre/tdTomato/Dnmt1* KO mice display behavioral abnormalities

To investigate whether functional impairments in *Dnmt1* KO mice translated into aberrant behavior, we tested the animals' sensory perception. Therefore, we employed a visuotactile evidence accumulation task and trained head-fixed *Dnmt1* KO and control mice to detect tactile and visual stimuli. Both genotypes learned the stimulus detection at similar rates, performed both uni- and multisensory task conditions with equal behavioral performances, and showed multisensory enhancement (Supplementary Fig. S12a–d), indicating intact sensory perception in *Dnmt1*-deficient mice. Moreover, we tested learning and memory capabilities using the Morris water maze, which also showed no differences between genotypes (Supplementary Fig. S12e–g).

Notably, we observed abnormal eye movements in head-fixed *Dnmt1* KO mice during the tests on the running wheel, characterized by brief episodes of significantly dilated pupils (more than double their normal size) lasting 30–60 s, accompanied by facial spasms or salivation (Fig. 8a). These episodes of pupil dilation correlated with a marked

increase in low-frequency activity (1–4 Hz), indicating epileptiform activity in KO mice (Fig. 8b). Further analysis of the propagation of this activity across cortical depth revealed that these events were particularly prominent in the superficial layers (Fig. 8c). In line with this, we detected prolonged seizure durations for *Dnmt1* KO mice in response to administration of PTZ (pentylenetetrazol), which is widely utilized to examine epileptic events in animal models and to assess changes in seizure responses[75] (Fig. 8d–f, Supplementary Movies 3-9). The latencies to the first occurrence of various epileptic events were comparable between genotypes (Supplementary Fig. S12h, i). However, *Dnmt1*-deficient mice exhibited higher Racine scale scores[76], indicating increased seizure severity (Fig. 8d–f). This aligns with studies reporting that alterations in SST[+] cIN activity and function can lead to epileptic events[77], and γ-oscillations have also been proposed to play a role in seizure generation[78].

Moreover, we observed frequent repetitive motor behavior of *Dnmt1* KO mice in their home cages (Supplementary Movie 10). Such behavioral abnormalities are often linked to neurological and neuropsychiatric diseases[79]. Notably, seizures frequently co-occur with comorbid symptoms, including autism-like stereotypies (e.g., repetitive motor behavior) and mood disorder-related phenotypes[80]. Thus, we next assessed nest-building performance, as this test assessed behavior associated with neuropsychiatric disorders[81]. In line with our functional results, we found that *Dnmt1* KO mice showed a reduced performance in nest-building and diminished interest in the provided nest-building material (Fig. 8g–j).

In summary, our cellular, functional, and behavioral results demonstrate that *Dnmt1* deletion in postmitotic SST[+] interneurons leads to altered cortical architecture and network function in adult mice, manifesting in disease-related behavioral abnormalities. This strongly suggests that DNMT1 plays a crucial role in the development of SST[+] interneurons and has non-cell autonomous functions essential for the proper formation of the cerebral cortex.

## Discussion

Our data propose that DNMT1 is critical for cortical development by (i) directly regulating the dispersion of SST[+] interneurons along the superficial migratory stream within the marginal zone, and (ii) facilitating their influence on radial glial cells, thereby contributing to proper cortical layering. Conditional *Dnmt1* deletion in SST[+] interneurons caused a premature exit from the MZ at E14.5, accompanied by a rise in EOMES[+] IPC numbers at E14.5, and more TBR1[+] postmitotic neurons at E14.5 and E16.5, likely arising from an altered division mode of RGCs. This resulted in an increased generation of deep-layer neurons and expanded deep layers at adult stages, while upper layers were reduced in radial extension. These changes in cortical architecture, together with altered densities of SST[+] interneurons in adult mice, were associated with prominent functional and behavioral deficits, demonstrating the importance of DNMT1 for establishing accurate network function in the adult cortex.

While DNMT1 is well recognized as a maintenance methyltransferase, primarily responsible for preserving DNA methylation during replication, emerging evidence—including our findings—suggests a broader role for DNMT1 in postmitotic neurons[27,28,35]. DNMT1 is expressed in non-dividing POA- and MGE-derived cINs, where it engages in both canonical DNA methylation and non-canonical regulatory mechanisms modulating distinct aspects of neuronal development[20,28]. Structural studies in the crystal phase suggest only a weak binding of DNMT1's catalytic domain to UMDNA[25]. Solution-based affinity measurements challenge this hypothesis, demonstrating a high-affinity interaction for both HDNA/DNMT1 and UMDNA/DNMT1 complexes[26], that is comparable in scale to other DNA-protein complexes. Our MD simulations resolve this contradiction by showing that UMDNA undergoes a significant conformational rearrangement in solution, increasing the interactions of its CpG sites with DNMT1's catalytic domain, as observed for the HDNA/DNMT1 complex[24]. Given

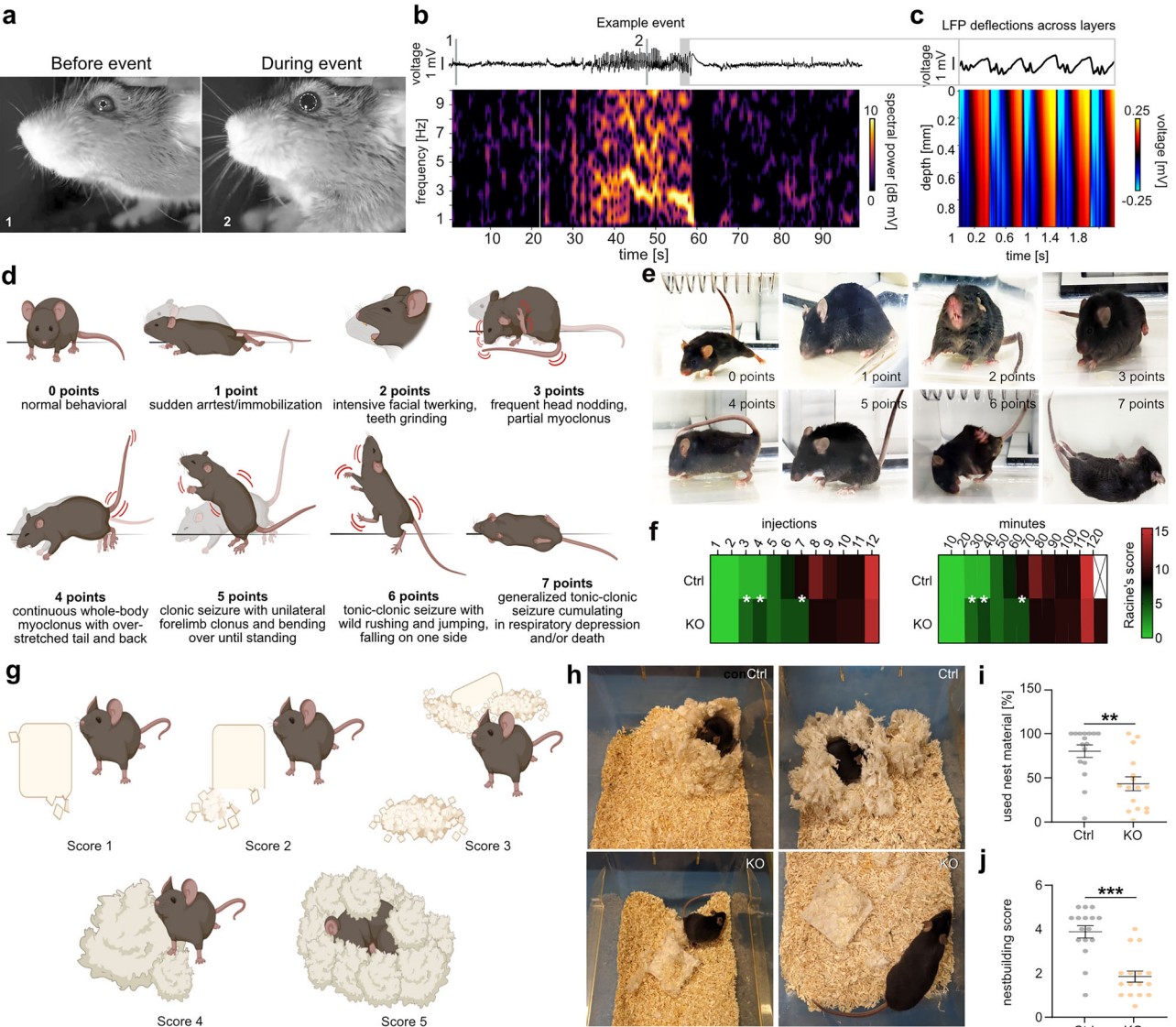

**Fig. 8 | Adult *Dnmt1*-deficient mice show behavioral abnormalities. a** Example images of a head-fixed *Sst-Cre/tdTomato/Dnmt1 loxP²* (KO) mouse before and during an epileptiform event with dilated pupil (white circles) and postural changes. **b** Example trace and spectrogram from cortical LFP recording during the event show a strong increase in theta oscillations for ~30 s. Times of video images 1 and 2 (white numbers in (**a**)) are shown as gray lines in the top trace. **c** Quantification of LFP deflections across depth for a 2-s time window during the event. Raw data are available via hyperlinks listed in the Data Availability Statement. (**d**–**f**) Pentylene-tetrazol (PTZ)-injections every 10 min in three-month-old *Sst-Cre/tdTomato* (Ctrl) and KO males. **d** Schematic illustrations defining PTZ-induced seizures using Racine's scoring (see methods). **e** Footages captured from video monitoring of PTZ-induced convulsions for respective Racine's scores, which are depicted in (**d**). **f** Heat maps and quantification of Racine's scale events associated with the given number of PTZ injections or the passed time. Within every column (illustrating the number

of applied injections (left) or minutes after the first injection of PTZ (right)), all detected epileptiform events were correlated to their respective severity score (0-7), added up, and averaged for each genotype (rows). Higher Racine's scale points (= severity levels) are indicated by red colors. Two-tailed Wilcoxon *rank-sum* test. *N* = 10 animals per genotype. **g**–**j** Nest building test for evaluating given material usage and forming proper nests in adult KO animals compared to Ctrl mice. With the help of the scoring index depicted in (**g**) and formerly described[117] (see methods), all mice were ranked based on nest quality and the amount of used material (**h**–**j**). Representative photographs are shown in (**h**) for control and KO mice. Tests were performed overnight and twice within one week. Respective values were averaged for both trials (**i**, **j**). Unpaired, two-tailed Student's *t* test and additional unpaired Welch's *t* test. *N* = 16 male adult mice for both genotypes. *$p < 0.05$, **$p < 0.01$, ***$p < 0.001$. Exact *p* values: Supplementary Data 10. Error bars: ±SEM.

that our findings in solution are also valid for in vivo conditions, they support the hypothesis that DNMT1 has a functional role beyond maintenance methylation. In line with this, DNMT1 has been shown to interact with CFP1 (CysxxCys finger protein 1), which presents a high affinity for unmethylated DNA[82]. Moreover, de novo methylation activity at certain repetitive elements and single-copy sequences has already been shown for DNMT1[83]. Functional cooperation of DNMT1 during de novo methylation of DNA has further been described[84], and gene-specific de novo methylation can be initiated by reintroduction of DNMT1 in cells lacking DNMT1 and DNMT3B, which present

with nearly absent genomic methylation[85]. Taken together, these observations strongly propose DNMT1´s function beyond maintenance DNA methylation, likely contributing to (de novo) DNA methylation-dependent gene regulation in postmitotic neurons. This provides an additional layer of epigenetic control during neuronal differentiation and function. In combination with its described non-canonical actions through crosstalk with histone-modifying mechanisms[20,86], our findings render DNMT1 as an active participant in epigenetic programming in postmitotic cINs and putatively also other neuronal subsets[27,28,35].

We found that *Dnmt1* deletion in SST⁺ cINs impairs their migration within the MZ, an effect likely restricted to Martinotti cells, constituting the majority of SST⁺ interneurons[64], which preferentially follow this route during corticogenesis[23]. Our integrated transcriptomic and DNA methylome analyses revealed increased expression and reduced methylation of several transcription factors essential for cIN development, including *Arx*, *Zeb2*, *Dlx2*, *Dlx5*, and *Tcf4*[11,39,87]. These results support a model in which DNMT1 represses specific transcriptional programs via promoter and gene body methylation. Other transcription factors such as *Sox6*, *Maf*, *Lhx6*, and *Satb1* also displayed elevated expression levels upon *Dnmt1* deletion, but without corresponding reductions in promoter methylation, suggesting indirect regulation—possibly via upstream factors like ARX or compensatory activity by DNMT3A.

Importantly, DNA methylation is not invariably linked to transcriptional repression. A recent study highlighted the context-dependence of promoter DNA methylation. While genes were found silenced by DNA methylation, being in line with the model of repressive promoter methylation, other genes were unaffected or even increased in expression, likely due to loss of methylation-sensitive repressors[88]. Moreover, the biological functions of DNA methylation extend beyond transcriptional control and include, e.g., modulation of alternative splicing[89,90] and promoter usage[91].

We identified *Arx* as a key downstream target of DNMT1-mediated repression, showing increased expression and reduced methylation at both the TSS and gene body in *Sst-Cre/tdTomato/Dnmt1-loxP²* cells.

ARX regulates numerous genes involved in various stages of cIN development, including *Maf*, *Mafb*, *Mef2c*, *Nkx2.1*, *Lmo1*, *Nrg1*, *Erbb4*, *Npr1*, and *EphA4*[16], many of which were dysregulated upon *Dnmt1* deletion. Thus, DNMT1 appears to shape the transcriptional landscape of SST⁺ cINs through both direct repression and indirect modulation of transcriptional networks centered on ARX.

Moreover, epigenetic modifiers such as the lncRNAs *Dlx6os1* and *Kcnq1ot1*, in addition to the chromatin modifiers *Atrx*, *Kmt2a*, *Ash1l*, and *Setd2*, are regulated by DNMT1, broadening the spectrum of DNMT1-mediated mechanisms that could converge on interneuron-specific transcriptional programs and chromatin landscapes.

The dysregulation of *Erbb4* and *Efnb2*, genes implicated in cIN migration, likely contributes to the premature invasion of SST⁺ cINs into the CP. ERBB4, activated by NRG3 in the CP, facilitates the tangential-to-radial switch in interneuron migration[18]. Moreover, the increased *Efnb2* expression in *Dnmt1*-deficient SST⁺ cINs could contribute to the precocious invasion of the CP. EPHB1 is expressed in the MZ[12], and EPHB1-triggered reverse signaling has already been shown to elicit a repulsive response in migrating interneurons[55]. Of note, DNA methylation-dependent regulation of *Efnb2* expression was already proposed in neuronal stem cells[92]. Thus, the upregulation of *Erbb4* and *Efnb2* in *Dnmt1*-deficient SST⁺ cINs supports a model where DNMT1-mediated repression controls the exit of interneurons from the MZ.

Despite the limitations of bulk sequencing approaches, particularly in resolving population heterogeneity, key transcriptional alterations relevant to the MZ migration defects were validated using single-nucleus RNA-seq. Additionally, some dysregulated genes—such as MEF2C and DLX5—play roles in cIN fate determination. Both promote PV lineage commitment[54,66,67] and were upregulated in embryonic *Dnmt1*-deficient SST⁺ cINs. MEF2C drives chromatin remodeling at PV-specific loci, initiating distinct transcriptional programs in immature PV⁺ interneurons upon settling within the cortex[66,67]. Consistent with this, we observed a modest increase in PV⁺ SST-lineage interneurons in the adult *Dnmt1* KO cortex, suggesting that DNMT1 also helps stabilize SST⁺ identity.

The altered expression of cell fate-determining genes such as *Mef2c* could contribute to the observed functional and behavioral defects seen in the adult *Sst-Cre/tdTomato/Dnmt1-loxP²* mice, even though we did not detect intrinsic abnormalities in SST⁺ cIN firing properties. However, the effects may be more subtle and require more detailed electrophysiological, morphological, or circuit-level analyses, potentially in combination with single-cell transcriptomic data across developmental stages—an approach that lies beyond the scope of the present study.

The detected functional defects could also result from structural disruptions evident in the adult *Sst-Cre/tdTomato/Dnmt1-loxP²* cortices, which likely arise from the non-cell autonomous effect of *Dnmt1* deletion in SST⁺ cINs on cortical progenitors. This early developmental disruption resulted in altered SST⁺ cIN densities in the adult cortex, thereby linking early migratory disturbances to the adult phenotype. Our data suggest that SST⁺ cINs influence the output of RGC divisions, as evidenced by changes in the IPC pool and the generation of deep-layer neurons upon *Dnmt1* deletion, perhaps through EPHA4/ephrinB2-mediated interactions. This is reminiscent of EFNA5-EPHA4 signaling, which modulates apical progenitor division mode and the timed production of neurons for the deep versus upper cortical layers[4]. The increased *Efnb2* expression in *Dnmt1*-deficient cINs and its known role in cortical neurogenesis support this hypothesis. While progenitor-restricted *Efnb2* deletion delays neurogenesis, EFNB2-driven EPHB signaling transiently boosts neuronal output[57], mirroring our pair cell assay results. These findings support a crosstalk between *Efnb2*-expressing SST⁺ cINs and RGCs. Of note, the reported EFNB2-dependent neurogenic shift occurred within a specific temporal window of corticogenesis[57], aligning with prior studies showing that temporally restricted neurogenic changes, like we found in our study, shape cortical layer thickness[4,93].

As already evidenced by Sessa et al.[56], the development of cINs and excitatory neurons must have been effectively integrated throughout evolution, since an increase in the production of excitatory neurons also necessitates a rise in interneuron generation to maintain the proper balance of excitation and inhibition. Invading cINs may likewise influence cortical progenitors, as it had been shown for IPCs[8]. Our findings extend this hypothesis by providing evidence that cINs also impact RGCs, and through this, the generation of IPCs and neurons.

Their altered density in the superficial layers of *Dnmt1* KO mice might contribute to the observed functional and behavioral deficits. In support of this, functional impairments were most pronounced in these layers. LFP recordings showed weaker, delayed tactile responses with disrupted spatiotemporal processing, reduced activation, and prolonged spiking activity, particularly in the superficial layers. Alongside diminished γ-oscillations, low-frequency activity was increased. Additionally, abnormal eye movements with severe pupil dilations correlated with heightened low-frequency activity (1–4 Hz) in superficial layers, reinforcing the link between structural changes and functional deficits.

Although optogenetic stimulation revealed no intrinsic deficits in SST⁺ cIN firing properties upon *Dnmt1* deletion, we cannot exclude additional DNMT1-dependent functional impairments in adult SST⁺ cINs, akin to DNMT1's role in regulating GABAergic transmission in PV⁺ interneurons[28]. Future studies using inducible *Sst-Cre/tdTomato/Dnmt1 loxP²* mice to analyze methylation and gene expression signatures in adult SST⁺ cINs could clarify embryonic versus adult effects, though this is beyond the scope of the present study.

The functional deficits of adult *Sst-Cre/tdTomato/Dnmt1-loxP²* mice could additionally be a direct consequence of the impaired migration, as cIN migration and terminal differentiation are closely linked. In Martinotti cells, the proper formation of long-range axon collaterals in layer I depends on their migration along the MZ[23], enabling them to modulate numerous pyramidal neurons as primary targets[64]. Thus, the formation of these powerful axon collaterals in layer I could be compromised upon *Dnmt1* deletion, similar to what was described for the *Mafb* KO mice[23].

In sum, the altered expression of signaling molecules, together with disrupted migration of *Dnmt1*-deficient SST⁺ cINs, likely underlie the observed defects in cortical layering, resulting in altered interneuron densities. This highlights a key role for cINs in timing excitatory neuron production and emphasizes extensive crosstalk between

immature interneurons and progenitors. Our findings thus position DNMT1 as a central regulator of cortical development via both cell-autonomous and non-cell-autonomous mechanisms.

## Methods

### Molecular dynamics simulations

The *AMBER 22* software suite was employed[94]. Details on the DNMT1/UMDNA/SAM complex and force field parameters are given in the Supplementary Fig. S1 and Supplementary Table S3–5. Long-range electrostatic interactions were calculated using the Particle Mesh Ewald method[95]. Non-bonded interactions were treated with a 10 Å cutoff.

The system was energy-minimised in vacuo with 10,000 steps of steepest descent, followed by 10,000 steps of conjugate gradient minimization in order to eliminate atomic collision. The heavy-atom RMSD of the optimized configurations with respect to the initial geometry was 1.4 Å. Then, the complex was solvated in a truncated octahedral water box with a minimum distance of 12 Å from the solute to the box edge (Supplementary Table S5). $K^+$, $Na^+$, and $Cl^-$ ions were added to neutralize the system. The concentration of $K^+$ was similar to that in the nucleus (Supplementary Table S5). Periodic boundary conditions were applied. The whole system was then subjected to (i) 10,000 steps of steepest descent followed by 10,000 steps of conjugate gradient minimization with a 100 kcal/(mol·Å²) restraint on the whole solute. (ii) the same as (i), but with restraints only on heavy atoms of the solute; (iii) the same as (i), but without restraints. Successively, the system was heated from 0 K to 100 K over 5 ps, then relaxed at 100 K for 5 ps, then from 100 K to 310 K in 0.5 ns, and finally relaxed at 310 K for 4.5 ns by using Langevin dynamics[96], with a restraint of 100 kcal/(mol Å²) was applied to the heavy atoms of the solute. The time integration step was set to 1 fs. Finally, three independent 500 ns isobaric-isothermal simulations with different initial velocities were performed without restraints and with a time step of 2 fs. Langevin dynamics[96] and the Monte Carlo barostat[97] were used to maintain a constant temperature (310 K) and pressure (1 atm), respectively. *CPPTRAJ*[98] was used for trajectory analyses.

### Animals

Transgenic mouse strains with a genetic C57BL/6 J background (initially obtained from the University Hospital UKA Aachen, Germany) were used. *Sst^{tm2.l(cre)Zjh}/*J x *B6.CgGt(ROSA)26Sor^{tml 4(CAG−tdTomato)Hze}* (*Sst^{+/−}*-Cre/tdTomato) served as control animals whereas *Sst^{tm2.l(cre)Zjh}*/J x B6.CgGt(ROSA) 26Sor^{tm14(CAG−tdTomato)Hze} x *B6; 129S-Dnmt1tm2Jae/*J (*Sst-Cre^{+/−}/tdTomato/ Dnmt1 loxP^2*) were used as *Dnmt1* knockout (KO) model. Detailed information on mouse strains, animal housing, and ethics oversights is provided in the Supplementary Information and Supplementary Fig. S2a. Sex was not distinguished in embryonic analyses due to technical limitations during the experimental procedures. For adult experiments, only male mice (3–6 months old) were used to avoid confounding genotype-specific effects by hormonal fluctuations associated with the female estrous cycle. The number of embryos and adult mice used is specified in the respective figure legends and methods.

### Isolation of embryonic and adult brains

Individuals of embryonic stage 14.5, 16.5, or 18.5 were isolated as described in Symmank et al.[99]. Briefly, pregnant females were anesthetized by intraperitoneal administration of ketamine/xylazine (200/25 mg/kg living weight per injection up to a maximum application of 600/75 mg ketamine/xylazine per kg living weight in total). Upon reaching surgical tolerance, the abdominal cavity was opened to expose the uterine horns that were removed. Embryos were isolated from the uteri and decapitated. For histological analyses of E14.5 and E16.5 brains, heads were directly transferred to 4% paraformaldehyde (PFA)/1× phosphate-buffered saline (PBS), while at E18.5, brains were removed from the skull before transferring them to 4% PFA/1× PBS. E14.5 heads were fixed for 5 h, while E16.5 heads and E18.5 brains were

fixed overnight at 4 °C on an orbital shaker (50 rpm). Afterwards, stepwise cryopreservation was performed by overnight incubation first in 10% Sucrose/1× PBS, followed by 30% sucrose/1× PBS at 4 °C on an orbital shaker (50 rpm). Tissue was frozen in liquid nitrogen and stored at −80 °C.

For the isolation of adult brains, male mice were sacrificed with an overdose of 5% (v/v) isoflurane. For histological analyses and identification of Neuropixels recording sites, transcardial perfusion was conducted with 1× PBS (pH 7.4) followed by 4% PFA/1× PBS (pH 7.4) with a pump[100]. After brain preparation, post-fixation was conducted in 4% PFA/1× PBS for 24 h at 4 °C on a roller mixer with constant rotation (approx. 50 rpm). Then, cryopreservation was performed in 10% sucrose/1× PBS and 30% sucrose/1× PBS for 24 h each at 4 °C on a roller mixer with constant rotation (~50 rpm) before brains were frozen in liquid nitrogen and stored at −80 °C.

### Organotypic brain slices and single-cell preparations

E14.5 MGE- and cortical single cells were prepared according to previous studies[12,28]. The dorsal part of the MGE and the medial part of the cortex were dissected in Gey's Balanced Salt Solution (GBSS, pH 7.4)/0.65% D-glucose on ice and collected in Hanks' Balanced Salt Solution (HBSS, w/ phenol red, w/o calcium, w/o magnesium)/0.65% D-glucose on ice. After treatment with 0.04% of trypsin/EDTA (Thermo Fisher Scientific, USA) for 17 min at 37 °C, the tissues were dissociated in cold Dulbecco's Modified Eagle Medium (DMEM) with additional L-glutamine and 4.5 g/L D-glucose (Thermo Fisher Scientific, USA), 10% fetal bovine serum (FBS; Biowest, USA), and 1% penicillin/streptomycin (P/S; Thermo Fisher Scientific, USA) by trituration with glass Pasteur pipettes and by subsequent filtering through a nylon gauze (pore size 140 μm; Merck, USA). Dissociated MGE cells were seeded on laminin (19 μg/mL; Sigma-Aldrich, USA)/poly-L-lysine- (10 μg/mL; Sigma-Aldrich, USA) coated glass coverslips at a density of 455 cells/mm²[2]. Cortical cells were seeded at clonal densities (150 cells/mm²). After 5–6 h of incubation (37 °C, 5% CO₂ and 95% relative humidity) in Neurobasal medium with phenol red, 1× B27™, and 0.25x GlutaMAX (Gibco, USA), MGE cells were transfected with *Dnmt1* siRNA (30 nM, #sc-35203, Santa Cruz, USA) and scrambled control siRNA oligos (15 nM, BLOCK-iT™ Fluorescent Oligo, #2013, Thermo Fisher Scientific, USA) for 24 h using Lipofectamine™ 3000 following the manufacturer's instructions (Thermo Fisher Scientific, USA). For fixation, 4% PFA/1× PBS was applied for 10 min.

Cortical cells plated at clonal density were incubated in culture medium (Neurobasal medium with phenol red, 1× B27™, 0.25x Gluta-MAX (Gibco, USA), and 0.4% methylcellulose (Sigma, USA)) for 5–6 h at 37 °C, 5% CO₂, and 95% relative humidity. Afterwards, a treatment with either 5 μg/mL of a recombinant human Fc control protein (Rockland Immunochemicals, USA) or 5 μg/mL of a recombinant human efnB2-Fc (R&D Systems, USA), pre-clustered with 10 μg/mL of an anti-human IgG antibody (Thermo Fisher Scientific, USA), was conducted for 24 h, prior to fixation with 4% PFA/1× PBS for 10 min.

For organotypic brain slice preparations[20], brains of E14.5 embryos were embedded in 4% low-melt agarose (37 °C) in Krebs buffer (126 mM NaCl, 2.5 mM KCl, 1.2 mM NaH₂PO₄, 1.2 mM MgCl₂ * 6H₂O, 2.1 mM CaCl₂; pH 7.4; supplemented with 10 mM D-glucose and 12.5 mM NaHCO₃, sterile-filtered) and cut into 350 μM coronal slices, using a 5100mz-Plus vibrating microtome (speed: 0.7–0.8 mm/s and blade oscillation frequency: 5 Hz; Campden Instruments, United Kingdom). Sections were collected in ice-cold post-holding buffer (1× Krebs buffer, 10 mM HEPES, 1% P/S, 0.2% gentamicin, pH 7.4), and then transferred to μ-slide 4 Well imaging plates (ibidi GmbH, Germany) coated with 19 μg/mL laminin (Sigma-Aldrich, USA) and 10 μg/mL poly-L-lysine (Sigma-Aldrich, USA) in Neurobasal without phenol red, 1× B27™, 1% P/S, 0.5% D-glucose, 10 mM HEPES. Life cell imaging was performed in tile scans including z-stacks of 8–10 μm step size every 15 min with a confocal-like Leica DMi8 fluorescent microscope in

combination with a THUNDER® imager unit (Leica, Germany) and the corresponding software *LASX* (Leica, Germany) equipped with an incubation chamber (37 °C, 5% CO$_2$, and 95% relative humidity; excitation wavelength of 544 nm with TRITC emission filter). Post-processing was executed in the *LASX* software using the *"Mosaic Merge"*- and *"Thunder Lightning (Large Volume)"* tools. A maximum intensity projection of the merged tile scans was processed using the *Fiji* software[101] and *LASX*. Analysis was performed blindly using the manual track plugin from *ImageJ* (NIH, USA) to track cells with a migration time of at least 6 h.

### Histology and immunocytochemistry

For embryonic brains, on-slide immunohistochemistry was conducted on 50-μm coronal sections (CM3050 S Cryostate (Leica, Germany)), collected on Superfrost®Plus (Avantor, USA) object slides as described in Pensold et al.[20]. All samples were washed for $5 \times 20$ min in $1\times$ PBS /0.5% Triton X-100/0.5% Tween®20 with constant horizontal shaking (50 rpm) and treated with blocking solution (4% bovine serum albumin (BSA) and 10% normal goat serum in $1\times$ PBS/0.5% Triton X-100/0.5% Tween®20) for 2 h at room temperature (RT). Primary antibody incubation diluted in blocking solution was conducted overnight at RT in a humid chamber. After washing ($5 \times 20$ min) in $1\times$ PBS/0.5% Triton X-100/0.5% Tween®20, secondary antibodies were applied in blocking solution for 2 h at RT. After washing for $3 \times 20$ min, samples were stained with 4′,6-diamidino-2-phenylindol dihydrochloride (DAPI, 1:10,000 in PBS, Carl Roth, Germany) for 15 min at RT, followed by washing twice for 5 min with PBS. Sections were embedded in Mowiol or Fluoromount (Thermo Fisher Scientific, USA).

For adult brains, immunohistochemistry was performed in free-floating sagittal and coronal sections (30 μm; CM3050 S Cryostate (Leica, Germany)). Slices were transferred to $1\times$ PBS and stored at 4 °C. For sagittal sections, Bregma 1.32 and 1.44 were used, while Bregma 1.18, 1.10, 0.14, −0.22, −3.08, and −3.28 were taken for coronal sections. The same procedure as for embryonic sections was applied, except that antigen retrieval with heated HistoVT One® ($1\times$ HistoVT One®/H$_2$O bidest., Nacalai, Japan) at 70 °C for 20 min was performed for staining against DNMT1. Sections that were stained for SST, calretinin, SOX2, and NPY underwent an antigen retrieval with citrate buffer (95 °C; 10 mM, pH 6.0, supplemented with 0.5% Tween®20) for 15 min and subsequent cool-down for 30 min before washing. Finally, all sections were transferred onto glass slides in 0.5% (w/v) gelatine/0.05% (w/v) chromium(III) potassium sulfate dodecahydrate (KCrS$_2$O$_8$)/H$_2$O solution prior to embedding in Mowiol.

Immunostaining on dissociated MGE and cortical single cells was conducted by washing coverslips with $1\times$ PBS/0.1% Triton X-100 for $3\times5$ min and blocking with $1\times$ PBS/0.1% Triton X-100/4% BSA for 30 min at room temperature. Primary antibodies were applied for 2 h at room temperature. After washing for $3\times5$ min with $1\times$ PBS/0.1% Triton X-100, the secondary antibody was incubated for 1 h at room temperature. Phalloidin-647 diluted 1:1000 in $1\times$ PBS (#ab176759, Abcam, USA) was applied for 20 min after washing. Then, a final washing step with $1\times$ PBS for 10 min and a DAPI staining (1:10000/$1\times$ PBS; Carl Roth, Germany) for 5 min were performed, and coverslips were mounted in Mowiol.

Following primary antibodies were used: mouse anti-Calretinin (1:500; Swant, Switzerland, #6B3), rabbit anti-NPY (1:2500; Immunostar, USA, #2940), rabbit anti-TBR1 (1:200; Abcam, USA, #ab31940); rabbit anti-EOMES (1:500; Abcam, USA, #ab23345); rat anti-SST(1:100; Millipore, USA, #MAB354), mouse anti-Parvalbumin (1:2000; Swant, Switzerland, #235); rabbit anti-DNMT1 (1:100; Santa Cruz, USA, # sc20701); rabbit anti-CUX1 (CDP; 1:100, Santa Cruz, USA, #sc13024); mouse anti-Nestin (1:100; Merck USA, #MAB353), rabbit anti-ß-Tubulin III (1:500; Sigma Aldrich, USA, #T2200), mouse anti-SOX2 (1:200; #MA1-014, Invitrogen, USA) and rabbit anti-ERBB4 (1:1000; Proteintech, USA, #22387-1-AP).

Following secondary antibodies conjugated with respective fluorophores were used at 1:1000 dilutions: Alexa488-goat anti-Rat IgG (Invitrogen, USA, #A11006) Cy5-Goat anti-Rabbit IgG (Life Technologies, USA, #A10523), A488-Donkey anti-Mouse IgG (Jackson, USA, #15454150), Cy5-Goat anti-Mouse IgG (Jackson, USA, #115175146), A488-Goat anti-Rabbit IgG (Life Technologies, USA, #A11008).

Detailed information on the microscopes and the settings for capturing immunocyto- and histochemical stainings is provided in the supplementary methods.

### Transcriptome data on neurons overexpressing DNMT1

*Dnmt1^tet/tet* mouse embryonic stem cells have been previously described and obtained by Prof. K. Naga Mohan (Hyderabad University, Hyderabad, India)[102]. The levels of DNMT1 in these cells are four times higher than those of the normal wild-type *R1* ESCs. Upon differentiation, the induced *Dnmt1^tet/tet* neurons produced contain twice the DNMT1 levels as in *R1* neurons[102]. Differential expression gene data on *Tet/Tet* neurons were taken from Singh et al.[36] and genes with significantly altered transcript levels were compared with the transcriptome data on the tissues lacking DNMT1.

### Enrichment of E14.5 tdTomato$^+$ cells for RNA and methyl-sequencing

For FACS-mediated enrichment of *Sst-Cre/tdTomato* cells, telencephalons were prepared from E14.5 embryos and subjected to cell dissociation. Nuclease-free reaction tubes were used during isolation and long-term storage of the resulting material. The telencephalons were collected in cold HBSS (w/ phenol red, w/o calcium, w/o magnesium)/0.65% D-glucose; 4 μg/μL (600 U) of DNAse I (AppliChem GmbH, Germany). After treatment with 0.04% trypsin for 17 min at 37 °C, HBSS was replaced by DMEM with additional L-glutamine and 4.5 g/L D-glucose, 10% FBS, and 1% P/S to stop the trypsinization. Subsequently, the cells were pelleted, resuspended, and triturated in cold HBSS (w/o phenol red, w/o calcium, w/o magnesium)/0.65% D-glucose, before being filtered through a nylon gauze (pore size 140 μm, Merck, USA) for FACS.

FACS was performed by the Flow Cytometry Facility (FCF, University Hospital RWTH Aachen, Germany). Respective parameters for the procedure using a BD FACS Aria Fusion (BD Biosciences, USA) were defined as follows: 5-laser (FCS, SSC, PE, BV421), 18-color (3-6-2-4-3). TdTomato-positive cells were either collected in 100 μL of cold TRIzol™ (Thermo Fisher Scientific, USA) for subsequent RNA sequencing or in 100 μL of cold HBSS (w/o phenol red, w/o calcium, w/o magnesium)/0.65% D-glucose for further processing for DNA methylation analysis. Finally, all samples were stored at −80 °C.

### Total RNA seq

TdTomato-positive E14.5 *Sst-Cre/tdTomato or Sst-Cre/tdTomato/ Dnmt1 loxP$^2$* cells from the basal telencephalon and the cortex were FAC-sorted into 100 μL TRIzol™ reagent (Thermo Fisher Scientific, USA). Samples were mixed by inversion and stored at −80 °C until processing. RNA was isolated from 50,000 cells pooled per genotype. The samples were filled up to 1 mL with TRIzol™ and dounced 25 times in a glass homogenizer using the small-clearance pestle. Samples were incubated for 5 min at room temperature, supplemented with 200 μL of chloroform, and mixed by shaking vigorously until a homogenous milky solution appeared. Following another incubation at room temperature (until a visible phase separation appeared), samples were centrifuged at $13,000 \times g$ and 4 °C for 15 min in a tabletop centrifuge with cooling function. For precipitation, the clear upper phase was collected and mixed with 70% ethanol. Samples were bound onto RNeasy mini columns (Qiagen, Germany) and washed according to the instructions in the manual. RNA was eluted with 30 μL nuclease-free water.

RNA concentrations were determined using a Qubit™ 4 fluorometer (Thermo Fisher Scientific, USA) with RNA HS reagents (Thermo Fisher Scientific, USA). RNA integrity was checked on a Tape Station 4200 using the HS RNA kit (Agilent Technologies, USA).

Prior to library preparation, samples were digested with HL dsDNAse I (ArcticZymes Technologies ASA, Norway) for 10 min at 37 °C to remove traces of gDNA. To avoid digestion of the newly generated cDNA during library preparation, the dsDNAse I was heat-inactivated for 5 min at 58 °C. Thereafter, samples were subjected to library preparation using the TAKARA SMARTer Stranded Total RNA-Seq Kit v3 - Pico Input Mammalian for Illumina (Takara Bio, Japan). Fragmentation was done at 94 °C for 4 min. Illumina adaptors and indices were ligated to cDNA in a PCR reaction with five cycles (98 °C for 15 s, 55 °C for 15 s, and 68 °C for 30 s). RRNA depletion was performed on the amplified libraries. Final libraries were subjected to an upscale PCR with 13 cycles (98 °C for 15 s, 55 °C for 15 s, and 68 °C for 30 s). Amplified libraries were cleaned with magnetic AMPure XP beads (Beckman Coulter, USA). Library concentration was determined with a Qubit™ 4 fluorometer using the 1× dsDNA HS reagent (Thermo Fisher Scientific, USA). Library size was assessed on a Tape Station 4200 using the DNA D1000 kit (Agilent Technologies, USA).

Libraries were diluted, equimolarly pooled, denatured, and loaded for clustering onto an Illumina NovaSeq 6000 SP (200 cycles) flow cell (Illumina, USA) in conjunction with a 1% PhiX control library, and run on an Illumina NovaSeq machine (Illumina, USA) in 75 bp paired-end mode. The 1% PhiX control library was spiked in to improve base calling accuracy.

*FASTQ* files were generated using *bcl2fastq* (Illumina, USA). To facilitate reproducible analysis, samples were processed using the publicly available *nf-core/rnaseq* pipeline version 3.12[103] implemented in *Nextflow* 23.10.0[104] with minimal command. In brief, lane-level reads were trimmed using *Trim Galore* 0.6.7[105] and aligned to the mouse genome (GRCm39) using *STAR* 2.7.9a[106]. Gene-level and transcript-level quantification was done by *Salmon* v1.10.1[107]. All analysis was performed using custom scripts in *R* version 4.3.2 using the *DESeq2* v.1.32.0 framework[108].

## MethylSeq

TdTomato-positive E14.5 *Sst-Cre/tdTomato* and *Sst-Cre/tdTomato/Dnmt1 loxP²* cells were FAC-sorted into 100 μL HBSS. Samples were mixed by inversion and stored at −80 °C until processing.

DNA was isolated from at least 50,000 pooled cells per sample using the PureLink genomic DNA Mini Kit (Thermo Fisher Scientific, USA). Per genotype, the samples were initially pooled by volume (200 μL each), supplemented with lysis buffer provided in the kit, and incubated with 2 μL proteinase K (Thermo Fisher Scientific, USA) at 55 °C for 10 min. DNA was precipitated with 100% ethanol. Samples were bound to one column per genotype each, washed as instructed, briefly dried, and eluted with 35 μL Tris-EDTA (TE) buffer. DNA concentrations were determined using the Qubit™ 4 fluorometer with the 1× dsDNA HS reagent (Thermo Fisher Scientific, USA). Quality/integrity was checked on a Tape Station 4200 using the Genomic DNA kit (Agilent Technologies, USA).

Libraries were generated using the Enzymatic Methyl-seq kit for Illumina (#E7120, New England Biolabs, USA). After enzymatic shearing for 25 min at 37 °C with the NEBNext UltraShear enzyme and buffer, followed by an inactivation at 65 °C for 15 min, 5-methylcytosines were oxidized by TET2 after adaptor ligation. Samples were denatured with formamide and deaminated with APOBEC. Libraries were subjected to 7 cycles of upscale PCR (98 °C for 10 s, 62 °C for 30 s, and 65 °C for 60 s). Library concentrations were determined with a Qubit™ 4 fluorometer and the 1× dsDNA HS reagent (Thermo Fisher Scientific, USA). All recommended clean-up steps were performed with magnetic AMPure XP beads (Beckman Coulter, USA). Library size was assessed on a Tape Station 4200 using the DNA D5000 kit (Agilent

Technologies, USA). Concentrations were determined with a Qubit™ 4 fluorometer and the 1× dsDNA HS reagent (Thermo Fisher Scientific, USA).

Libraries were diluted, equimolarly pooled, denatured, and clustered onto a NovaSeq 6000 S1 v1.5 (300 cycles) flow cell (Illumina, USA). The 1% PhiX control library was spiked in to improve base calling accuracy. Paired-end sequencing was performed with 151 cycles.

*FASTQ* files were generated using *bcl2fastq* (Illumina, USA). To facilitate reproducible analysis, samples were processed using the publicly available *nf-core/methylseq* pipeline version 3.0.0[103] implemented in *Nextflow* 24.10.5[104] with the minimal command. Briefly, *FastQC* (http://www.bioinformatics.babraham.ac.uk/projects/fastqc) and *Trim Galore*[109] were utilized to perform quality control, and clean reads were aligned to the reference genome using *Bismark*[110] to account for bisulfite conversion and identify methylated cytosines. Differentially methylated sites and regions were called by the *R* package *DSS*[111] and annotated using the *GENCODE VM23* dataset. Regions were similarly annotated with a margin of 3 kb in front of the TSS. Methylation tracks were converted from mm10 to mm39 using *rtracklayer*[112] for integration with RNA seq data and visualized in *IGV*[113]. The sequence of the promoter regions, defined as 3 kb before and after the TSS, was further analyzed for the enrichment of previously identified DNMT1-binding motif using the *AME* algorithm provided within the *MEME suite*[114]. Gene sets of interest were analyzed for the enrichment of gene ontology terms using *ShinyGO* 0.80/0.82 with the *biological processes'* dataset[115].

## Single-cell RNA-sequencing

The dorsal telencephalons from E14.5 C57BL/6 J mice were dissected. Single-cell suspensions were prepared using a papain dissociation system (#LK003150; Worthington Biochemical Corporation, USA), and all centrifugation steps were done at 4 °C. Cells were manually counted using Trypan Blue exclusion, which consistently showed a viability greater than 95%.

ScRNA-seq libraries were prepared using the 10x Genomics 3' Gene Expression Kit v3.1 (10x Genomics, USA. Each sample had a target of 10'000 cells and was independently loaded in the Chromium Controller (10x Genomics, USA). Quality control of the cDNA and final libraries was conducted using the High Sensitivity RNA D5000 and D1000 ScreenTape assays (Agilent Technologies, USA), respectively. Libraries were pooled and sequenced on a NovaSeq 6000 using the S2 Reagent Kit v1.5 with 100 cycles (Illumina, USA.

*FASTQ* files were processed using *CellRanger* v7.0.0. All bioinformatics analyses were performed in *R* 4.4.0 running under *Ubuntu* 20.04.6 *LTS*. The *Seurat* v5 package[116] was used for filtering, quality control, clustering, and annotation of the data. *CellChat* v2.1.2[58] was used to compute the cell communication probabilities among different clusters.

## Single-nucleus RNA-sequencing

The medial neocortices from E16.5 *Sst-Cre/tdTomato* and *Sst-Cre/tdTomato/Dnmt1 loxP²* mice were dissected. The samples were snap-frozen on dry ice and stored at −80 °C until library preparation to ensure RNA integrity was maintained while the genotype and the sex of each sample could be verified for a bias-free transcriptomic analysis. Consequently, single-nucleus RNA sequencing (snRNA-seq) was selected over scRNA-seq to accommodate the processing of frozen samples. The nuclei were extracted from the frozen tissue by discontinuous sucrose gradient ultracentrifugation. Briefly, the tissue was homogenized in 1 mL of lysis buffer (0.32 M sucrose, 10 mM Tris-HCl (pH 8.0), 5 mM CaCl2, 3 mM Mg acetate, 1 mM dithiothreitol (DTT), 0.1 mM EDTA, 0.1% Triton X-100, 50 U/mL RNase inhibitors (#M0314S, New England Biolabs, USA)) using a 2-mL glass Dounce homogenizer on ice and stroking 25 times with pestle A followed by 100 strokes with pestle B. The homogenates were transferred to 5-mL ultracentrifuge

tubes (#344057, Beckman Coulter) and under-layered with 3 mL of sucrose cushion (1.8 M sucrose, 10 mM Tris-HCl (pH 8.0), 3 mM Mg acetate, 1 mM DTT, 50 U/ml RNase inhibitors). The samples were loaded onto a Sw-55 Ti rotor and centrifuged for 1 h at 4 °C and 28,300 rpm (97,500x $g$). The nuclei pellets were resuspended in 150 μl of cold 1% BSA in PBS and immediately fixed using the Evercode™ Nuclei Fixation kit (Parse Biosciences, USA) following the manufacturer's guidelines. The fixed nuclei were stored at −80 °C until library preparation.

The snRNA-seq libraries were generated with the Evercode™ WT kit (Parse Biosciences, USA). The quality check of the cDNA and the final libraries was performed using the High Sensitivity D5000 and D1000 ScreenTape kits (Agilent Technologies, USA). Libraries were equimolarly pooled and sequenced in conjunction with 5% PhiX control library on an Illumina NovaSeq machine in paired-end mode using the S2 Reagent Kit v1.5 (200 cycles; Illumina, USA).

The sequencing dataset was processed using *Trailmaker*™ (https://app.trailmaker.parsebiosciences.com/; Parse Biosciences, 2024). Unfiltered count matrices were further processed using the *Seurat* package for R (R 4.4.1, Seurat 5.0.1.9001)[116]. Cells were filtered based on counts, discarding cells below 1500 and above 60000 reads per cell. Dead or dying cells were removed by filtering droplets with high mitochondrial content (10% cut-off).

Subsequently, data was normalized prior to principal-component analysis (PCA) and Leiden-clustering. A Uniform Manifold Approximation and Projection (UMAP) embedding was calculated to visualize the results. Cluster-specific marker genes were identified by comparing cells of each cluster to all other cells using the *presto* package implementation of the Wilcoxon *rank-sum* test. The top marker genes for each cluster were cross-referenced with known cell type-specific markers from the literature and publicly available databases.

The Annotated Clusters were subsequently used to infer cell-cell communication probabilities using the *CellChat* Database and corresponding *R* package (*CellChat* version 2.1.2)[58].

## MERSCOPE analysis

Spatial transcriptomic analysis was performed on 10 μm brain sections from fresh-frozen E16.5 *Sst-Cre/tdTomato* and *Sst-Cre/tdTomato/Dnmt1 loxP²* embryos using the MERSCOPE™ system (Vizgen, USA) according to the manufacturer's instructions. Briefly, tissue samples were mounted onto specialized slides (MERSCOPE slides, #20400001, Vizgen, USA) and fixed to preserve RNA integrity. Multiplexed Error-Robust Fluorescence In Situ Hybridization (MERFISH) was then applied, where gene-specific oligonucleotide probes hybridized to target RNA molecules. We used the commercially available Pan Neuro panel, covering 500 transcripts used for the identification of cell types in the mouse Brain.

Following hybridization, high-resolution fluorescence imaging was conducted on the MERSCOPE platform to capture the spatial distribution of individual transcripts at subcellular resolution. The imaging data was preprocessed on the instrument, which decodes RNA signals, assigns molecular identities, segments the tissue into cells, and quantifies transcript abundance across individual cells within the tissue.

Two brain tissue sections from comparable anatomical planes were analyzed using *Seurat* in R (R 4.4.1, Seurat 5.0.1.9001). After normalization and identification of highly variable features for each dataset, the datasets were integrated using reciprocal PCA to account for shared sources of variation. Clustering was performed using the *Louvain* algorithm at a resolution of 0.8, and clusters were manually annotated based on marker genes identified through differential expression analysis. Heterogeneous clusters were further resolved through subclustering, with iterative marker analysis and annotation revealing additional distinct cell types and states within the brain tissue.

## Neuropixels recordings and behavioral experiments

**Surgical procedures.** Mice received systemic analgesia prior to surgical procedures in form of subcutaneous injections of carprofen (4 mg/kg, Rimadyl, Zoetis GmbH, Germany) and buprenorphine (0.1 mg/kg, Buprenovet sine, Bayer Vital GMBH, Germany) and were then anesthetized using (1%–2.5%) isoflurane in oxygen. The eyes were covered with eye ointment (Bepanthen, Bayer Vital GmbH, Germany) to prevent them from drying out. Following a local injection of bupivacaine (0.08 mL of 0.25% Bucain 7.5 mg/mL, Puren Pharma GmbH, Germany), the scalp was incised and pushed outward to fix it in place using tissue adhesive (Vetbond, 3 M). A circular headbar was attached on top using dental cement (C&B Metabond, Parkell USA; Ortho-Jet, Lang Dental). For electrophysiological recordings, craniotomies were performed to inject a viral vector into the cortex (*AAV1.-shortCAG.dlox.hChR2(H134R).WPRE.hGHp*, Zurich Vector Core, viral titer = -8 × 10^{12} vg/mL). We targeted primary somatosensory (S1) and visual (V1) cortex (stereotactic coordinates; S1: −1.5 mm anteroposterior and −3.5 mm mediolateral from Bregma; V1: −4 mm anteroposterior and −2 mm mediolateral from Bregma) and injected a volume of 200 nL in bouts of 10 nL with a 14-s delay between bouts (Nanoject III, Drummond Scientific, USA). For each coordinate, injections were performed at two cortical depths (300 and 600 μm), adding up to a total volume of 400 nL per injection site. The glass pipette was removed at least 5 min after the full volume was injected to reduce the risk of backflow. After viral injections, a 3-mm wide round coverslip was placed inside the craniotomy and fixed with light-curable dental cement (DE Flowable composite, Nordenta, Germany). The coverslips contained a small opening (-0.2 mm) that was covered with silicon to allow access of the Neuropixels probes to the cortex. After the surgery, all animals received the same injections of carprofen and buprenorphine as in the beginning. They also received buprenorphine (0.009 mg/mL, Buprenovet sine, Bayer Vital GmbH, Germany) and enrofloxacin (0.0227 mg/mL, Baytril 5%, Bayer Vital GMBH, Germany) in their drinking water for three days following the surgery. Behavioral training started after at least two weeks of recovery

## Neuropixels recordings

Electrophysiological recordings were done with Neuropixels 1.0 Probes in head-fixed mice, freely running on a wheel. For each mouse, we performed four recordings on consecutive days. To later recover the probe position in each recording, the probes were painted with DiD cell labeling solution (Invitrogen V22887, USA) before each recording. To record neural responses to tactile stimulation across all cortical layers, the probes were inserted at an orthogonal angle to the brain surface (-10°). The stimulation protocol was started 5–10 min after insertion. We recorded high pass-filtered data above 300 Hz at 30 kHz and low pass-filtered signals between 0.5 and 100 Hz at 2.5 kHz from the bottom 384 channels of the Neuropixels probe (˜3.8 mm active recording area). Signals were acquired with an external Neuropixels PXIe card (IMEC, Belgium) used in a PXIe-Chassis (PXIe-1071, National Instruments, USA). Triggers and control signals for different stimuli and wheel movements were separately recorded as analog and digital signals using the *SpikeGLX* software (Janelia Farm Research Campus, USA; Bill Karsh).

Visual stimulation to induce gamma oscillations consisted of 5-s long full-field drifting square wave gratings at horizontal orientation and a spatial frequency of 0.04 cycles per degree. All visual stimuli were presented at a 17 cm distance from the right eye on a gamma-corrected LED-backlit LCD monitor (Viewsonic VX3276-2K-MHD-2, 32", 60 Hz refresh rate). Tactile stimulation was controlled by a microcontroller-based finite state machine (Bpod r2, Sanworks, USA) using custom *Matlab* code (2020b, MathWorks). Tactile stimuli consisted of 20-ms long air puffs at 0.03 bar pressure directed at the distal parts of the whiskers. In each trial, two air puffs were presented at an inter-stimulus interval of 500 ms. The overall stimulation protocol

consisted of 50 tactile trials and 50 visual trials in randomized order. Inter-trial intervals were also randomized between 3 and 5 s.

To extract spiking activity, channels that were broken or outside of the brain were detected and removed from the recording using the *SpikeInterface* analysis toolbox. The remaining channels were time-shifted and median-subtracted across channels and time. Corrected channels were then spike-sorted with the *Kilosort 2.5* software and manually curated using *phy*. All sorted spiking and local field potential data were analyzed using custom *Matlab* code (2020b, MathWorks).

To identify strong oscillatory events in some recordings, we analyzed the LFP signal at a cortical depth of 500 μm and computed the spectrogram with a short-time Fourier transform and a moving window approach (2-s-long Hamming window with a step size of 0.1 s). We then calculated the mean spectral power between 0.5 and 10 Hz after subtracting the median power for each frequency over the entire session. The resulting trace was then z-scored, and we identified potential epileptiform events where the mean spectral power remained above 2 standard deviations for at least 10 s. This identified a total of eight putative epileptiform events between 18.2 and 39.5 s across all *Dnmt1* KO recordings, whereas no events were detected in WT mice. To ensure that electrophysiological results were not affected by this activity, we excluded all data during, as well as 15 s before and after each event, from all subsequent analyses.

Peri-event time histograms (PETHs) for sensory responses in each cluster were computed with a bin size of 10 ms and baseline-subtracted by the mean activity within 1 s before the laser onset (between 2 to 1 s before the first tactile stimulus). Trial-averaged local field potentials (LFPs) in response to sensory stimulation were similarly baseline-corrected. To compute current source densities (CSDs) from LFP signals, we used the inverse CSD method by Pettersen et al.[69]. We applied the spline iCSD method, assuming a smoothly varying CSD between electrode contacts, based on polynomial interpolation. We assumed a homogeneous, isotropic conductivity of $\sigma = 0.3$ S/m within and directly above the cortex[69]. To reduce spatial noise, the estimated CSD was subsequently convolved with a Gaussian spatial filter with a standard deviation of 0.1 mm. For LFP and CSD analyses, the signals from neighboring contacts at the same depth were averaged together to improve signal quality. After recordings, brains were prepared and cryo-sectioned coronally for control of electrode positioning. Images were acquired using an inverted fluorescence microscope (EVOS M5000, Thermo Fisher) using either a ×1.25 or a ×10 objective. For DiD imaging in the infrared range, a filter set with an excitation wavelength of 635/18 nm and an emission wavelength of 692/40 nm (Light Cube, Cy5) was used. For DAPI, a filter set with an excitation wavelength of 357/44 nm and an emission wavelength of 447/60 nm (Light Cube, DAPI 2.0) was applied.

## Evidence accumulation task

Head-fixed, water-deprived mice (8–12-week-old males, $N = 7$ Ctrl mice and $N = 6$ KO animals) were stepwise introduced to performing a series of uni- or multisensory evidence accumulation tasks in a custom-built setup. The stimulus presentation, lick detection, and reward delivery were driven by a microcontroller-based finite state machine running on a *Teensy* 3.2 (PJRC) and using custom *Python* code (*Python* 3.7; https://github.com/NabbefeldG/MS_task_gerion_V2_5). For all trials, the general scheme of events was similar: for 3 s, randomized sequences of up to 6 visual, tactile, or visuotactile stimuli were presented on one or both sides of the mouse. After a 0.5-s-long delay period, the animal had to respond within 2 s to obtain a reward. The ITI between trials was 3.5 s. Mice were first trained on reliably detecting visual stimuli, consisting of 6 visual gratings (spatial frequency = 0.018 cycles per degree) that were moving over the screen at a temporal frequency of 2 Hz. After mice learned to accurately respond on the side, where visual stimuli were presented, tactile stimuli were introduced in the form of 6 air puffs, as described above (see Neuropixels

Recordings). Air puffs occurred at regular intervals of 0.5 s during the 3-s stimulus period. After mice also accurately detected the side of tactile stimulation, visuotactile stimuli were introduced, where coherent visual and tactile stimuli were presented. Once mice achieved stable detection performances in all modalities, distractor stimuli of the same modality were introduced on the opposing side. While the number of stimuli in these trials varied to confront animals with different levels of difficulty, the general goal was for the mice to indicate the side with the higher number of presented stimuli.

## Morris water maze (MWM)

Visual navigation and learning capacities were investigated using a MWM. Mice (8–12-week-old males, $N = 7$ Ctrl. mice and $N = 6$ KO animals) were trained to search a translucent platform (Ø 10 cm) in a circular pool (Ø 1 m) containing tepid water. The pool was divided into four virtual quadrants, each of them equipped with distinct visual cues. Mice were tested on 6 consecutive days. Start positions for each trial were chosen at random. Mice remained on the platform for at least 5 s after finding it, to enable memorization of spatial cues. If a mouse did not find the platform within the given time limit, it was manually guided to the platform. In between trials, mice were kept underneath a red lamp for at least 10 min. Animals were habituated to the task on their first day of experiments by performing four trials for a maximum duration of 90 s each. The translucent platform slightly protruded from the water surface to facilitate quick locating. On the following 5 training days, the platform was submerged 1–2 cm below the water surface. All animals conducted 12 trials per day with three starts per quadrant, respectively. The maximum duration per trial was set to 60 s. Mice were recorded from above, and behavioral data were processed using *ANY-maze* (Stoelting Co., USA).

## Nest-building procedure

Nest-shredding and -building tests were performed with male mice between 12 and 24 weeks of age ($N = 16$ animals per genotype). Both *Sst-Cre* strains were tested by placing respective animals in separate housing cages with *ad libitum* access to food and water but without further enrichment except for fresh bedding and nest-building material (standardized autoclaved nestlets with a weight of 2–3 g (5 cm × 5 cm, Lexx, Germany)). For each nestlet, the weight was determined before the experiment, which took place during the murine active phase (5 p.m. to 9 a.m.). After testing, all mice were transferred to their home cages, and the percentage of shredded material as well as the corresponding nest-building score[117] were recorded. Mice were tested twice within one week, with two days between both trials. The resulting data were averaged for each mouse. The higher the nest-building score, the better the performance: Score 1 (nestlet > 90% intact), Score 2 (nestlet 50-90% intact and only partially torn), Score 3 (nestlet <50% intact but no distinct nest site, < 90% within a defined quarter of cage), Score 4 (nestlet <90% intact, identifiable but flat nest in a defined cage quarter, nest walls <50% of mouse's body height), and Score 5 (nestlet <90% intact, identifiable nest in a defined quarter of cage, roundish nest with walls covering > 50% of mouse's body height).

## Pentylenetetrazole (PTZ)-induced epileptic seizure protocol

Male *Sst-Cre/tdTomato* and *Sst-Cre/tdTomato/Dnmt1 LoxP²* mice (13–17-week-old, $N = 10$ animals per genotype) underwent an intraperitoneal administration of pentylenetetrazole (PTZ, Sigma-Aldrich, USA) with 10 mg/kg body weight. PTZ was diluted in sterile Ringer's solution (pH 7.3–7.4) to a concentration of 5 μg/μL beforehand. All mice received an intraperitoneal injection of PTZ (40 μL of PTZ solution per 20 g body weight) every ten minutes. All tested animals were monitored in transparent cages with no additional enrichment. Within every period of ten minutes, the occurrence of different epileptic events ranked from severity level 0 (no abnormalities) to the most severe score 7 (final tonic-clonic seizure with overstretched limbs and

tail) was detected. For each period of ten minutes, all detected severity scores were added up and compared between *Sst-Cre/tdTomato* and *Sst-Cre/tdTomato/Dnmt1 LoxP²* mice. Animals that depicted a final tonic-clonic seizure before the end of the experiment (a maximum of 12 injections and spending 150 min in the setup) were not included in the calculations after their final epileptic event (score 7). Severity levels and corresponding events: 1: immobilization by sudden arrest and lying flat on the ground for several seconds up to minutes. 2: facial twerking, intensive mouth movement, and/or teeth grinding without convulsions. 3: frequent head nodding and/or partial myoclonic events (mostly unilateral limb- or tail clonus). 4: continuous whole-body myoclonus indicated by a stretched back, as well as extensive and dynamic stiffening of the tail with overhead pointing towards the rostrum. 5: clonic seizure characterized by unilateral forelimb clonus and bending over until reaching a sitting/standing position. 6: clonic-tonic seizure with falling on one side, wild rushing, and jumping. 7: generalized tonic-clonic seizure indicated by wild rushing, jumping, and tonic extension culminating in overstretched limbs and tail. This epileptiform event results in respiratory depression and/or death (end of the experiment). After the PTZ protocol, all mice were sacrificed with an overdose of 5 vol% isoflurane (v/v), followed by cervical dislocation after reaching surgical tolerance, verified by loss of withdrawal response to toe pinch.

### Statistics and figure illustration
Statistical tests were performed using *Matlab*, *Python*, *R*, and *GraphPad Prism*. Unless otherwise stated, normally distributed data was tested applying unpaired and two-tailed Student's *t*-test and an unpaired Welch's *t*-test, nested two-way ANOVA, and two-way ANOVA followed by Bonferroni correction. For non-normally distributed data, we used a Wilcoxon *rank-sum* test. Respective statistical tests can be found in the main text and individual captions.

Unless otherwise stated, all depicted error bars represent the standard error of the mean (±SEM). Violin plots show the distribution of the numeric data using density curves, with the respective frequency of data points represented by the width. Straight lines represent the median and dotted lines the corresponding interquartile range. Single-value plots (XY-graphs) display all detected data points together with their mean and SEM used to illustrate analyses, including two-way nested ANOVA tests. Data derived from one specimen (embryo, adult mouse) are illustrated by different shapes of the respective data points and can therefore be distinguished from values derived from other animals. For boxplots, the central box represents the interquartile range (IQR), with the median denoted as a horizontal line within the box. Moreover, the 25th percentile (Q1) and the 75th percentile (Q3) of the data are shown. The whiskers extend to the data's minimum and maximum values.

### Reporting summary
Further information on research design is available in the Nature Portfolio Reporting Summary linked to this article.

## Data availability
The data supporting the results of this study are available as described in the results and the method part as hyperlinks or were directly uploaded to the corresponding data platforms, which are also stated, respectively. We confirm there is no (privacy) conflict in sharing our data openly. Since our study did not use human/patient material, there is no need to anonymize corresponding data to comply with ethical and legal standards. In addition to the original data generated during this research, secondary data used in the analysis are also referenced and can be accessed from the provided raw data overview, the mentioned data servers, or via hyperlinks stated in the study. For further information regarding data access or requests, please contact Geraldine Zimmer-Bensch.

The sequencing datasets generated and/or analyzed for this study have been deposited in the Gene Expression Omnibus repository with the following accession numbers: GSE276510 [ChIP-seq], GSE298336 [snRNA-seq], GSE298865 [Methyl-seq], GSE276516 [RNA-seq bT], and GSE300332 [RNA-seq ctx]. The MERSCOPE datasets have been deposited in the Gene Expression Omnibus repository with the accession number GSE298591 [MERFISH]. Single-cell RNA datasets from C57BL6/J embryos used for *CellChat*-based binding predictions are deposited under: GSE300648, GSE300649, and GSE291845 for the E14.5 dataset, and GSE291845, GSE300653 and GSE300678 for the E16.5 dataset. As this dataset is currently under consideration for publication in a different context, access is temporarily restricted and can be granted upon request by contacting Prof. Tanja Vogel (tanja.-vogel@anat.uni-freiburg.de). The MD simulation data have been deposited in the MDposit IRB node under the project number A022O. Neuropixels recording data are available in a figshare repository [https://doi.org/10.6084/m9.figshare.28282838].

## Code availability
Custom codes used in this study are available as described in the methods as hyperlinks uploaded to GitHub. We confirm there is no (privacy) conflict in sharing our codes openly. For further information regarding data access or requests, please contact Geraldine Zimmer-Bensch. The code for the single-cell RNA datasets of *CellChat*-based binding predictions on E14.5 and E16.5 C57BL6/J embryos is deposited here: https://github.com/Vogel-lab/Reichard-2024_DNMT1-Mediated_Regulation-of_IN_Migration. The analysis for the electrophysiological dataset is deposited here: https://github.com/musall/dnmt1SOM. The custom code for Evidence Accumulation Tasks is available under this address: https://github.com/NabbefeldG/MS_task_gerion_V2_5.

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

## Acknowledgements

This work was supported by the Sequencing Facility and the Flow Cytometry Facility, two core facilities of the Interdisciplinary Centre for Clinical Research (IZKF) Aachen within the Faculty of Medicine at RWTH Aachen University (DFG project number: 439895892). We thank Dr. Mira Jakovcevski, Hendro Langecker, Dorothee Hoffmann, Pia Döring, and Ananya Sangeetha for their experimental support, and Sandra Brill for her support with mouse care and genotyping. We further thank Prof. Dr. Christoph Kuppe, Emilia Scheidereit, and Vanessa Künstler for the access to and support with MERSCOPE. This research was funded by the Deutsche Forschungsgemeinschaft (DFG, German Research Foundation)—368482240/GRK2416, ZI-1224/13-1, ZI-1224/19-1, dedicated to G.Z.B, and 322977937/GRK2344 to C.L.F. and T.V.; the study was further funded under the Excellence Strategy of the Federal Government and the Länder (OPSF678, OPSF812, SFASIA002) dedicated to G.Z.B.; S.M. was supported by the Helmholtz association (VH-NG-1611) and the iBehave Network sponsored by the Ministry of Culture and Science of the State of North Rhine-Westphalia; S.X. acknowledges the support of the China Scholarship Council program (Project ID: 202306650006); K.Z. received financial support from the European Union-NextGenerationEU-Project (CUP F53D23001170006).

## Author contributions

J.R. performed experiments and data analysis, figure illustration, manuscript editing; P.W. performed experiments and data analysis, figure illustration, manuscript editing; S.X. performed the in silico experiments, data analysis, and figure illustration; K.Z. performed the in silico experiments and data analysis; C.L.F. performed experiments and data analysis; J.D. performed experiments and data analysis; S.G. performed experiments and data analysis; J.L. performed experiments and data analysis; C.B.Y. performed experiments and data analysis; G.P. performed experiments and data analysis; G.N. methods support and data analysis; L.D. performed experiments and data analysis; J.V. performed experiments and data analysis; L.B. performed experiments and data analysis; S.K. performed experiments and data analysis; M.S. performed experiments and data analysis, K.N.M. Data analysis, data discussion; C.C.K. data analysis; T.V. data discussion and analysis, manuscript editing; P.C. data interpretation and discussion, wrote the in silico analysis; S.M. data analysis and discussion, figure illustration, manuscript preparation; G.Z.B. conceptual design, administration and supervision, experimental design, wrote the manuscript, figure illustration, data analysis, interpretation and discussion.

## Funding

## Competing interests

The authors declare no competing interests.
