## [Transparent Peer Review file · Nature Communications]

DNMT1-Mediated Regulation of Somatostatin-positive Interneuron Migration Impacts Cortical Architecture and Function

Corresponding Author: Professor Geraldine Zimmer-Bensch

Version 0:

Reviewer comments:

Reviewer #1

(Remarks to the Author)

In this manuscript, Reichard et al. test the roles of Dnmt1 in Sst+ interneuron development and migration. They first show that Dnmt3a deletion causes Sst+ interneurons to exit prematurely from the superficial migratory stream. The neurons had perturbed gene expression signatures and migration patterns at the embryonic stage. The Dnmt1 deletion manifests in an altered cortical structure in adults displaying behavioral changes.

Overall, the manuscript is well-written and I feel the work is important. However, there are several issues in the manuscript listed below:

- (1) The authors exclusively study Sst+ neurons, thus throughout the text, the conclusions must be clear that the findings are only in Sst+ neurons. They should avoid making general statements. For example, the title should state "in Sst interneurons."
- (2) The changes in methylation in the conditional knockout model (Sst-Cre/tdTomato/ Dnmt1 loxP) were compared to Sst-Cre/tdTomato. A critical control will be to analyze the effects of Dnmt1 loxP itself on methylation. That is, are the methylation changes they detect because of Dnmt1 deletion in Sst+ interneurons or because of the presence of the loxP site in Dnmt1?
- (3) The authors nicely show that the genes upregulated upon Dnmt1 deletion in E14.5 Sst+ interneurons were downregulated in ESC-derived Dnmt1 tet/tet neurons that overexpressed Dnmt1 (Fig 1e and Supplementary Tables 6,7). However, my understanding is that these ESC-derived neurons are not Sst interneurons thus I am not sure why they chose to make these? Or is this a published dataset? If so the work must be cited. For a more rigorous comparison, the authors should consider overexpressing Dnmt1 in Sst+ interneurons.
- (4) The authors show that adult Sst-Cre/TdTomato/Dnmt1 KO mice display functional and behavioral abnormalities. They conclude that this is because of deletion of Dnmt1 impairs the development of Sst+ interneurons. Another possibility is that Dnmt1 is important for Sst+ interneurons in the adult brain as well. A key experiment will be to look at the methylation and gene expression signatures in Sst+ Dnmt1 KO interneurons.
- (5) The single cell RNA-seq presented in Fig. 4 is interesting. However, because only the Wildtype cortex is analyzed the results are very correlative. The point will be much strengthened by comparing Sst+/Dnmt1 KO and controls.

Reviewer #2

(Remarks to the Author)

This study from Reichard et al. uses a cell type-specific loss of function manipulation in mice to investigate the role of DNMT1 in regulating the migration of SST cortical interneurons (INs), and its impact on cortical anatomy, network activity,

and behavior. Although DMT1 has been known to regulate gene expression programs for INs originating in the preoptic area during cortical development, it is unclear if it plays similar roles for MGE-derived INs. The authors use a wide array of different techniques: RNAseq, histology, immunocytochemistry, in vivo Neuropixels recordings, and two behavioral assays. Based on their findings, the authors conclude that epigenetic mechanisms controlled by DMT1 influence, albeit transiently, IN migration into the cortex, but also permanently change the thickness of certain cortical layers, as well as weaker sensory evoked responses of cortical neurons and differences in nesting behavior and in PTZ-induced seizures in adults. There were no serious concerns with the methods or analyses.

Main criticism: Altogether the results constitute an interesting collection of different phenotypes (transcriptional, cellular, circuit, and behavioral) observed after *Dnmt1* deletion in SST INs in mice. But these different phenotypes seem presently isolated, as nothing links them to one another. In particular, the authors do not mechanistically link the neuroanatomical differences, which are transient or subtle (Figs. 2-5g), to the network differences in S1/V1 (Fig. 5h-q) or the behavioral deficits in adult animals (Fig. 6). It is also hard to explain how the adult behavioral deficits would be linked to an alteration of SST INs, when the differences in SST IN migration are only transient. The authors try to explain the differences in gamma, whisker-evoked activity, on the basis of a lack of SST-feedforward inhibition, but this is never pursued.

In that sense, the significance of the paper is limited by this lack of connection. In the end maybe there are 2 separate stories here, one about cortical IN migration and lamination that is fairly well developed, and another one about circuit and behavioral phenotypes of these mice that seems disconnected and also underdeveloped. Considering how the authors mention *Mef2c* as being dysregulated in the SST-Cre;*Dnmt1* null mice, it might be interesting to pursue this as a potential mechanism, in light of how loss of *Mef2c* disrupts PV cell maturation (Allaway, 2021). This might be a way to connect the interesting data in Fig. 5a-c (higher proportion of SST neurons that co-express PV) with the rest of the paper in a mechanistic way.

Other concerns:

1. If cortical size is similar between WT and KO mice at E14.5, E16.5 and E18.5 (Suppl. Fig. 7 & 8), then isn't it likely that the "nuanced" differences in cortical thickness in Fig. 4m are not biologically meaningful? In other words, the authors have decided that a small difference in the thickness of Layer VI is important to show in main Fig. 4, but relegate to suppl. data the negative results that seem more important; why? Others might interpret things entirely differently and conclude that SST IN-specific knockdown of *Dnmt1* is NOT important for cortical maturation.
2. Neuropixels data: It feels like panels 5h-q should be its own separate figure. Why is visual stim data represented differently than whisker stim data? Why not compare mean spontaneous firing rates across layers (electrode depths)? (it would be nice to compare the results to S1 and V1). Did they catch any abnormal epileptiform discharges in LFP during recordings? Also, why not look at fast-spiking neurons based on spike width analysis, since this paper is about interneurons? There is no quantification of results in Fig. 5p-q.
3. Behavior analysis: The assays chosen are not specific to cortical areas they study. Why not use sensory discrimination tasks for S1 and V1, or novel object recognition tasks?

Smaller issues:

Methods:

- o I was surprised that they start looking at E14.5, because *Sst* is barely expressed at that age; how can they get enough Cre expression driven by the promoter?
- o Extended Fig. 1: At high magnification, the immunos appear to show a lack of expression of DNMT1 within SST INs in adult mice. This should be done at E14.5 as well, since so much of their data is from that age (see concern about previous bullet point). Also it would be better if they showed a Western blot of FACS sorted SST INs and then probe for DNMT1?

Fig. 1:

- o How come *Dnmt1* is not the top dysregulated gene in Fig. 1A?
- o In Fig. 1a, they should also show a list of the top 50 genes that are the most and least significantly dysregulated based on LogFold change (not p value)
- o In Fig. 1d there is 1 gene overlap between DEG down and DEG up, is that a mistake?
- o Were authors surprised that *Sst* is upregulated in *Dnmt1* null mice? If anything with higher expression of *Mef2c* you should have less SST neurons (based on Allaway et al. Nature 2021) and perhaps more PV (was *Pvalb* not upregulated?)
- o They later refer to higher expression of *Mef2c* (line 356-7); they should indicate which dot is *Mef2c* in Fig. 1a.

Fig. 2:

- o As stated by the authors in the introduction (lines 95-100), and in accordance with previous findings (Lim et al Nat Neurosci 2018), SST-IN have two migratory paths (MZ vs. SVZ) depending on expression of different transcription factors that are dysregulated in the SST-Cre;*Dnmt1*- mice (line 151). What was the rationale for only analyzing one of the 2 migratory routes for SST-IN. Are there differences in the SVZ route in the mutant animals?

Fig. 4:

- o If IN migration is fine after E16.5, how come TBR1+ neurons are not, but then they are at E18.5, but then they are not in adults?
- o Panels b-k: they should indicate the age of embryos for these images/quantifications
- o Panel l: green label that says Layer I-V should be layer II-V
- o Panels n & o: hard to tell the difference in size of dots (related to p values)

Fig. 5:

- o PV immuno is of poor quality especially in WT control, they should replace these panels
- o They should cite papers that originally showed co-expression of SST and PV in same INs (Nassar et al., 2015; PMC4424818)
- o They should show histological tissue sections confirming the location of electrode track within S1 or V1

Fig. 6:

- o Why was frequent jumping not quantified? Could it be spontaneous seizures?
- o The standard in the field for seizure assessment is to use a Racine scale
- o Was the latency to seizures different? what about the mean cumulative dose of PTZ to trigger a seizure?

I co-reviewed this manuscript with a trainee as part of the Nature Communications initiative to facilitate training in peer review and to provide appropriate recognition for Early Career Researchers who co-review manuscripts. However, I wrote the entire review.

Reviewer #3

(Remarks to the Author)

Reviewer #4

(Remarks to the Author)

Reichard et al. present in an interesting study entitled "DNMT1-Mediated Regulation of Inhibitory Interneuron Migration Impacts Cortical Architecture and Function" that elucidates the role of DNMT1 in cortical interneurons during development. The paper provides several key advancements: This study specifically examines the role of DNMT1 in somatostatin-positive (SST+) interneurons, providing a more targeted analysis compared to the broader interneuron populations discussed in other sources. The research describes that Dnmt1 deletion results SST+ interneurons premature exit from the superficial migratory stream, a specific finding not previously reported. A novel finding is that altered migration of DNMT1-deficient SST+ interneurons affects cortical progenitors non-cell autonomously, impacting the generation of deep-layer neurons. The study reveals that the effects on cortical progenitors are temporally restricted, with increased numbers of EOMES+ intermediate progenitor cells observed at embryonic day 14.5. This temporal specificity is a new insight. The paper provides some hints into the molecular mechanisms, including DNMT1's regulation of key genes like Erbb4, which influences the timing of the tangential to radial migration switch in interneurons. The research extends to examining the long-term consequences of embryonic DNMT1 deficiency on adult cortical architecture, linking early developmental changes to later structural alterations. These advancements offer a more detailed and specific understanding of DNMT1's role in interneuron development, particularly for SST+ interneurons, and its broader implications for cortical development and architecture.

The presented data supports the main claims of the paper. However, there are some points that would benefit from additional support, improved data presentation, and discussion by the authors:

Mechanistic details: The main function of DNMT1 is to maintain DNA methylation patterns during DNA replication and mitosis. Here the authors describe the effects of DNMT1 KO "in *post-mitotic* SST+ interneurons primarily originating in the MGE" (quote from line 123). Assuming that the KO affects only post-mitotic cells as stated by the authors, this means that the observed effects on cortical development cannot be the result of impaired maintenance methylation pattern. Thus, the authors implicitly propose a new secondary function of DNMT1. However, this proposed mechanism is never discussed or explained in detail, but only briefly hinted at in the introduction as vague "non-canonical interactions with histone modifications". While proving the exact mechanistic details of DNMT1 function in cortical interneurons is out of scope of this study, the authors should nonetheless devote a paragraph in the discussion section to propose a potential mechanism how DNMT1 can affect gene expression in post-mitotic cells. Is this proposed function carried out by DNMT1's methyltransferase domain, or is it completely independent from DNMT1's ability to methylate DNA? If the first option is true, why did the authors observe methylation changes in both directions when the enzyme's catalytic domain can only add but not remove methylation? If the second option is true, how did DNMT1 KO seemingly affect DNA methylation, and why did the authors decide to profile DNA methylation in the first place? These questions should be discussed and the authors should at least carefully speculate what the underlying mechanism might be.

Sample size and replication: In some experiments like those depicted in Fig. 5m-o, the sample size or number of biological replicates (i.e. mice) is not clearly stated. This makes it difficult to assess the statistical power and reproducibility of certain findings. Some tests appear to use cells (or sections) as the unit of replication and not individual mice (e.g. Fig. 5m-o, please clarify). We understand that this may be common practice in the field as complex experiments cannot be easily performed on large numbers of animals, but it should be apparent to the reader that the p-values reported in Fig. 5m-o are based on few animals. Please correct this for every statistical analysis.

Bioinformatic analysis: The bioinformatic analysis presented in Fig. 1 serves as a good foundation and reveals some of the molecular effects of DNMT1 KO. However, the data is sometimes presented in a manner that is difficult to grasp or slightly misleading:

In Fig. 1h, many candidate genes that are explicitly highlighted in the manuscript text, like Erbb4, appear to have ~0% methylation change. This is likely because the authors plotted methylation at promoters here, even though Fig. 1i suggests that DMRs tend to be located downstream of the TSS in the gene body. It might be more appropriate to plot methylation of

the closest DMR (or of intersecting DMRs) for each gene, instead of promoter methylation. Furthermore, there is no apparent anticorrelation between methylation change and expression change in Fig. 1i, even though this might be expected. Maybe this will change when other genomic regions are plotted or the analysis is improved, otherwise it needs to be discussed. In general, the analysis in Fig. 1 barely addresses the directionality of the observed methylation changes, i.e. gain or loss of methylation upon KO, and how this relates to expression change. For example, the authors write “349 of these 600 genes increased in expression upon Dnmt1 deletion, pointing to putative targets of DNMT1-dependent repressive DNA methylation in SST+ interneurons of control mice”. If this is true, there should be a tendency for DMRs associated with these genes to be more highly (not lowly) methylated in control compared to KO mice.

Fig. 1c adds little useful information, it would be more informative to plot DMRs instead of sites here, and maybe add labels to the top DMRs denoting the nearest genes (or genes intersecting the DMR).

The authors used DSS for DMR detection. This software is a bit outdated and benchmarks by Korthauer et al.

(<https://doi.org/10.1093/biostatistics/kxy007>) have shown that it reports many false positive DMRs. Of course the authors of the present study are not to blame for the shortcomings of DSS, but nonetheless some of the DMRs depicted in Fig. 1j don't look very convincing. In my personal experience, the software developed by Korthauer et al. (dmsseq) indeed produces more robust results by reporting overall fewer DMRs of higher quality; maybe using dmsseq instead of DSS will improve results and reveal a stronger correlation between methylation and gene expression change.

Minor points:

Please plot the individual data points on top of each violin plot, and in plots like Fig. 4c,e,g. For experiments with a nested design (e.g. multiple sections per animal) please also denote which points correspond to the same animal, for example by plotting a different color or symbol for each animal.

For some tests, the authors need to clarify whether the unit of replication is one animal or a cell/section etc, for

“Nested t test” is a term introduced by the Prism software and not common jargon. According to the Prism website this test is just a nested two-way anova, it would be clearer to use this term instead or explain in the methods.

Please clarify at which point the Cre enzyme is expressed in the transgenic mouse line.

The authors should mention the limitations of performing bulk RNA and methylome sequencing in comparison to single-cell studies. Cell type heterogeneity among sorted populations might impact the result seen in the bulk sequencing of the KO mice.

Since in the study you generated scRNAseq data, it would be useful to show the sequenced cells on a UMAP, in order to be more precise on which cell types are produced upon sorting of the dorsal telencephalon.

Please also report some quality metrics (coverage etc) for the bulk methylome data.

The authors report that Eph/ephrin genes are regulated by DNA methylation in cortical interneurons. The same observation was recently reported in adult neurogenesis of olfactory bulb interneurons (<https://doi.org/10.1038/s41586-024-07898-9>), mentioning this would strengthen the present paper by showing that the results are in line with previous research.

Figure 4. Please provide a scheme of the experimental layout for panels p-s. In addition, labelling the microphotographs of panels b-j with the corresponding developmental stage would make it easier to follow this figure.

“normally distributed data was tested applying unpaired and two-tailed Student's t-test including a subsequent Welch's correction”. Do the authors mean Welch's t-test?

Version 1:

Reviewer comments:

Reviewer #1

(Remarks to the Author)

The authors have satisfactorily addressed my comments and I am supportive of publication of this manuscript in Nature Communications.

Reviewer #2

(Remarks to the Author)

The authors have responded to my critique thoroughly. They have addressed all the comments that I considered “smaller” issues to my satisfaction. Although they didn't resolve my main concern and they did not do some of the experiments and analyses I requested (see below), I acknowledge that this paper includes an enormous amount of work I am sympathetic about its publication. My interpretation of all the data would have been different, but the readers will decide.

My main criticism had been that the paper lacked a mechanistic link between neuroanatomical differences (which were transient and/or subtle) and the changes in nesting behavior and PTZ-induced seizures in adult Dnmt1 cKO mice. I suggested that they pursue Mef2c based on promising results from RNAseq, but they did not do so. Instead, the authors have done a lot of other new experiments, including snRNAseq that hints at molecular dysregulation that could in theory explain the differences in cortical thickness.

I also asked that they examine the activity of fast-spiking neurons (PV cells) from their existing Neuropixels data, but instead they did new recordings from SST neurons using opto-tagging, and, surprisingly, they found no differences in cKO mice. The cKO mice also show normal behavior in a visuo-tactile evidence accumulation task or on the Morris water maze. The authors did find bouts of epileptiform activity that are accompanied by behavioral changes (midriasis and excess salivation), but there was no manipulation (opto-/chemogenetics) to indicate that changes in network activity (Fig. 5) relate to these

seizure-like events. Because these events are neither infrequent (2X per hr) nor brief (30 s), and therefore it's possible that they could affect the ephys recordings, the authors should clarify whether such epochs were excluded from the neural activity analyses (Fig. 5G–L).

All of the new negative results (behavior, SST activity) and much of the normal histology are relegated to supplementary figures. And yet, based on those results, a different lab might have concluded that Dnmt1 knockdown has no overt phenotype. It would be good for the authors to at least acknowledge this caveat or alternate interpretation of their results in the Discussion.

Perhaps the bigger conundrum is to explain how deleting this critical gene from SST neurons, which eventually does not affect these interneurons, somehow causes problems to other neurons in the superficial cortical layers. 'Non-cell-autonomous' sounds cool, but it's hard to wrap my head around it when there are no cell-type-specific phenotypes to speak of. The authors speculate on possible theories, but it's mostly mental gymnastics. Since SST-INs are known to fine-tune network scaffolding and activity in both infragranular and supragranular layers (PMID: 27225074), perhaps the authors can speculate on why these non-cell-autonomous changes are restricted to superficial cortical layers, given inside-out cortical development and how deep layer SST-IN can regulate superficial layer connectivity (PMID: 38926335, 26844832).

Again, this paper represents an impressive effort and the results, if not definitive, are certainly interesting. The authors have gone to great lengths to improve the manuscript based on reviewer feedback.

Reviewer #3

(Remarks to the Author)

Reviewer #4

(Remarks to the Author)

We thank the authors for incorporating our comments and applying the necessary changes to the manuscript, which has greatly improved from the original version. However, the following comments need to be addressed by the authors.

In Figure 1i, the color code for the scale is missing

The authors mention that they discuss possible compensatory effects of Dnmt3 attributing to the lack of anti-correlation between DNA methylation and transcription. If that's the case, then the authors should definitely tone down the claim of the crucial role of Dnmt1 in the functional changes reported in this study.

In the rebuttal letter, the authors also discuss that DNA methylation does not always mean more transcription, and that the biological relevance of DNA methylation extends beyond transcription regulation. References to these statements are missing.

Please provide a scheme of the experimental layout for panels p-s: The experimental layout of this experiment, which is now Figure 4 c-f, is still missing

Reviewer #1 (Remarks to the Author):

In this manuscript, Reichard et al. test the roles of Dnmt1 in Sst+ interneuron development and migration. They first show that Dnmt3a deletion causes Sst+ interneurons to exit prematurely from the superficial migratory stream. The neurons had perturbed gene expression signatures and migration patterns at the embryonic stage. The Dnmt1 deletion manifests in an altered cortical structure in adults displaying behavioral changes.

Overall, the manuscript is well-written and I feel the work is important. However, there are several issues in the manuscript listed below:

(1) The authors exclusively study Sst+ neurons, thus throughout the text, the conclusions must be clear that the findings are only in Sst+ neurons. They should avoid making general statements. For example, the title should state “in Sst interneurons.”

We totally agree with the reviewer, and the title has changed, and we revised the text to remove general conclusions.

(2) The changes in methylation in the conditional knockout model (Sst-Cre/tdTomato/ Dnmt1 loxP) were compared to Sst-Cre/tdTomato. A critical control will be to analyze the effects of Dnmt1 loxP itself on methylation. That is, are the methylation changes they detect because of Dnmt1 deletion in Sst+ interneurons or because of the presence of the loxP site in Dnmt1?

We thank the reviewer for raising this important point. The loxP sites in this well-characterized Dnmt1 floxed model are positioned within introns between exons 3-4 and 5-6 and should not interfere with Dnmt1 function independently of Cre-mediated deletion. If loxP insertion significantly impaired DNMT1 activity, the floxed line would exhibit a hypomorphic or null-like phenotype, such as severe cortical degradation (as seen in Dnmt1 hypomorphs; Hutnick et al.¹) or embryonic lethality². However, we and others did not observe such abnormalities for this line³⁻⁵.

The reviewer may refer to Cre-dependent DNA methylation at loxP sites, which has been linked to gene silencing in some contexts^{6,7}. However, this phenomenon is primarily described in meiotic cells and plants and is highly context-dependent^{6,7}. While such an effect could theoretically silence Dnmt1, resulting in a phenotype similar to DNMT1 deletion, it could also propagate to nearby regions, potentially confounding gene expression changes^{6,7}.

However, several lines of evidence argue against the loxP sites being responsible for the methylation changes and phenotypes observed in our study:

- DMRs are not enriched on chromosome 9, where Dnmt1 is located, but are evenly distributed across the genome (Supplementary Table 2), arguing against a local loxP-driven methylation effect.
- The Dnmt1 floxed mouse line is well-documented in the literature (Refs) with no prior reports of non-specific methylation effects.
- If loxP sites impaired Dnmt1 function, the floxed line would exhibit severe phenotypic abnormalities—yet it shows normal health, mating behavior, and brain architecture (as required for breeding and housing approvals).
- Genes affected by Dnmt1 deletion in SST interneurons significantly overlap with those altered by DNMT1 overexpression in ESC-derived neurons, reinforcing a DNMT1-specific mechanism.

Many differentially expressed and methylated genes align with the premature exit phenotype from the superficial migratory stream, further supporting DNMT1's role in migration regulation. Investigating potential loxP-specific methylation effects in isolation would require complex and costly experiments. Moreover, as these putative LoxP effects are reported to be Cre-dependent, distinguishing direct effects of DNMT1 deletion from Cre-mediated loxP methylation would necessitate FACS-sorted methylation analysis of SST-Cre/TdTomato/Dnmt1 floxed cells vs. SST-Cre/TdTomato controls—a comparison we have already performed.

Thus, based on these considerations, we are confident that the observed methylation changes and phenotypes arise from DNMT1 deletion rather than non-specific loxP effects.

(3) The authors nicely show that the genes upregulated upon Dnmt1 deletion in E14.5 Sst+ interneurons were downregulated in ESC-derived Dnmt1 tet/tet neurons that overexpressed Dnmt1 (Fig 1e and Supplementary Tables 6,7). However, my understanding is that these ESC-derived neurons are not Sst interneurons thus I am not sure why they chose to make these? Or is this a published dataset? If so the work must be cited. For a more rigorous comparison, the authors should consider overexpressing Dnmt1 in Sst+ interneurons.

We thank the reviewer for this important comment. Indeed, the overexpression data were a published dataset. We had included the reference in the method part, which we now also state in the results description for clarity in the revised version. Although, this dataset stems from ESC-derived neurons with mixed identity and not SST-neurons, we chose to include the comparison with the DEG of Dnmt1 KO SST-cells to illustrate that DNMT1 regulates "common" targets across neuronal subsets. Moreover, these data strengthen the evidence that the transcriptional changes observed in the KO mouse model are indeed a consequence of Dnmt1 deletion, rather than an effect of the loxP sites (as discussed above).

We hope for understanding that overexpression of DNMT1 in ESC-derived SST neurons is beyond the scope of this study. There are existing protocols to drive ESCs into cIN progenitors and enrich SST, requiring a Nkx2.1::mCherry:Lhx6::GFP dual reporter mESC line⁸. However, we don't have the expertise and resources to address this aspect in a reasonable time. Our methodological focus rather lies on mouse work, however, overexpression of DNMT1 in mice is a challenge having been discussed in other manuscripts^{9,10}.

(4) The authors show that adult Sst-Cre/TdTomato/Dnmt1 KO mice display functional and behavioral abnormalities. They conclude that this is because of deletion of Dnmt1 impairs the development of Sst+ interneurons. Another possibility is that Dnmt1 is important for Sst+ interneurons in the adult brain as well. A key experiment will be to look at the methylation and gene expression signatures in Sst+ Dnmt1 KO interneurons.

To address this important point, we have improved our analysis and included new experimental data to further strengthen the link between the embryonic phenotype and resulting cortical architectural changes with the functional abnormalities. Our new electrophysiological analysis shows that the functional impairments were most pronounced in the superficial cortical layers, matching the reduced layer thickness and increased SST+ cIN density. Moreover, we added new experimental results, where we used cell-type-specific optogenetic stimulation to selectively isolate SST+ cINs in our recordings and compare their function between KO mice and controls. In line with the assumption that functional abnormalities are not primarily due to impaired SST+ cIN function in adults, we found no significant differences across several functional metrics, such as the action potential waveform, firing rate, or the ability of SST+ cINs to suppress other cortical neurons. These results are now shown in our new Extended Data Fig. 11.

Together, these new findings suggest that the functional disruption of cortical activity is primarily linked to structural changes that likely arise from embryonic changes rather than altered SST+ cIN function in adult mice. However, we now also discuss that further studying

the methylation and gene expression signatures of SST⁺ cINs in adult inducible *Sst-Cre/tdTomato/Dnmt1-loxP²* mice would be an important future direction, to distinguish embryonic from adult effects. Given the significant mouse and sequencing costs involved, and the timeline required for establishing a new (inducible) Cre-Line, we ask for understanding that this is beyond the scope of our current study. Especially as we had to focus on several other points already (new scRNA seq and RNA bulk seq experiments as well as additional behavioral studies, see below).

(5) The single cell RNA-seq presented in Fig. 4 is interesting. However, because only the Wildtype cortex is analyzed the results are very correlative. The point will be much strengthened by comparing *Sst*+/*Dnmt1* KO and controls.

We thank the reviewer for this important remark, we followed the suggestion and conducted scRNA sequencing experiments of the E16.5 cortex of *SST-Cre/tdTomato/Dnmt1 floxed* and *SST-Cre/tdTomato* control embryos, and included the results in the revised manuscript. The findings we obtained underlined our hypothesis that the cortical progenitors and their output is altered in *SSt-Cre/Dnmt1* KO embryos and strengthen the altered interactions between MGE cINs and RGCs in *Dnmt1* KO embryos.

Reviewer #2 (Remarks to the Author):

This study from Reichard et al. uses a cell type-specific loss of function manipulation in mice to investigate the role of DNMT1 in regulating the migration of SST cortical interneurons (INs), and its impact on cortical anatomy, network activity, and behavior. Although DMT1 has been known to regulate gene expression programs for INs originating in the preoptic area during cortical development, it is unclear if it plays similar roles for MGE-derived INs. The authors use a wide array of different techniques: RNAseq, histology, immunocytochemistry, in vivo Neuropixels recordings, and two behavioral assays. Based on their findings, the authors conclude that epigenetic mechanisms controlled by DMT1 influence, albeit transiently, IN migration into the cortex, but also permanently change the thickness of certain cortical layers, as well as weaker sensory evoked responses of cortical neurons and differences in nesting behavior and in PTZ-induced seizures in adults. There were no serious concerns with the methods or analyses.

Main criticism: Altogether the results constitute an interesting collection of different phenotypes (transcriptional, cellular, circuit, and behavioral) observed after *Dnmt1* deletion in SST INs in mice. But these different phenotypes seem presently isolated, as nothing links them to one another. In particular, the authors do not mechanistically link the neuroanatomical differences, which are transient or subtle (Figs. 2-5g), to the network differences in S1/V1 (Fig. 5h-q) or the behavioral deficits in adult animals (Fig. 6). It is also hard to explain how the adult behavioral deficits would be linked to an alteration of SST INs, when the differences in SST IN migration are only transient. The authors try to explain the differences in gamma, whisker-evoked activity, on the basis of a lack of SST-feedforward inhibition, but this is never pursued.

In that sense, the significance of the paper is limited by this lack of connection. In the end maybe there are 2 separate stories here, one about cortical IN migration and lamination that is fairly well developed, and another one about circuit and behavioral phenotypes of these mice that seems disconnected and also underdeveloped.

We appreciate the reviewer's valid point and largely agree with the concern regarding the need to better connect the embryonic defects with the functional phenotypes in adults. While the effect of *Dnmt1* deletion on SST interneuron migration appears transient, we would like to

emphasize that the non-cell-autonomous effects on cortical progenitors and the associated changes in neuronal output result in persistent changes, manifesting in the altered cortical layer thicknesses, which alters SST interneuron densities in the upper layers.

To strengthen the link between these structural changes and functional impairments in adult mice, we have performed additional electrophysiological analyses that demonstrate that functional impairments are also most pronounced in the superficial cortical layers. These impairments correlate with increased SST+ interneuron densities, which result from reduced layer thickness - a consequence of embryonic defects in progenitor output and interneuron migration. This link is further supported by our single-cell RNA sequencing (scRNA-seq) analysis of Dnmt1 KO and control embryos at E16.5, which highlights alterations in cell populations based on transcription profiles, being consistent with these observed alterations in cortical progenitors and their output, as well as the interaction between cINs and cortical progenitors (RGCs). In addition, we have added new experimental results, using targeted optogenetic stimulation to selectively isolate SST+ interneurons and compare their function between KO mice and controls. Here, we found that SST+ interneurons in KO mice showed no clear functional alterations compared to controls, suggesting that functional impairments are not directly due to disrupted SST+ interneuron function in adult mice but rather a consequence of structural alterations that occurred as a consequence of changes in neurodevelopment.

We have also clarified and extended our analysis on gamma oscillations. In our previous version we showed that visual stimulation induces gamma oscillations in V1 of control but not KO mice. In our refined version of the manuscript, we now provide a better quantification of this effect and show that changes in oscillatory power also differ across cortical layers. Moreover, we now show that the LFP power is generally reduced in both S1 and V1, suggesting a general disruption in oscillatory activity across cortical areas (Fig. 6j-l). We also agree with the reviewer that we cannot directly test if these changes are due to an alteration of SST-feedforward inhibition and have therefore adjusted the text to point out other potential explanation, such as structural changes in cortical circuits that might disrupt oscillatory activity. Lastly, to better link our functional results to the behavioral abnormalities, we further analyzed the video data when mice were running on the wheel during electrophysiological recordings. We found that KO mice exhibited abnormal eye movements, characterized by brief episodes of pronounced pupil dilation (over twice the normal size) that lasted 30–60 seconds and were often accompanied by facial spasms or salivation. These episodes coincided with significant low-frequency activity (1–4 Hz), indicative of epileptiform activity. Deeper cortical analysis revealed that this activity was particularly prominent in the superficial layers, further linking the architectural changes to functional and behavioral deficits.

Considering how the authors mention Mef2c as being dysregulate in the SST-Cre;Dnmt1 null mice, it might be interesting to pursue this as a potential mechanism, in light of how loss of Mef2c disrupts PV cell maturation (Allaway, 2021). This might be a way to connect the interesting data in Fig. 5a-c (higher proportion of SST neurons that co-express PV) with the rest of the paper in a mechanistic way. We thank the reviewer for this remark, and we agree that the increased Mef2c expression we see at E14.5 would be an interesting route to connect with the finding that in adulthood, as some SST-Cre cells acquire PV marker expression, proposing an altered developmental trajectory. Yet, to further strengthen this connection, time, mouse and cost-consuming ChIP seq or Cut&Tag and ATAC Seq experiments would need to be performed at distinct developmental stages that go far beyond the scope of this study. We believe, that with the deeper electrophysiological characterization we now present (described above), we could make a better link between the embryonic findings and the adult phenotype. However, we will follow this route in future studies using Gad-Cre/tdTom/Dnmt1 loxP mice, where we also found a shift in proportion towards higher PV cell numbers (data not shown). We further have shifted

the PV Immunostainings into the extended data figure 10, the present that interesting observation without overstatement.

Other

concerns:

1. If cortical size is similar between WT and KO mice at E14.5, E16.5 and E18.5 (Suppl. Fig. 7 & 8), then isn't it likely that the "nuanced" differences in cortical thickness in Fig. 4m are not biologically meaningful? In other words, the authors have decided that a small difference in the thickness of Layer VI is important to show in main Fig. 4, but relegate to suppl. data the negative results that seem more important; why? Others might interpret things entirely differently and conclude that SST IN-specific knockdown of Dnmt1 is NOT important for cortical maturation.

We agree with your observation. In the original figures, we did not provide data on the overall radial extension. We have now addressed this in the revised version, where we included these measurements and demonstrate that the radial extension is indeed increased. However, the lateral-to-medial extension does not show a significant increase, although there is a trend in this direction. In our opinion, this does not present a major issue. If all dimensions were uniformly increased, one might argue that these differences could be attributed to slight variations in developmental staging. Timed matings and Tyler staging of embryos inherently allow for a degree of variability, including minor age shifts due to the number of embryos and their position within the uterine horn. To account for this, we carefully measured all possible dimensions to ensure accurate comparisons.

2. Neuropixels data: It feels like panels 5h-q should be its own separate figure. Why is visual stim data represented differently than whisker stim data? Why not compare mean spontaneous firing rates across layers (electrode depths)? (it would be nice to compare the results fo S1 and V1).

The main reason why the structural and functional data were shown together is to demonstrate that the observed changes in layer thickness and cellular density are reflected in functional changes in neural network activity in adult mice. We agree with the reviewer that this link between structure and function was not clear enough in the previous figure and have therefore improved the analysis and reduced the total number of figure panels. We feel that the histology panel showing the layer shift now fits to the functional data (with the PV immunostaining now being shifted into the extended data), collected in the new Figure 5.

The reason why the visual and whisker stim data are shown differently is because we were asking two separate questions for these recordings: the S1 recordings with whisker stimulation were performed orthogonal to the brain surface to obtain an accurate measure of sensory responses across all cortical layers and relate this functional data to our earlier structural results in the same figure. This is now also visualized in our new Fig. 5. We found that whisker stim responses are weaker in KO mice versus controls, suggesting that the observed structural changes translate into disrupted neural network activity. Importantly, we found the most pronounced structural changes in the superficial cortical layers, which was also reflected in the functional data where whisker stim responses were particularly disrupted in the superficial layers. We now demonstrate this important point in our new Fig. 5f-i where we show that superficial responses are particularly reduced in KO mice, both for the population current source density as well as individual neurons (Fig. 5). Following the reviewers' suggestions, we also quantified the spontaneous firing rates across layers and found a general reduction in firing rates in KO mice versus controls. Supporting our earlier results, this difference was particularly pronounced in the superficial cortical layers. This result is shown in our new Extended Data Fig. 11.

The reason for including the visual results was because it is known that high-contrast visual stimulation induces gamma oscillations in V1. This allowed us to further study if other network functions, such network oscillations, are also disrupted in KO mice. However, for practical reasons, the Neuropixels probe in V1 was inserted at a very shallow angle, making a precise

mapping from recording depth to different cortical layers difficult. Since the visual stimuli were also much longer than the whisker stimulation, the sensory responses in V1 and S1 are therefore not directly comparable. To clarify this conceptual distinction, we have therefore improved the writing and figure structure to make it clear that the S1 results show a layer-specific functional disruption that matches our structural results while the V1 recordings show specific disruptions in cortical network oscillations.

To make the transition from S1 to V1 clearer, we now also show that the oscillatory power of local field potentials across all electrodes is reduced in both areas for KO mice (Fig. 5k) and now clearly state that visual stimulation in V1 allowed us to further investigate changes in stimulus-induced gamma oscillations. Interestingly, despite the caveat with the probe insertion angle, we found that stimulus-induced low-frequency oscillations were increased in the superficial cortex of KO mice (Fig. 5l). This is highly unusual for stimulus-induced oscillations and further points to a particular disruption of neural activity in the superficial cortex.

Did they catch any abnormal epileptiform discharges in LFP during recordings?

We thank the reviewer for this valuable question. Upon further investigating the data, we indeed found brief episodes with abnormal epileptiform discharges in KO mice, marked by strong increases in low-frequency local field potentials. These episodes were also accompanied by large pupil dilations, changes in body posture and salivation. We now show an example from S1 in our new Fig. 6a-c to further support our behavioral results that are shown in the same figure.

The episodes lasted for about 30 seconds and only occurred very rarely (about 2 times per hour in some of the recordings for both mice). Individual deflections were stronger in superficial versus deep cortical layers (Fig. 6c), matching our earlier results. However, outside of these short episodes we found no interictal spikes or other signs of epileptiform activity, suggesting that epileptiform discharges may have also invaded cortex from other brain regions.

Also, why not look at fast-spiking neurons based on spike width analysis, since this paper is about interneurons?

Since SST interneurons are not fast-spiking, we were unable to further investigate their function based on spike width analysis alone. However, we had also expressed channelrhodopsin-2 in these neurons, allowing us to instead identify SST neurons based on a clear increase in spiking rate upon optogenetic light stimulation.

We therefore performed additional analyses to isolate SST interneurons in KO and control mice and compared their spike waveform and duration as well as their spontaneous firing rates. We found no significant differences in either metric, suggesting that action potential generation and general function of SST interneurons was not clearly disrupted in KO mice. Moreover, we quantified how strongly other cortical neurons were inhibited when stimulating SST interneurons and also found no significant differences.

Together, these results suggest that SST interneuron function was still intact in KO mice and the observed functional changes are therefore more likely due to structural changes in cortical layers and network architecture. We now show these new results in our extended data Fig. 11c-e and also discuss them in the main text.

There is no quantification of results in Fig. 5p-q.

This is now addressed in our new Fig. 5k.

3. Behavior analysis: The assays chosen are not specific to cortical areas they study. Why not use sensory discrimination tasks for S1 and V1, or novel object recognition tasks? We thank the reviewer for this remark and now include additional data from several behavioral tests to further probe for potential behavioral changes. As suggested by the reviewer, we performed additional behavioral experiments and trained KO and control mice in a visuo-tactile evidence accumulation task (Extended data Figure 12). Surprisingly, we found no clear differences in learning speed or visuo-tactile discrimination performance, suggesting that the functional alterations in KO mice do not lead to a strong perceptual impairment. A potential reason for this could be that KO mice compensate for cortical impairments by instead relying on subcortical sensory processing pathways. However, future studies with functional recordings in task-performing mice will be needed to fully address this question. Furthermore, we tested the learning capacities of KO mice by using the Morris Water Maze but also found largely similar results for both genotypes (Extended data Figure 12).

The strongest link to behavioral abnormalities were the strong pupil dilations that correlated with a marked increase in low frequency activity, indicating epileptiform activity in the *Dnmt1* KO mice. This is also in line with the PTZ data (Figure 6). Together with the nest building assay our new results therefore point to behavioral abnormalities related to neuropsychiatric diseases rather than to defects in sensory perception. Unfortunately, we could not conduct the novel recognition task, as we had no license for this task and the process of getting such licenses takes a very long time in Germany.

Smaller

issues:

Methods:

o I was surprised that they start looking at E14.5, because Sst is barely expressed at that age; how can they get enough Cre expression driven by the promoter?

At transcript level its prominently expressed at E14.5 at postmitotic level, as shown in various publications (e.g. Fig. 2B in Elbert et al., 2019¹¹, Munguba et al., 2023¹², Taniguchi et al., 2011¹³ and others¹⁴), explaining why also others have used the SST-Cre line to investigate different aspects cortical interneuron development¹². In line with these studies from other groups, we do see SST-Cre-dependent LoxP recombination. SST Expression is further confirmed by the (sc)RNA-seq datasets and Merescope analysis presented in the manuscript (Extended Data Figure E8).

o Extended Fig. 1: At high magnification, the immunos appear to show a lack of expression of DNMT1 within SST INs in adult mice. This should be done at E14.5 as well, since so much of their data is from that age (see concern about previous bullet point). Also it would be better if they showed a Western blot of FACS sorted SST INs and then probe for DNMT1?

We thank the reviewer for this remark. Apart from the RNA seq data showing reduced *Dnmt1* transcript levels in E14.5 FAC-sorted SST-Cre/*Dnmt1* KO cells, we performed immunostaining against DNMT1 at E14, further confirming the reduction in DNMT1 levels. As we only obtain between 5000-10000 SST-Cre/*tdTom* cells per hemisphere, Western Blot of FACS samples is not feasible, as high cell numbers are required.

Fig.

1:

o How come *Dnmt1* is not the top dysregulated gene in Fig. 1A?

This is a good point, which we explain as follows: Cre-mediated deletion begins when SST+ cIN leave the MGE (being in line with other studies¹²⁻¹⁴). Although the exact degradation time of transcripts depends on various factors, it can be within the range of hours¹⁵, for which we might still detect *Dnmt1* transcripts being produced in the prior proliferative state. Moreover,

only exon 4 and 5 is deleted, for which transcripts are produced, which we also detect, however they lead to a dysfunctional protein being degraded¹⁶.

o In Fig. 1a, they should also show a list of the top 50 genes that are the most and least significantly dysregulated based on LogFold change (not p value)

We now show the Log2FC in the new heatmaps (Fig. 1i and Figure 2s).

o In Fig. 1d there is 1 gene overlap between DEG down and DEG up, is that a mistake?

This is not a mistake; indeed, one isoform was up- and the other downregulated.

o Were authors surprised that Sst is upregulated in Dnmt1 null mice? If anything with higher expression of Mef2c you should have less SST neurons (based on Allaway et al. Nature 2021) and perhaps more PV (was Pvalb not upregulated?)

We have extended our RNA seq dataset in the revised manuscript including additional cortical samples, where we did not see Sst-upregulation. In line with the increased Mef2C level, we further detected a higher proportion of SST⁺ cINs co-expressing PV in adult Dnmt1 KO mice. However, PV is not expressed at embryonic stages, so we cannot assess its level at E14.5 nor at E16.5. However, as the reviewer mentioned, Mef2C expression level drive PV fate and this transcript was significantly upregulated, along with Dlx5, and Maf and MafB, all influencing PV fate, as discussed in the revised manuscript.

o They later refer to higher expression of Mef2c (line 356-7); they should indicate which dot is Mef2c in Fig. 1a.

This is now indicated in the new Figure 1h.

Fig. 2:

o As stated by the authors in the introduction (lines 95-100), and in accordance with previous findings (Lim et al Nat Neurosci 2018), SST-IN have two migratory paths (MZ vs. SVZ) depending on expression of different transcription factors that are dysregulated in the SST-Cre;Dnmt1- mice (line 151). What was the rationale for only analyzing one of the 2 migratory routes for SST-IN. Are there differences in the SVZ route in the mutant animals?

Thank you for giving us the opportunity to clarify this issue. We indeed analyzed both migratory streams as shown in Fig. 2. However, we only found the MZ stream being affected by *Dnmt1* deletion.

Fig. 4:

o If IN migration is fine after E16.5, how come TBR1+ neurons are not, but then they are at E18.5, but then they are not in adults?

Dnmt1-deficient SST cINs show an altered migration pattern and e.g. elevated ephrinB2 expression levels at E14.5. Our scRNA seq data propose a likely interaction between efnB2 (expressed by SST cINs) with RGCs and IPCs (expressing EphA4) at E14.5 (Fig. 4) and E16.5 (Extended data Figure 8). Our vitro data indicate that efnB2 stimulation of cortical progenitors leads to increased generation of neurons and basal progenitors from nestin positive RGCs (Fig. 4). Cell cycle length for E13.5 cortical progenitors is about 10 hours¹⁷, and migration from the VZ to the cortex takes at least another 24 hours¹⁸ or even longer when generated indirectly through EOMES positive IPCs. Thus, we think it makes sense, that we see effects on EOMES+ IPCs at E14.5 and E16.5 and TBR1+ positive neurons in the cortical plate two days later. Also, the expansion of the TBR1+ positive layer 6 is still visible at E18 in Dnmt1 KO mice, being in line with the increased thickness of the deep layers observed in adults. So, we argue that transient alterations of cIN migration in combination with the altered expression of signaling cues such

as ephrinB2, have an impact on cortical progenitors and their neuronal output, becoming obvious 2 days later (due to the time neurogenesis and migration to the cortical plate takes).

o Panels b-k: they should indicate the age of embryos for these images/quantifications
We apologize for this omission, which we now corrected as suggested by the reviewer, we agree that this is necessary for more clarity.

o Panel l: green label that says Layer I-V should be layer II-V
We thank the reviewer for this remark and corrected this.

o Panels n & o: hard to tell the difference in size of dots (related to p values)
We now provided improved figures.

Fig. 5:
o PV immuno is of poor quality especially in WT control, they should replace these panels
We did replace that by improved panels.

o They should cite papers that originally showed co-expression of SST and PV in same INs (Nassar et al., 2015; PMC4424818)
We thank the reviewer for this advice and did so.

o They should show histological tissue sections confirming the location of electrode track within S1 or V1
We thank the reviewer for this remark, we have now implemented the histological sections in the Extended Data Fig. 11.

Fig. 6:
o Why was frequent jumping not quantified? Could it be spontaneous seizures?
The jumping and repetitive movements were observed during cleaning in the mouse facility and is difficult to quantify as it occurs occasionally in Dnmt1 KO but is absent in control mice. However, we also observed abnormal eye movement during the electrophysiological recordings on the running wheel, for KO mice, characterized by brief episodes of pronounced pupil dilation being accompanied by facial spasms or salivation. Here we had the opportunity to analyze the Neuropixels recordings during these episodes, and we determined coincided significant low-frequency activity (1–4 Hz), which is indicative of epileptiform activity. This finding, together with the PTZ results and the repetitive movement behavior are indicative of a seizure phenotype¹⁹.

o The standard in the field for seizure assessment is to use a Racine scale
We thank the reviewer for this remark and now applied the Racine scale in the revised version.

o Was the latency to seizures different? what about the mean cumulative dose of PTZ to trigger a seizure?
Generally, the animals do not differ with regard to the required dosages of PTZ, as the times until the first occurrence of the individual events are similar, and the necessary dosages do not differ. However, it should be noted that the KO are more severely affected (heat maps Figure 6f), as they have, in some cases, higher Racine scores (which take into account all events occurring every 10 minutes and sum the values).

I co-reviewed this manuscript with a trainee as part of the Nature Communications initiative to facilitate training in peer review and to provide appropriate recognition for Early Career Researchers who co-review manuscripts. However, I wrote the entire review.

Reviewer #3 (Remarks to the Author):

Reviewer #4 (Remarks to the Author):

Reichard et al. present in an interesting study entitled "DNMT1-Mediated Regulation of Inhibitory Interneuron Migration Impacts Cortical Architecture and Function" that elucidates the role of DNMT1 in cortical interneurons during development. The paper provides several key advancements: This study specifically examines the role of DNMT1 in somatostatin-positive (SST+) interneurons, providing a more targeted analysis compared to the broader interneuron populations discussed in other sources. The research describes that Dnmt1 deletion results SST+ interneurons premature exit from the superficial migratory stream, a specific finding not previously reported. A novel finding is that altered migration of DNMT1-deficient SST+ interneurons affects cortical progenitors non-cell autonomously, impacting the generation of deep-layer neurons. The study reveals that the effects on cortical progenitors are temporally restricted, with increased numbers of EOMES+ intermediate progenitor cells observed at embryonic day 14.5. This temporal specificity is a new insight. The paper provides some hints into the molecular mechanisms, including DNMT1's regulation of key genes like Erbb4, which influences the timing of the tangential to radial migration switch in interneurons. The research extends to examining the long-term consequences of embryonic DNMT1 deficiency on adult cortical architecture, linking early developmental changes to later structural alterations. These advancements offer a more detailed and specific understanding of DNMT1's role in interneuron development, particularly for SST+ interneurons, and its broader implications for cortical development and architecture.

The presented data supports the main claims of the paper. However, there are some points that would benefit from additional support, improved data presentation, and discussion by the authors:

Mechanistic details: The main function of DNMT1 is to maintain DNA methylation patterns during DNA replication and mitosis. Here the authors describe the effects of DNMT1 KO "in *post-mitotic* SST+ interneurons primarily originating in the MGE" (quote from line 123). Assuming that the KO affects only post-mitotic cells as stated by the authors, this means that the observed effects on cortical development cannot be the result of impaired maintenance methylation pattern. Thus, the authors implicitly propose a new secondary function of DNMT1. However, this proposed mechanism is never discussed or explained in detail, but only briefly hinted at in the introduction as vague "non-canonical interactions with histone modifications". While proving the exact mechanistic details of DNMT1 function in cortical interneurons is out of scope of this study, the authors should nonetheless devote a paragraph in the discussion section to propose a potential mechanism how DNMT1 can affect gene expression in post-mitotic cells. Is this proposed function carried out by DNMT1's methyltransferase domain, or is it completely independent from DNMT1's ability to methylate DNA? If the first option is true, why did the authors observe methylation changes in both directions when the enzyme's catalytic domain can only add but not remove methylation? If the second option is true, how did DNMT1 KO seemingly affect DNA methylation, and why did the authors decide to profile DNA methylation in the first place? These questions should be discussed and the authors should at least carefully speculate what the underlying mechanism might be.

We thank the reviewer for that thoughtful and very important remark.

DNMT1 has long been viewed as a maintenance methyltransferase due to weak binding to unmethylated DNA observed in crystal structures²⁰. However, recent findings, including ours, demonstrate DNMT1 expression in and point to DNA methylation-dependent functions even in adult neurons^{1,3,21}. Most studies investigate the function by knockout approaches and then measuring changes in expression and methylation. Changes in DNA methylation patterns upon DNMT1 knockout in post-mitotic neurons have been observed frequently, indicating that DNMT1 influences methylation dynamics even in non-dividing cells. In support of this, it was indeed reported that, DNMT1 binds unmethylated DNA with high affinity in solution ($K_d = 2.3 \pm 0.7$ nM;²² Table S1), which is in contrast to the crystal structure findings²⁰.

To address this discrepancy, we conducted molecular dynamics simulations revealing that crystal lattice artifacts caused underestimation of binding in the 5' region, whereas in solution, this region tightly binds DNMT1's catalytic domain (Figs. 1B, Extended Data Figure 1 and supplementary data), which is now included in the revised manuscript. These results reconcile structural and functional data, supporting DNMT1's capacity to interact with unmethylated DNA, which challenges the "maintenance methyltransferase-only-view". This aligns with our Methyl-seq analysis, revealing differential methylation which correlates significantly with transcriptional changes.

In line with this, DNMT1 has been shown to interact with CFP1 (CysxxCys finger protein 1), which presents a high affinity for unmethylated DNA. CFP1 has been shown to interact with DNMT1 (via aa 169–493, TS and 970–1616)²³, and CFP1^{-/-} cells showed a reduction of 70% of DNA methylation²⁴. Moreover, a considerable de novo methylation activity for Dnmt1 at certain repetitive elements and single copy sequences has been detected²⁵. Functional cooperation of Dnmt1 during de novo methylation of DNA has been described²⁶. Gene-specific de novo methylation can be initiated by reintroduction of DNMT1 in cells lacking DNMT1 and DNMT3b, that display nearly absent genomic methylation²⁷. These observations strongly suggest a cooperation between DNMT1 and DNMT3s in de novo methylation activity, which in line with DNMT1 KO studies in postmitotic neurons from us and others.

In sum, DNMT1's presence and activity in post-mitotic neurons, along with its ability to bind unmethylated DNA double-strands, support the hypothesis that DNMT1 has functions beyond maintenance methylation during replication. These findings justify the investigation of DNA methylation changes in post-mitotic neurons upon DNMT1 deletion.

We included and discussed the new data and these aspects in the revised version of the manuscript.

Moreover, we also included the non-canonical functions in the discussion.

Sample size and replication: In some experiments like those depicted in Fig. 5m-o, the sample size or number of biological replicates (i.e. mice) is not clearly stated. This makes it difficult to assess the statistical power and reproducibility of certain findings. Some tests appear to use cells (or sections) as the unit of replication and not individual mice (e.g. Fig. 5m-o, please clarify). We understand that this may be common practice in the field as complex experiments cannot be easily performed on large numbers of animals, but it should be apparent to the reader that the p-values reported in Fig. 5m-o are based on few animals. Please correct this for every statistical analysis.

We thank the reviewer for this remark and corrected this throughout the manuscript.

Bioinformatic analysis: The bioinformatic analysis presented in Fig. 1 serves as a good foundation and reveals some of the molecular effects of DNMT1 KO. However, the data is sometimes presented in a manner that is difficult to grasp or slightly misleading: In Fig. 1h, many candidate genes that are explicitly highlighted in the manuscript text, like *ErbB4*, appear to have ~0% methylation change. This is likely because the authors plotted methylation at promoters here, even though Fig. 1i suggests that DMRs tend to be located downstream of the TSS in the gene body. It might be more appropriate to plot methylation of the closest DMR (or of intersecting DMRs) for each gene, instead of promoter methylation.

We followed the reviewer's advice and changed the data presentation in that matter. We also included in the discussion the aspect, that some transcriptionally dysregulated genes might arise from indirect or secondary effects, such as DNMT1-mediated changes in the expression of certain transcription factors such as ARX. This could trigger altered expression of ARX target genes.

Furthermore, there is no apparent anticorrelation between methylation change and expression change in Fig. 1i, even though this might be expected. Maybe this will change when other genomic regions are plotted or the analysis is improved, otherwise it needs to be discussed.

We discussed this issue in the new version of the manuscript. First, we cannot exclude compensatory effects by DNMT3A. Moreover, DNA methylation does not always lead to transcriptional silencing, and the DNA methylation dependent (dys)regulation of transcription can have secondary effects on transcription of genes (without a correlation with DNA methylation). It is included in the discussion.

In general, the analysis in Fig. 1 barely addresses the directionality of the observed methylation changes, i.e. gain or loss of methylation upon KO, and how this relates to expression change. For example, the authors write "349 of these 600 genes increased in expression upon *Dnmt1* deletion, pointing to putative targets of DNMT1-dependent repressive DNA methylation in SST+ interneurons of control mice". If this is true, there should be a tendency for DMRs associated with these genes to be more highly (not lowly) methylated in control compared to KO mice.

We have addressed this aspect by including a heatmap for DNA methylation changes for the most relevant key genes mentioned in the manuscript. Moreover, we provide the supplementary tables with detailed information.

Fig. 1c adds little useful information, it would be more informative to plot DMRs instead of sites here, and maybe add labels to the top DMRs denoting the nearest genes (or genes intersecting the DMR).

We agree and have removed this figure and included DMRs in the heatmap.

The authors used DSS for DMR detection. This software is a bit outdated and benchmarks by Korthauer et al. (<https://doi.org/10.1093/biostatistics/kxy007>) have shown that it reports many false positive DMRs. Of course, the authors of the present study are not to blame for the shortcomings of DSS, but nonetheless some of the DMRs depicted in Fig. 1j don't look very convincing. In my personal experience, the software developed by Korthauer et al. (*dmrseq*) indeed produces more robust results by reporting overall fewer DMRs of higher quality; maybe using *dmrseq* instead of DSS will improve results and reveal a stronger correlation between methylation and gene expression change.

We thank the reviewer for this advice, and we tried this out. However, there was no improvement compared to our DSS based analysis, and we think that there are biological valid reasons (as discussed above), that there are DEGs that do not correlate with DMRs and vice versa. Not every DMR must yield in expression changes, and DNA methylation does not always lead to transcriptional silencing, as it also can create binding motifs for transcription

factors (REFs). Also based on the literature, we think that the biological relevance of DNA methylation is by far not yet revealed completely and that functions other than gene repression are very likely. Further, compensatory DNMT3A actions can be likely in Dnmt1 KO cells, and DNA methylation dependent changes in expression of diverse transcription factors (that we have identified) can secondarily influence the expression of genes without a correlation with DNA methylation changes. Colleagues from the community reported similar observations (of many DEGs not correlating with DMRs) in personal communications.

Minor points:
Please plot the individual data points on top of each violin plot, and in plots like Fig. 4c,e,g.
We have changed that in the revised version, and showed the individual data points.

For experiments with a nested design (e.g. multiple sections per animal) please also denote which points correspond to the same animal, for example by plotting a different color or symbol for each animal.
We thank the reviewer for this remark and have used different symbols (being also explained in the legends).

For some tests, the authors need to clarify whether the unit of replication is one animal or a cell/section etc, for
“Nested t test” is a term introduced by the Prism software and not common jargon. According to the Prism website this test is just a nested two-way anova, it would be clearer to use this term instead or explain in the methods.
We thank the reviewers for this comment. For all nested analyses (stated in the respective Figure legends) we now state the correct name of the underlying test (nested two-way ANOVA) and explained that this is named “nested t-test” in the PRISM software in the methods section.

Please clarify at which point the Cre enzyme is expressed in the transgenic mouse line.
Cre is expressed at postmitotic levels as soon as the cells leave the MGE (e.g. Fig, Taniguchi et al., 2011¹³).

The authors should mention the limitations of performing bulk RNA and methylome sequencing in comparison to single-cell studies. Cell type heterogeneity among sorted populations might impact the result seen in the bulk sequencing of the KO mice.
We now have also included snRNA seq, with altered expression of relevant key genes being confirmed. Also, bulk RNA seq of FAC-sorted cells from the cortex reproduced relevant changes seen in the basal telencephalon. Still, we agree and mentioned the limitation in the discussion.

Since in the study you generated scRNAseq data, it would be useful to show the sequenced cells on a UMAP, in order to be more precise on which cell types are produced upon sorting of the dorsal telencephalon.
We agree and presented the UMAPs (Figure 4 and Extended Data Figure 8).

Please also report some quality metrics (coverage etc) for the bulk methylome data.
We included the quality metrics for the bulk sequencing in the respective Supplementary tables 1, 2 and 9.

The authors report that Eph/ephrin genes are regulated by DNA methylation in cortical interneurons. The same observation was recently reported in adult neurogenesis of olfactory

bulb interneurons (<https://doi.org/10.1038/s41586-024-07898-9>), mentioning this would strengthen the present paper by showing that the results are in line with previous research. We thank the reviewer for this valuable remark and included this aspect and reference in the discussion.

Figure 4. Please provide a scheme of the experimental layout for panels p-s. In addition, labelling the microphotographs of panels b-j with the corresponding developmental stage would make it easier to follow this figure.

We have followed the advice of the reviewer and included everything in the new Figure 3.

“normally distributed data was tested applying unpaired and two-tailed Student’s t-test including a subsequent Welch’s correction”. Do the authors mean Welch’s t-test?

Yes, and we corrected that.

First of all, we thank the reviewers for their appreciative and insightful comments, all of which we have carefully addressed in the revised manuscript.

In addition to responding to and addressing the reviewers' points, we took the opportunity to generate new methylome data with higher sequencing depth. This allowed us to validate the robustness of our findings with improved coverage and resolution. The results from this second dataset closely mirrored those of the original analysis and fully supported our initial conclusions. Given the superior quality and depth of the new dataset, we have chosen to present only this version in the revised manuscript.

REVIEWER COMMENTS

Reviewer #1 (Remarks to the Author):

The authors have satisfactorily addressed my comments and I am supportive of publication of this manuscript in Nature Communications.

We thank the reviewer for supporting our manuscript in its revised version

Reviewer #2 (Remarks to the Author):

The authors have responded to my critique thoroughly. They have addressed all the comments that I considered “smaller” issues to my satisfaction. Although they didn't resolve my main concern and they did not do some of the experiments and analyses I requested (see below), I acknowledge that this paper includes an enormous amount of work I am sympathetic about its publication. My interpretation of all the data would have been different, but the readers will decide.

My main criticism had been that the paper lacked a mechanistic link between neuroanatomical differences (which were transient and/or subtle) and the changes in nesting behavior and PTZ-induced seizures in adult Dmnt1 cKO mice. I suggested that they pursue Mef2c based on promising results from RNAseq, but they did not do so. Instead, the authors have done a lot of other new experiments, including snRNAseq that hints at molecular dysregulation that could in theory explain the differences in cortical thickness.

We very much agree that Mef2c is a promising route to follow, which however, as mentioned previously, is beyond the scope of this study. We further appreciate that the reviewer also acknowledges the other experiments we have conducted to support our theory of the mechanistic link between neuroanatomical differences and functional readouts.

I also asked that they examine the activity of fast-spiking neurons (PV cells) from their existing Neuropixels data, but instead they did new recordings from SST neurons using opto-tagging, and, surprisingly, they found no differences in cKO mice. The cKO mice also show normal behavior in a visuo-tactile evidence accumulation task or on the Morris water maze. The authors did find bouts of epileptiform activity that are accompanied by behavioral changes (midriasis and excess salivation), but there was no manipulation (opto-/chemogenetics) to indicate that changes in network activity (Fig. 5) relate to these seizure-like events. Because these events are neither infrequent (2X per hr) nor brief (30 s), and therefore it's possible that they could affect the ephys recordings, the authors should clarify whether such epochs were excluded from the neural activity analyses (Fig. 5G–L).

We agree with the reviewer that examining the activity of fast-spiking (FS) interneurons would be of interest. However, we found no clear differences when exploring separate cell types based on their waveform, in part because we did not see a clear clustering of FS in the cKO recordings. The absence of a distinct FS cluster might reflect a phenotype of the cKO mice but, due to the limitations of waveform-based classification in this dataset, we instead chose to focus on the opto-tagging approach, which provides a more definitive identification of specific interneuron types. While we agree that the lack of clear functional differences in cKO mice is surprising, we strongly believe that these results are more robust and therefore chose to include them as a strong addition to the manuscript.

We also agree that it is important to ensure that our electrophysiological results were not affected by seizure-like events and therefore performed a more careful analysis to isolate and remove these events. To identify strong oscillatory events we computed the spectrogram with a short-time fourier transform and calculated the mean spectral power between 0.5 to 10 Hz. Potential epileptiform events were then identified when the mean spectral power remained above 2 standard deviations for at least 10 seconds. This identified a total of 8 putative epileptiform events between 18.2 and 39.5 seconds across all Dnmt1 KO recordings whereas no events were detected in WT mice. To ensure that electrophysiological results were not affected by this activity, we then excluded all trials during, as well as 15 seconds before and after each event from all subsequent analyses. However, the impact of this correction on our results was negligible and all our earlier results remained unaffected. We have now added a description of this correction to the methods and updated the data in Fig. 6F-L and Extended Data Fig. 11C-D with the corresponding captions to reflect these new results.

All of the new negative results (behavior, SST activity) and much of the normal histology are relegated to supplementary figures. And yet, based on those results, a different lab might have concluded that Dnmt1 knockdown has no overt phenotype. It would be good for the authors to at least acknowledge this caveat or alternate interpretation of their results in the Discussion.

We addressed this point in the discussion and lowered the tone of our interpretation.

Perhaps the bigger conundrum is to explain how deleting this critical gene from SST neurons, which eventually does not affect these interneurons, somehow causes problems to other neurons in the superficial cortical layers. 'Non-cell-autonomous' sounds cool, but it's hard to wrap my head around it when there are no cell-type-specific phenotypes to speak of. The authors speculate on possible theories, but it's mostly mental gymnastics. Since SST-INs are known to fine-tune network scaffolding and activity in both infragranular and supragranular layers (PMID: 27225074), perhaps the authors can speculate on why these non-cell-autonomous changes are restricted to superficial cortical layers, given inside-out cortical development and how deep layer SST-IN can regulate superficial layer connectivity (PMID: 38926335, 26844832).

The non-cell autonomous function actually refers to the embryonic effect of altered migration pattern/expression of signaling molecules such as efnB2 in Dnmt1 deficient SST interneurons, which then affects the division mode of cortical progenitors, resulting in an increased generation of EOMES+ IPCs and deep layer neurons at E14.5/E16.5. This is in line with the increased thickness of layer 5 and 6 seen in adults (and at E18.5). In line with other studies showing that increased generation of deep layers comes at the expense of neurons destined for the upper layers (Gerstmann et al., 2015; Development), the upper layers were found reduced. This in turn is in line with the increased SST+cIN density (as absolute numbers do not change), which in our theory disturbs the balance of excitation

and inhibition and the overall cortical network responses, which could contribute to the behavioral abnormalities.

Again, this paper represents an impressive effort and the results, if not definitive, are certainly interesting. The authors have gone to great lengths to improve the manuscript based on reviewer feedback.

We thank the reviewer for this appreciation of our work.

Reviewer #3 (Remarks to the Author):

Reviewer #4 (Remarks to the Author):

We thank the authors for incorporating our comments and applying the necessary changes to the manuscript, which has greatly improved from the original version. However, the following comments need to be addressed by the authors.

In Figure 1i, the color code for the scale is missing

We have included the color code.

The authors mention that they discuss possible compensatory effects of Dnmt3 attributing to the lack of anti-correlation between DNA methylation and transcription. If that's the case, then the authors should definitely tone down the claim of the crucial role of Dnmt1 in the functional changes reported in this study.

We agree with the reviewer and included respective phrasing in the discussion.

In the rebuttal letter, the authors also discuss that DNA methylation does not always mean more transcription, and that the biological relevance of DNA methylation extends beyond transcription regulation. References to these statements are missing.

We have included this information together with respective references in the discussion of the revised manuscript.

Please provide a scheme of the experimental layout for panels p-s: The experimental layout of this experiment, which is now Figure 4 c-f, is still missing

We agree that this suggestion is very helpful, and now have included a scheme.

To further support the pair cell assay results we included a SOX2 staining and analysis of the SST-CRe/DNMT1 KO and control embryos at E14.5 in the supplements to show that this population is only mildly altered, while the EOMES pool and the newly generated neurons are.